# On Statistical Rates and Provably Efficient Criteria of Latent Diffusion Transformers (DiTs)

**Jerry Yao-Chieh Hu**[*†‡]   **Weimin Wu**[*†‡]   **Zhuoru Li**[♭]

**Sophia Pi**[‡]   **Zhao Song**[§]   **Han Liu**[†‡♮]

[†]Center for Foundation Models and Generative AI, [‡]Department of Computer Science, [♭]Department of Statistics and Data Science, Northwestern University, Evanston, IL 60208, USA
[§]Simons Institute for the Theory of Computing, UC Berkeley, Berkeley, CA 94720, USA

{jhu,wwm}@u.northwestern.edu;
magic.linuxkde@gmail.com; hanliu@northwestern.edu

## Abstract

We investigate the statistical and computational limits of latent Diffusion Transformers (DiTs) under the low-dimensional linear latent space assumption. Statistically, we study the universal approximation and sample complexity of the DiTs score function, as well as the distribution recovery property of the initial data. Specifically, under mild data assumptions, we derive an approximation error bound for the score network of latent DiTs, which is sub-linear in the latent space dimension. Additionally, we derive the corresponding sample complexity bound and show that the data distribution generated from the estimated score function converges toward a proximate area of the original one. Computationally, we characterize the hardness of both forward inference and backward computation of latent DiTs, assuming the Strong Exponential Time Hypothesis (SETH). For forward inference, we identify efficient criteria for all possible latent DiTs inference algorithms and showcase our theory by pushing the efficiency toward almost-linear time inference. For backward computation, we leverage the low-rank structure within the gradient computation of DiTs training for possible algorithmic speedup. Specifically, we show that such speedup achieves almost-linear time latent DiTs training by casting the DiTs gradient as a series of chained low-rank approximations with bounded error. Under the low-dimensional assumption, we show that the statistical rates and the computational efficiency are all dominated by the dimension of the subspace, suggesting that latent DiTs have the potential to bypass the challenges associated with the high dimensionality of initial data.

## 1 Introduction

We investigate the statistical and computational limits of latent diffusion transformers (DiTs), assuming the data is supported on an unknown low-dimensional linear subspace. This analysis is not only practical but also timely. On one hand, DiTs have demonstrated revolutionary success in generative AI and digital creation by using Transformers as score networks [Esser et al., 2024, Ma et al., 2024, Chen et al., 2024a, Mo et al., 2023, Peebles and Xie, 2023]. On the other hand, they require significant computational resources [Liu et al., 2024], making them challenging to train outside of specialized industrial labs. Therefore, it is natural to ask whether it is possible to make them lighter and faster without sacrificing performance. Answering these questions requires a fundamental understanding of the DiT architecture. This work provides a timely theoretical analysis of the fundamental limits of DiT architecture, aided by the analytical feasibility provided by the low-dimensional data assumption.

---

[*]Equal contribution. Version: January 3, 2025. Future updates are on arXiv.

Empirically, Latent Diffusion is a go-to design for effectiveness and computational efficiency [Rombach et al., 2022, Liu et al., 2021, Pope et al., 2021, Su and Wu, 2018]. Theoretically, it is capable of hosting the assumption of low-dimensional data structure (see Assumption 2.1 for formal definition) for detailed analytical characterization [Chen et al., 2023, Bortoli, 2022]. In essence, diffusion models with low-dimensional data structures manifest a natural lower-dimensional diffusion process through an encoder/decoder within a robust and informative latent representation feature space [Rombach et al., 2022, Pope et al., 2021]. Such lower-dimensional diffusion improves computational efficiency by reducing data complexity without sacrificing essential information [Liu et al., 2021]. With this assumption, Chen et al. [2023] decompose the score function of U-Net based diffusion models into on-support and orthogonal components. This decomposition allows for the characterization of the distinct behaviors of the two components: the on-support component facilitates latent distribution learning, while the orthogonal component facilitates subspace recovery.

In our work, we utilize low-dimensional data structure assumption to explore statistical and computational limits of latent DiTs. Our analysis includes the characterizations of statistical rates and provably efficient criteria. Statistically, we pose two questions and provide a theory to characterize the statistical rates of latent DiT under the assumption of a low-dimensional data:

**Question 1.** What is the approximation limit of using transformers to approximate the DiT score function, particularly in the low-dimensional data subspace?

**Question 2.** How accurate is the estimation limit for such a score estimator in practical training scenarios? With the score estimator, how well can diffusion transformers recover the data distribution?

Computationally, the primary challenge of DiT lies in the transformer blocks' quadratic complexity. This computational burden applies to both inference and training, even with latent diffusion. Thus, it is essential to design algorithms and methods to circumvent this $\Omega(L^2)$ where $L$ is the latent DiT sequence length. However, there are no formal results to support and characterize such algorithms. To address this gap, we pose the following questions and provide a fundamental theory to fully characterize the complexity of latent DiT under the low-dimensional linear subspace data assumption:

**Question 3.** Is it possible to improve the $\Omega(L^2)$ time complexity with a bounded approximation error for both forward and backward passes? What is the computational limit for such an improvement?

**Contributions.** We study the fundamental limits of latent DiT. Our contributions are threefold:

- **Score Approximation.** We address Question 1 by characterizing the approximation limit of matching the DiT score function with a transformer-based score estimator. Specifically, under mild data assumptions, we derive an approximation error bound for the score network, sub-linear in the latent space dimension (Theorem 3.1). These results not only explain the expressiveness of latent DiT (under mild assumptions) but also provide guidance for the structural configuration of the score network for practical implementations (Theorem 3.1).

- **Score and Distribution Estimation.** We address Question 2 by exploring the limitations of score and distribution estimations of latent DiTs in practical training scenarios. Specifically, we provide a sample complexity bound for score estimation (Theorem 3.2), using norm-based covering number bound of transformer architecture. Additionally, we show that the learned score estimator is able to recover the initial data distribution (Corollary 3.2.1).

- **Provably Efficient Criteria and Existence of Almost Linear Time Algorithms.** We address Question 3 by providing provably efficient criteria for latent DiTs in both forward inference and backward computation/training. For forward inference, we characterize all possible efficient DiT algorithms using a norm-based efficiency threshold for both conditional and unconditional generation (Proposition 4.1). Efficient algorithms, including almost-linear time algorithms (Proposition 4.2), are possible only below this threshold. For backward computation, we prove the existence of almost-linear time DiT training algorithms (Theorem 4.1) by utilizing the inherent low-rank structure in DiT gradients through a chained low-rank approximation.

Interestingly, both our statistical and computational results are dominated by the subspace dimension under the low-dimensional assumption, suggesting that latent DiT can potentially bypass the challenges associated with the high dimensionality of initial data.

**Organization.** Section 2 includes background on score decomposition and Transformer-based score networks. Section 3 includes DiTs' statistical rates. Section 4 includes DiTs' provably efficient criteria. Section 5 includes concluding remarks. We defer discussions of related works to Appendix C.

**Notations.** We use lower case letters to denote vectors, e.g., $z \in \mathbb{R}^D$. $\|z\|_2$ and $\|z\|_\infty$ denote its Euclidean norm and Infinite norm respectively. We use upper case letters to denote matrix, e.g., $Z \in \mathbb{R}^{d \times L}$. $\|Z\|_2$, $\|Z\|_{\mathrm{op}}$, and $\|Z\|_F$ denote the 2-norm, operator norm and Frobenius norm respectively. $\|Z\|_{p,q}$ denotes the $p, q$-norm where the $p$-norm is over columns and $q$-norm is over rows. Given a function $f$, let $\|f(x)\|_{L^2} := (\int \|f(x)\|_2^2 \mathrm{d}x)^{1/2}$, and $\|f(\cdot)\|_{Lip} = \sup_{x \neq y}(\|f(x) - f(y)\|_2 / \|x - y\|_2)$. With a distribution $P$, we denote $\|f\|_{L^2(P)} = (\int_P \|f(x)\|_2^2 \mathrm{d}x)^{1/2}$ as the $L^2(P)$ norm. Let $f_\sharp P$ be a pushforward measure, i.e., for any measurable $\Omega$, $(f_\sharp P)(\Omega) = P(f^{-1}(\Omega))$. We use $\psi$ for (conditional) Gaussian density functions.

## 2 Background

This section reviews the ideas we built on, including an overview of diffusion models (Section 2.1), the score decomposition under the linear latent space assumption (Section 2.2), and the transformer backbone in DiT (Section 2.3).

### 2.1 Score-Matching Denoising Diffusion Models

We briefly review forward process, backward process and score matching in diffusion models.

**Forward and Backward Process.** In the **forward** process, Diffusion models gradually add noise to the original data $x_0 \in \mathbb{R}^D$, and $x_0 \sim P_0$. Let $x_t$ denote the noisy data at the timestamp $t$, with marginal distribution and destiny as $P_t$ and $p_t$. The conditional distribution $P(x_t|x_0)$ follows $N(\beta(t)x_0, \sigma(t)I_D)$, where $\beta(t) = \exp(-\int_0^t w(s)\mathrm{d}s/2)$, $\sigma(t) = 1 - \beta^2(t)$, and $w(t) > 0$ is a nondecreasing weighting function. In practice, the forward process terminates at a large enough $T$ such that $P_T$ is close to $N(0, I_D)$. In the **backward** process, we obtain $y_t$ by reversing the forward process. The generation of $y_t$ depends on the score function $\nabla \log p_t(\cdot)$. However, this is unknown in practice, we use a score estimator $s_W(\cdot, t)$ to replace $\nabla \log p_t(\cdot)$, where $s_W(\cdot, t)$ is usually a neural network with parameters $W$. See Appendix D.1 for the details.

**Score Matching.** To estimate the score function, we use the following loss

$$\min_W \int_{T_0}^T \gamma(t) \mathbb{E}_{x_t \sim P_t} \left[ \|s_W(x_t, t) - \nabla \log p_t(x_t)\|_2^2 \right] \mathrm{d}t,$$

where $\gamma(t)$ is the weight function, and $T_0$ is a small value to stabilize training and prevent score function from blowing up [Vahdat et al., 2021]. However, it is hard to compute $\nabla \log p_t(\cdot)$ with available data samples. Therefore, we minimize the equivalent denoising score matching objective

$$\min_W \int_{T_0}^T \gamma(t) \mathbb{E}_{x_0 \sim P_0} \left[ \mathbb{E}_{x_t|x_0} \left[ \|s_W(x_t, t) - \nabla_{x_t} \log \psi_t(x_t \mid x_0)\|_2^2 \right] \right] \mathrm{d}t, \tag{2.1}$$

where $\psi_t(x_t|x_0)$ is the transition kernel, then $\nabla_{x_t} \log \psi_t(x_t|x_0) = (\beta(t)x_0 - x_t)/\sigma(t)$.

To train the parameters $W$ in the score estimator $s_W(\cdot, t)$, we use the empirical version of (2.1). We select $n$ i.i.d. data samples $\{x_{0,i}\}_{i=1}^n \sim P_0$, and sample time $t_i$ $(1 \leq i \leq n)$ uniformly from interval $[T_0, T]$. Given $x_{0,i}$, we sample $x_{t_i}$ from $N(\beta(t_i)x_{0,i}, \sigma(t_i)I_D)$. The empirical loss is

$$\widehat{\mathcal{L}}(W) = \frac{1}{n} \sum_{i=1}^n \|s_W(x_{t_i}, t_i) - x_{0,i}\|_2^2. \tag{2.2}$$

For convenience of notation, we denote population loss $\mathcal{L}(W) = \mathbb{E}_{P_0}[\widehat{\mathcal{L}}(W)]$.

### 2.2 Score Decomposition in Linear Latent Space

In this part, we review the score decomposition in [Chen et al., 2023]. We consider that the $D$-dimensional input data $x$ supported on a $d_0$-dimensional subspace, where $d_0 \leq D$.

**Assumption 2.1** (Low-Dimensional Linear Latent Space). *Let $x$ denote the initial data at $t = 0$. $x$ has a latent representation via $x = Bh$, where $B \in \mathbb{R}^{D \times d_0}$ is an unknown matrix with orthonormal columns. The latent variable $h \in \mathbb{R}^{d_0}$ follows the distribution $P_h$ with a density function $p_h$.*

**Remark 2.1.** By "linear latent space," we mean that each entry of a given latent vector is a linear combination of the corresponding input, i.e., $x = Bh$. This is also known as the "low-dimensional data" assumption in literature [Chen et al., 2023].

Let $\bar{x}$ and $\bar{h}$ denote the perturbed data and its associated latent variable at $t > 0$, respectively. Based on the low-dimensional data structure assumption, we have the following score decomposition theory: on-support score $s_+(B^\top \bar{x}, t)$ and orthogonal score $s_-(\bar{x}, t)$.

**Lemma 2.1** (Score Decomposition, Lemma 1 of [Chen et al., 2023]). Let data $x = Bh$ follow Assumption 2.1. The decomposition of score function $\nabla \log p_t(\bar{x})$ is

$$\nabla \log p_t(\bar{x}) = \underbrace{B\nabla \log p_t^h(\bar{h})}_{s_+(\bar{h},t)} - \underbrace{\left(I_D - BB^\top\right)\bar{x}/\sigma(t)}_{s_-(\bar{x},t)}, \quad \bar{h} = B^\top \bar{x}, \tag{2.3}$$

where $p_t^h(\bar{h}) := \int \psi_t(\bar{h}|h)p_h(h)\mathrm{d}h$, $\psi_t(\cdot|h)$ is the Gaussian density function of $N(\beta(t)h, \sigma(t)I_{d_0})$, $\beta(t) = e^{-t/2}$ and $\sigma(t) = 1 - e^{-t}$. We restate the proof in Appendix D.2 for completeness.

Additionally, our theoretical analysis is based on two following assumptions as in [Chen et al., 2023].

**Assumption 2.2** (Tail Behavior of $P_h$). The density function $p_h > 0$ is twice continuously differentiable. Moreover, there exist positive constants $A_0, A_1, A_2$ such that when $\|h\|_2 \geq A_0$, the density function $p_h(h) \leq (2\pi)^{-d_0/2}A_1\exp(-A_2\|h\|_2^2/2)$.

**Assumption 2.3** ($L_{s_+}$-Lipschitz of $s_+(\bar{h}, t)$). The on-support score function $s_+(\bar{h}, t)$ is $L_{s_+}$-Lipschitz in $\bar{h} \in \mathbb{R}^{d_0}$ for any $t \in [0, T]$.

## 2.3 Score Network and Transformers

In this part, we introduce the score network architecture and Transformers. Transformers are the backbone of the score network in DiT. By Assumption 2.1, $\bar{h} = B^\top \bar{x} \in \mathbb{R}^{d_0}$ with $d_0 < D$.

**(Latent) Score Network.** Following [Chen et al., 2023], we rearrange (2.3) into

$$\nabla \log p_t(\bar{x}) = B\underbrace{(\sigma(t)\nabla \log p_t^h(B^\top \bar{x}) + B^\top \bar{x})}_{:=q(B^\top \bar{x},t):\; \mathbb{R}^{d_0} \times [T_0, T] \to \mathbb{R}^{d_0}}/\sigma(t) - \bar{x}/\sigma(t). \tag{2.4}$$

We use $W_B \in \mathbb{R}^{D\times d_0}$ to approximate $B \in \mathbb{R}^{D\times d_0}$, and a neural network $f(W_B^\top \bar{x}, t)$ to approximate $q(B^\top \bar{x}, t)$. We adopt the following score network class for diffusion in latent space (i.e., in $\bar{h} \in \mathbb{R}^{d_0}$)

$$\mathcal{S} = \left\{s_W(\bar{x}, t) = W_B f(W_B^T \bar{x}, t)/\sigma(t) - \bar{x}/\sigma(t),\ W = \{W_B, f\}\right\}, \tag{2.5}$$

where the columns in $W_B$ are orthogonal, $f: \mathbb{R}^{d_0} \times [T_0, T] \to \mathbb{R}^{d_0}$ is a neural network. In this work, we focus on the diffusion transformers (DiTs), i.e., using Transformer for $f$ [Peebles and Xie, 2023].

**Transformers.** A Transformer block consists of a self-attention layer and a feed-forward layer, with both layers having skip connection. We use $\tau^{r,m,l}: \mathbb{R}^{d\times L} \to \mathbb{R}^{d\times L}$ to denote a Transformer block. Here $r$ and $m$ are the number of heads and head size in self-attention layer, and $l$ is the hidden dimension in feed-forward layer. Let $X \in \mathbb{R}^{d\times L}$ be the model input, then we have the model output

$$\mathrm{Attn}(X) = X + \sum_{i=1}^{r} W_O^i W_V^i X \cdot \mathrm{Softmax}\left((W_K^i X)^\top W_Q^i X\right), \tag{2.6}$$

$$\mathrm{FF} \circ \mathrm{Attn}(X) = \mathrm{Attn}(X) + W_2 \cdot \mathrm{ReLU}(W_1 \cdot \mathrm{Attn}(X) + b_1 \mathbb{1}_L^\top) + b_2 \mathbb{1}_L^\top, \tag{2.7}$$

where $W_K^i, W_Q^i, W_V^i \in \mathbb{R}^{m\times d}, W_O^i \in \mathbb{R}^{d\times m}, W_1 \in \mathbb{R}^{l\times d}, W_2 \in \mathbb{R}^{d\times l}, b_1 \in \mathbb{R}^l, b_2 \in \mathbb{R}^d$.

In our work, we use Transformer networks with positional encoding $E \in \mathbb{R}^{d\times L}$. We define the Transformer networks as the composition of Transformer blocks

$$\mathcal{T}_P^{r,m,l} = \{f_{\mathcal{T}}: \mathbb{R}^{d\times L} \to \mathbb{R}^{d\times L} \mid f_{\mathcal{T}} \text{ is a composition of blocks } \tau^{r,m,l}\text{'s}\}.$$

For example, the following is a Transformer network consisting $K$ blocks and positional encoding

$$f_{\mathcal{T}}(X) = \mathrm{FF}^{(K)} \circ \mathrm{Attn}^{(K)} \circ \cdots \mathrm{FF}^{(1)} \circ \mathrm{Attn}^{(1)}(X + E). \tag{2.8}$$

# 3 Statistical Rates of Latent DiTs with Subspace Data Assumption

In this section, we analyze the statistical rates of latent DiTs. Section 3.1 introduces the class of latent DiT score networks. In Section 3.2, we prove the approximation limit of matching the DiT score function with the score network class, and characterize the structural configuration of the score network when a specified approximation error is required. Following this, in Section 3.3, utilizing the characterized structural configuration, we prove the score and distribution estimation for latent DiTs.

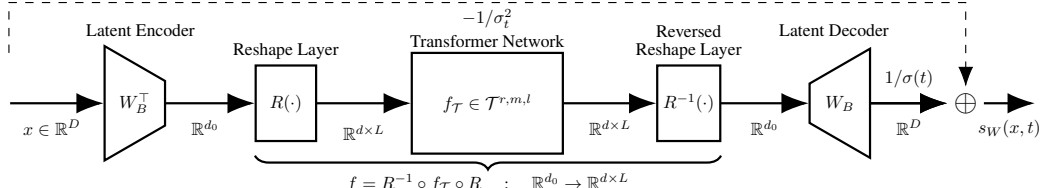

Figure 1: **Overview of DiT Score Network Architecture** $s_W(\cdot, t)$. $W_B^T$ denotes the linear layer from the input data space to the linear latent space. $f(\cdot) = R^{-1} \circ f_{\mathcal{T}} \circ R(\cdot)$ denotes the transformer network $f_{\mathcal{T}}(\cdot)$ with reshaping layer $R(\cdot)$, where $f_{\mathcal{T}}(\cdot) \in \mathcal{T}_p^{r,m,l}$. $W_B$ denotes the linear layer from the linear latent space to the input data space. $\sigma(t)$ denote the variance of the conditional distribution $P(x_t \mid x_0)$.

## 3.1 DiT Score Network Class

Here, we provide the details about DiT score network class used in our analysis. In (2.5), $f$ is a network with Transformer as the backbone, and $(\bar{h}, t) \in \mathbb{R}^{d_0} \times [T_0, T]$ denotes the input data. Following [Peebles and Xie, 2023], DiT uses time point $t$ to calculate the scale and shift value in the Transformer backbone, and it transforms an input picture into a sequential version. To achieve the transformation, we introduce a reshape layer.

**Definition 3.1** (DiT Reshape Layer $R(\cdot)$). Let $R(\cdot) : \mathbb{R}^{d_0} \to \mathbb{R}^{d \times L}$ be a reshape layer that transforms the $d_0$-dimensional input into a $d \times L$ matrix. Specifically, for any $d_0 = i \times i$ image input, $R(\cdot)$ converts it into a sequence representation with feature dimension $d := p^2$ (where $p \geq 2$) and sequence length $L := (i/p)^2$. Besides, we define the corresponding reverse reshape (flatten) layer $R^{-1}(\cdot) : \mathbb{R}^{d \times L} \to \mathbb{R}^{d_0}$ as the inverse of $R(\cdot)$. By $d_0 = dL$, $R, R^{-1}$ are associative w.r.t. their input.

To simplify the self-attention block in (2.6), let $W_{OV}^i = W_O^i W_V^i$ and $W_{KQ}^i = (W_K^i)^\top W_Q^i$.

**Definition 3.2** (Transformer Network Class $\mathcal{T}_p^{r,m,l}$). We define the Transformer network class as

$$\mathcal{T}_p^{r,m,l}(K, C_{\mathcal{T}}, C_{OV}^{2,\infty}, C_{OV}, C_{KQ}^{2,\infty}, C_{KQ}, C_F^{2,\infty}, C_F, C_E, L_{\mathcal{T}}), \text{ satisfying the constraints}$$

- Model architecture with $K$ blocks: $f_{\mathcal{T}}(X) = \text{FF}^{(K)} \circ \text{Attn}^{(K)} \circ \cdots \text{FF}^{(1)} \circ \text{Attn}^{(1)}(X)$;
- Model output bound: $\sup_X \|f_{\mathcal{T}}(X)\|_2 \leq C_{\mathcal{T}}$;
- Parameter bound in $\text{Attn}^{(i)}$: $\left\|(W_{OV}^i)^\top\right\|_{2,\infty} \leq C_{OV}^{2,\infty}$, $\left\|(W_{OV}^i)^\top\right\|_2 \leq C_{OV}$, $\left\|W_{KQ}^i\right\|_{2,\infty} \leq C_{KQ}^{2,\infty}$, $\left\|W_{KQ}^i\right\|_2 \leq C_{KQ}$, $\left\|E^\top\right\|_{2,\infty} \leq C_E, \forall i \in [K]$;
- Parameter bound in $\text{FF}^{(i)}$: $\left\|W_j^i\right\|_{2,\infty} \leq C_F^{2,\infty}$, $\left\|W_j^i\right\|_2 \leq C_F, \forall j \in [2], i \in [K]$;
- Lipschitz of $f_{\mathcal{T}}$: $\|f_{\mathcal{T}}(X_1) - f_{\mathcal{T}}(X_2)\|_F \leq L_{\mathcal{T}} \|X_1 - X_2\|_F, \forall X_1, X_2 \in \mathbb{R}^{d \times L}$.

**Definition 3.3** (DiT Score Network Class $\mathcal{S}_{\mathcal{T}_p^{r,m,l}}$ (Figure 1)). We denote $\mathcal{S}_{\mathcal{T}_p^{r,m,l}}$ as the DiT score network class in (2.5), replacing $f$ with $R^{-1} \circ f_{\mathcal{T}} \circ R$, and $f_{\mathcal{T}}$ is from the Transformer class $\mathcal{T}_p^{r,m,l}$.

## 3.2 Score Approximation of DiT

Here, we explore the approximation limit of latent DiT score network class $\mathcal{S}_{\mathcal{T}_p^{r,m,l}}$ under linear latent space assumption. Recall that $P_t$ is the distribution of $x_t$, $\sigma(t)$ is the variance of $P(x_t|x_0)$, $d_0$ is the dimension of latent space, $L$ is the sequence length of transformer input, $T$ is the stopping time in forward process, $T_0$ is the early stopping time in backward process, and $L_{s_+}$ is the Lipschitz coefficient of on-support score function. Then we have the following Theorem 3.1.

**Theorem 3.1** (Score Approximation of DiT). For any approximation error $\epsilon > 0$ and any data distribution $P_0$ under Assumptions 2.1 to 2.3, there exists a DiT score network $s_{\widehat{W}}$ from $\mathcal{S}_{\mathcal{T}_p^{2,1,4}}$ (defined in Definition 3.2), where $\widehat{W} = \{\widehat{W}_B, \widehat{f}_{\mathcal{T}}\}$, such that for any $t \in [T_0, T]$, we have:

$$\left\|s_{\widehat{W}}(\cdot, t) - \nabla \log p_t(\cdot)\right\|_{L^2(P_t)} \leq \epsilon \cdot \sqrt{d_0}/\sigma(t),$$

where $\sigma(t) = 1 - e^{-t}$, and the upper bound of hyperparameters in $\mathcal{S}_{\mathcal{T}_p^{2,1,4}}$ are

$$K = \mathcal{O}(\epsilon^{-2L}), \ C_{\mathcal{T}} = \mathcal{O}\left(d_0 L_{s_+} \sqrt{d_0 \log(d_0/T_0) + \log(1/\epsilon)}\right),$$

$$C_{OV}^{2,\infty} = (1/\epsilon)^{\mathcal{O}(1)}, \; C_{OV} = (1/\epsilon)^{\mathcal{O}(1)}, \; C_{KQ}^{2,\infty} = (1/\epsilon)^{\mathcal{O}(1)}, \; C_{KQ} = (1/\epsilon)^{\mathcal{O}(1)},$$
$$C_E = \mathcal{O}(L^{3/2}), \; C_F^{2,\infty} = (1/\epsilon)^{\mathcal{O}(1)}, \; C_F = (1/\epsilon)^{\mathcal{O}(1)}, \; L_{\mathcal{T}} = \mathcal{O}\left(d_0 L_{s_+}\right).$$

*Proof Sketch.* Our proof is built on the key observation that there is a tail behavior of the low-dimensional latent variable distribution $P_h$ (Assumption 2.2). Recall that $\nabla \log p_t(\bar{x}) = Bq(\bar{h}, t)/\sigma(t) - \bar{x}/\sigma(t)$, where $\bar{h} = B^{\top}\bar{x}$ (defined in (2.4)). By taking $\widehat{W}_B = B$, our aim reduces to construct a transformer network to approximate $q(\bar{h}, t)$. To achieve this, we firstly approximate $q(\bar{h}, t)$ with a compact-supported continuous function, based on the tail behavior of $P_h$. Then we construct a transformer to approximate the compact-supported continuous function using the universal approximation capacity of transformer [Yun et al., 2020]. See Appendix F.1 for a detailed proof. □

Intuitively, Theorem 3.1 indicates the capability of the transformer-based score network to approximate the score function with precise guarantees. Furthermore, Theorem 3.1 provides empirical guidance for the design choices of the score network when a specified approximation error is required.

**Remark 3.1** (Comparing with Existing Works). Theoretical analysis of DiTs is limited. Previous works that do not specify the model architecture assume that the score estimator is well-approximated [Benton et al., 2024, Wibisono et al., 2024]. To the best of our knowledge, this work is the first to present an approximation theory for DiTs, offering the estimation theory in Theorem 3.2 and Corollary 3.2.1 based on the estimated score network, rather than a perfectly trained one.

**Remark 3.2** (Latent Dimension Dependency). Theorem 3.1 suggests that the approximation capacity and Transformer network size primarily depend on the latent variable dimension $d_0 = d \times L$. This indicates that DiTs can potentially bypass the challenges associated with the high dimensionality of initial data by transforming input data into a low-dimensional latent variable.

### 3.3   Score Estimation and Distribution Estimation

Besides score approximation capability, Theorem 3.1 also characterizes the structural configuration of the score network for any specific precision, e.g., $K, C_E, C_F$, etc. This characterization enables further analysis of the performance of score network in practical scenarios. In Theorem 3.2, we provide a sample complexity bound for score estimation. In Corollary 3.2.1, show that the learned score estimator is able to recover the initial data distribution.

**Score Estimation.**   To derive a sample complexity for score estimation using $\mathcal{S}_{\mathcal{T}_p^{2,1,4}}$, we rewrite the score matching objective in (2.2) as $\widehat{W} \in \arg\min_{s_W \in \mathcal{S}_{\mathcal{T}_p^{2,1,4}}} \widehat{\mathcal{L}}(s_W)$, $\widehat{W} = \{\widehat{W}_B, \widehat{f}_{\mathcal{T}}\}$.

Theorem 3.2 shows that as sample size $n \to \infty$, $s_W(\cdot, t)$ convergences to $\nabla \log p_t(\cdot)$.

**Theorem 3.2** (Score Estimation of DiT). Under Assumptions 2.1 to 2.3, we choose $\mathcal{S}_{\mathcal{T}_p^{2,1,4}}$ as in Theorem 3.1 using $\epsilon \in (0, 1)$ and $L > 1$, With probability $1 - 1/\text{poly}(n)$, we have

$$\frac{1}{T - T_0} \int_{T_0}^{T} \left\| s_{\widehat{W}}(\cdot, t) - \nabla \log p_t(\cdot) \right\|_{L^2(P_t)} \mathrm{d}t = \widetilde{\mathcal{O}}\left( \frac{1}{n^{1/3}T_0 T} \cdot 2^{(1/\epsilon)^{2L}} + \frac{1}{n^{1/3}T_0 T} + \frac{1}{T_0 T}\epsilon^2 \right), \tag{3.1}$$

where $\widetilde{\mathcal{O}}$ hides the factors related to $D, d_0, d, L_{s_+}$, and $\log n$.

*Proof.* See Appendix F.2 for a detailed proof. □

Intuitively, Theorem 3.2 shows a sample complexity bound for score estimation in practice.

**Remark 3.3** (Comparing with Existing Works). [Zhu et al., 2023] provides a sample complexity for simple ReLU-based diffusion models under the assumption of an accurate score estimator. To the best of our knowledge, we are the first to provide a sample complexity for DiTs, based on the learned score network in Theorem 3.1 and the quantization (piece-wise approximation) approach for transformer universality [Yun et al., 2020]. Furthermore, our first term shows a convergence rate of $1/T$, outperforming [Chen et al., 2023], in which the first term is independent of $T$.

**Remark 3.4** (Double Exponential Factor and Inconsistent Convergence). Theorem 3.2 reports an explicit result on sample complexity bounds for score estimation of latent DiTs: a double exponential factor $2^{(1/\epsilon)^{2L}}$ in the first term. We remark that this arises from the required depth $K$ is $\mathcal{O}(\epsilon^{-2L})$, and the norm of required weight parameters is $(1/\epsilon)^{\mathcal{O}(1)}$ as shown in Theorem 3.1, assuming the universality of transformers requires dense layers [Yun et al., 2020]. This double exponential factor causes inconsistent convergence with respect to sample size $n$, as its large value prevents setting $\epsilon$ as a function of $n$ to balance the first and second terms in (3.1). This motivates us to rethink transformer universality and explore new proof techniques for DiTs, which we leave for future work.

**Definition 3.4.** For later convenience, we define $\xi(n, \epsilon, L) := \frac{1}{n^{1/3}} \cdot 2^{(1/\epsilon)^{2L}} + \frac{1}{n^{1/3}} + \epsilon^2$.

**Distribution Estimation.** In practice, DiTs generate data using the discretized version with step size $\mu$, see Appendix D.1 for details. Let $\widehat{P}_{T_0}$ be the distribution generated by $s_{\widehat{W}}$ using the discretized backward process in Theorem 3.2. Let $P_{T_0}^h$ and $p_{T_0}^h$ be the distribution and density function of on-support latent variable $\bar{h}$ at $T_0$. We have the following results for distribution estimation.

**Corollary 3.2.1** (Distribution Estimation of DiT, Modified From Theorem 3 of [Chen et al., 2023]). Let $T = \mathcal{O}(\log n), T_0 = \mathcal{O}(\min\{c_0, 1/L_{s_+}\})$, where $c_0$ is the minimum eigenvalue of $\mathbb{E}_{P_h}[hh^\top]$. With the estimated DiT score network $s_{\widehat{W}}$ in Theorem 3.2, we have the following with probability $1 - 1/\text{poly}(n)$.

(i) The accuracy to recover the subspace $B$ is $\left\|W_B W_B^\top - BB^\top\right\|_F^2 = \widetilde{\mathcal{O}}\left(\xi(n, \epsilon, L)/c_0\right)$.

(ii) With the conditions $\mathsf{KL}(P_h \| N(0, I_{d_0})) < \infty$, there exists an orthogonal matrix $U \in \mathbb{R}^{d \times d}$ such that we have the following upper bound for the total variation distance
$$\mathsf{TV}(P_{T_0}^h, (W_B U)_\sharp^\top \widehat{P}_{T_0}) = \widetilde{\mathcal{O}}(\sqrt{\xi(n, \epsilon, L) \cdot \log n}), \tag{3.2}$$
where $\widetilde{\mathcal{O}}$ hides the factor about $D, d_0, d, L_{s_+}, \log n$, and $T - T_0$. and $(W_B U)_\sharp^\top \widehat{P}_{T_0}$ denotes the pushforward distribution.

(iii) For the generated data distribution $\widehat{P}_{T_0}$, the orthogonal pushforward $(I - W_B W_B^\top)_\sharp \widehat{P}_{T_0}$ is $N(0, \Sigma)$, where $\Sigma \preceq aT_0 I$ for a constant $a > 0$.

*Proof.* See Appendix F.3 for a detailed proof. □

Intuitively, Corollary 3.2.1 shows the estimation results in 3 parts: (i) the accuracy of recovering the subspace $B$; (ii) the estimation error between $\widehat{P}_{T_0}$ and $P_{T_0}^h$; and (iii) the vanishing behavior of $\widehat{P}_{T_0}$ in the orthogonal space. These indicate that the learned score estimator is capable of recovering the initial data distribution. Notably, Corollary 3.2.1 is agnostic to the specifics of $\xi(n, \epsilon, L)$.

**Remark 3.5** (Comparing with Existing Works). Oko et al. [2023] analyze the distribution estimation under the assumption that the initial density is supported on $[-1, 1]^D$ and smooth in the boundary. Our Assumption 2.2 demonstrates greater practical relevance. This suggests that our method of distribution estimation aligns more closely with empirical realities.

**Remark 3.6** (Subspace Recovery Accuracy). (i) of Corollary 3.2.1 confirms that the subspace is learned by DiTs. The error is proportional to the sample complexity for score estimation and depends on the minimum eigenvalue of the covariance of $P_h$.

## 4 Provably Efficient Criteria

Here, we analyze the computational limits of latent DiTs under low-dimensional linear subspace data assumption (i.e., Assumption 2.1). The hardness of DiT models ties to both forward and backward passes of the score network in Definition 3.3. We characterize them separately.

### 4.1 Computational Limits of Backward Computation

Following Section 2, suppose we have $n$ i.i.d. data samples $\{x_{0,i}\}_{i=1}^n \sim P_d$, and time $t_{i_0}$ ($1 \le i \le n$) uniformly sampled from $[T_0, T]$. For each data $x_{0,i} \in \mathbb{R}^D$, we sample $x_{t_{i_0}} \in \mathbb{R}^D$ from $N(\beta(t_{i_0})x_{0,i}, \sigma(t_{i_0})I_D)$. Let $(W_A R^{-1}(\cdot))^\dagger$ be the inverse transformation of $W_A R^{-1}(\cdot)$, and denote

$Y_{0,i} := (W_A R^{-1})^\dagger (x_{0,i}) \in \mathbb{R}^{d \times L}$. We rewrite the empirical denoising score-matching loss (2.2) as

$$\frac{1}{n}\sum_{i=1}^{n}\left\|W_A R^{-1}(f_{\mathcal{T}}(R(\underbrace{W_A^\top x_{t_{i_0}}}_{d_0 \times 1}))) - x_{0,i}\right\|_F^2 = \frac{1}{n}\sum_{i=1}^{n}\left\|\underbrace{W_A}_{D \times d_0} R^{-1}(\overbrace{\underbrace{f_{\mathcal{T}}(R(W_A^\top x_{t_{i_0}}))}_{d_0 \times 1}}^{d \times L}) - \underbrace{Y_{0,i}}_{d \times L})\right\|_F^2. \tag{4.1}$$

For efficiency, it suffices to focus on just transformer attention heads of the DiT score network due to their dominating quadratic time complexity in both passes. Thus, we consider only a single layer attention for $f_{\mathcal{T}}$, to simplify our analysis. Further, we consider the following simplifications:

(S0) To prove the hardness of (4.1) for both full gradient descent and stochastic mini-batch gradient descent methods, it suffices to consider training on a single data point.

(S1) For the convenience of our analysis, we consider the following expression for attention mechanism. Let $X, Y \in \mathbb{R}^{d \times L}$. Let $W_K, W_Q, W_V \in \mathbb{R}^{s \times d}$ be attention weights such that $Q = W_Q X \in \mathbb{R}^{d \times L}$, $K = W_K X \in \mathbb{R}^{s \times L}$ and $V = W_V X \in \mathbb{R}^{s \times L}$. We write attention mechanism of hidden size $s$ and sequence length $L$ as

$$\text{Att}(X) = \underbrace{(W_O W_V X)}_{V \text{ multiplication}} \underbrace{D^{-1} \exp(X^\top W_K^\top W_Q X)}_{K\text{-}Q \text{ multiplication}} \in \mathbb{R}^{d \times L}, \tag{4.2}$$

with $D := \text{diag}(\exp(X W_Q W_K^\top X^\top)\mathbb{1}_L)$. Here, $\exp(\cdot)$ is entry-wise exponential function, i.e., $\exp(A)_{i,j} = \exp(A_{i,j})$ for any matrix $A$, $\text{diag}(\cdot)$ converts a vector into a diagonal matrix with the vector's entries on the diagonal, and $\mathbb{1}_L$ is the length-$L$ all ones vector.

(S2) Since $V$ multiplication is linear in weight while $K$-$Q$ multiplication is exponential in weights, we only need to focus on the gradient update of $K$-$Q$ multiplication. Therefore, for efficiency analysis of gradient, it is equivalent to analyzing a reduced problem with fixed $W_O W_V X =$ const..

(S3) To focus on the DiT, we consider the low-dimensional linear encoder $W_A$ to be pretrained and to not participate in gradient computation. This aligns with common practice [Rombach et al., 2022] and is justified by the trivial computation cost due to the linearity of $W_A$[2].

(S4) To further simplify, we introduce $A_1, A_2, A_3 \in \mathbb{R}^{s \times L}$ and $W \in \mathbb{R}^{d \times d}$ via

$$\left\|W_A R^{-1}\left(f_{\mathcal{T}}(\underbrace{R(W_A^\top x_{t_{i_0}})}_{:=X \in \mathbb{R}^{d \times L}}) - \underbrace{Y_{0,i}}_{:=Y \in \mathbb{R}^{d \times L}}\right)\right\|_F^2 \qquad (\text{By (S0), (S1) and (S2)})$$

$$= \left\|W_A R^{-1}\left(\underbrace{W_O W_V}_{:=W_{OV} \in \mathbb{R}^{d \times d}} \underbrace{X}_{:=A_3 \in \mathbb{R}^{d \times L}} D^{-1}\exp\left(\underbrace{X^\top}_{:=A_1^\top \in \mathbb{R}^{L \times d}}\underbrace{W_K^\top W_Q}_{:=W \in \mathbb{R}^{d \times d}}\underbrace{X}_{:=A_2 \in \mathbb{R}^{d \times L}}\right) - Y\right)\right\|_F^2. \tag{4.3}$$

Notably, $A_1, A_2, A_3, X, Y$ are constants w.r.t. training above loss with gradient updates.

Therefore, we simplify the objective of training DiT into

---

**Definition 4.1** (Training Generic DiT Loss). Given $A_1, A_2, A_3, Y \in \mathbb{R}^{d \times L}$ and $W_{OV}, W \in \mathbb{R}^{d \times d}$ following (S4), Training a DiT with $\ell_2$ loss on a single data point $X, Y \in \mathbb{R}^{d \times L}$ is formulated as

$$\min_W \mathcal{L}_0(W) = \min_W \frac{1}{2}\left\|W_A R^{-1}\left(W_{OV}A_3 D^{-1}\exp(A_1^\top W A_2) - Y\right)\right\|_F^2. \tag{4.4}$$

Here $D := \text{diag}(\exp(A_1^\top W A_2)\mathbb{1}_n) \in \mathbb{R}^{L \times L}$.

---

**Remark 4.1** (Conditional and Unconditional Generation). $\mathcal{L}_0$ is generic. If $A_1 \neq A_2 \in \mathbb{R}^{d \times L}$, Definition 4.1 reduces to cross-attention in DiT score net (for conditional generation). If $A_1 = A_2 \in \mathbb{R}^{d \times L}$, Definition 4.1 reduces to self-attention in DiT score net (for unconditional vanilla generation).

We introduce the next problem to characterize all possible gradient computations of optimizing (4.4).

---

[2]The gradient computation is linear in $W_A$, and hence the computation w.r.t. $W_A$ is cheap and upper-bounded by $L \cdot \text{poly}(d)$ time in a straightforward way.

**Problem 1** (Approximate DiT Gradient Computation (ADITGC($L, d, \Gamma, \epsilon$))). Given $A_1, A_2, A_3, Y \in \mathbb{R}^{d \times L}$. Let $\epsilon > 0$. Assume all numerical values are in $\mathcal{O}(\log(L))$-bits encoding. Let loss function $\mathcal{L}_0$ follow Definition 4.1. The problem of approximating gradient computation of optimizing empirical DiT loss (4.4) is to find an approximated gradient matrix $\widetilde{G}^{(W)} \in \mathbb{R}^{d \times d}$ such that $\left\| \widetilde{G}^{(W)} - \frac{\partial \mathcal{L}}{\partial \underline{W}} \right\|_{\max} \leq 1/\text{poly}(L)$. Here, $\|A\|_{\max} := \max_{i,j} |A_{ij}|$ for any matrix $A$.

In this work, we aim to investigate the computational limits of all possible efficient algorithms of ADITGC with $\epsilon = 1/\text{poly}(L)$. Yet, the explicit gradient of DiT denoising score matching loss (4.4) is too complicated to characterize ADITGC. To combat this, we make the following observations.

(O1) Let $g_1(\cdot) := W_A R^{-1}(\cdot) : \mathbb{R}^{d \times L} \to \mathbb{R}^{d_0}$, $g_2(\cdot) := \text{Att}(\cdot) : \mathbb{R}^{d \times L} \to \mathbb{R}^{d \times L}$, and $g_3(\cdot) := R(W_A^\top \cdot) : \mathbb{R}^D \to \mathbb{R}^{d \times L}$ such that $g_3(x) = X$ for $x \in \mathbb{R}^D$ (with $D > d_0 = dL$).

(O2) **Vectorization of $f_\mathcal{T}$.** For the ease of presentation, we use notation flexibly that $f_\mathcal{T}$ to denote both a matrix in $\mathbb{R}^{d \times L}$ and a vector in $\mathbb{R}^{dL}$ in the following analysis. This practice does not affect correctness. The context in which $f_\mathcal{T}$ is used should clarify whether it refers to a matrix or a vector. Explicit vectorization follows Definition D.1.

(O3) **Linearity of $g_1$.** By linearity of $W_A R^{-1}(\cdot)$, we treat $g_1$ as a matrix in $\mathbb{R}^{d_0 \times dL}$ acting on vector $f_\mathcal{T}(\cdot) \in \mathbb{R}^{dL}$.

Therefore, we have $\mathcal{L}_0 = \|g_1 \cdot [g_2(g_3) - Y]\|_2^2$, such that its gradient involves $\frac{\mathrm{d}\mathcal{L}_0}{\mathrm{d}W} = g_1 \frac{\mathrm{d}g_2}{\mathrm{d}W}$. From above, we only need to focus on proving the computation time and error control of term $\frac{\mathrm{d}g_2}{\mathrm{d}W}$ for gradient w.r.t $W$. Luckily, with tools from fine-grained complexity theory [Alman and Song, 2023, 2024a,b,c] and tensor trick (see Appendix D.3), we prove the existence of almost-linear time algorithms for Problem 1 in the next theorem. Let $\text{vec}(W) := \underline{W}$ for any matrix $W$ following Definition D.1.

**Theorem 4.1** (Existence of Almost-Linear Time Algorithms for ADITGC). Suppose all numerical values are in $\mathcal{O}(\log L)$-bits encoding. Let $\max(\|W_{OV} A_3\|_{\max}, \|W_K A_1\|_{\max}, \|W_Q A_2\|_{\max}) \leq \Gamma$. There exists a $L^{1+o(1)}$ time algorithm to solve ADITGC($L_p, L, d = \mathcal{O}(\log L), \Gamma = o(\sqrt{\log L})$) (i.e., Problem 1) with loss $\mathcal{L}_0$ from Definition 4.1 up to $1/\text{poly}(L)$ accuracy. In particular, this algorithm outputs gradient matrices $\widetilde{G}^{(W)} \in \mathbb{R}^{d \times d}$ such that $\left\| \widetilde{G}^{(W)} - \frac{\partial \mathcal{L}}{\partial \underline{W}} \right\|_{\max} \leq 1/\text{poly}(L)$.

*Proof Sketch.* Our proof is built on the key observation that there exist low-rank structures within the DiT training gradients. Using the tensor trick [Diao et al., 2019, 2018] and computational hardness results of attention [Hu et al., 2024b, Alman and Song, 2023], we approximate DiT training gradients with a series of low-rank approximations and carefully match the multiplication dimensions so that the computation of $\frac{\mathrm{d}g_2}{\mathrm{d}W}$ forms a chained low-rank approximation. We complete the proof by demonstrating that this approximation is bounded by a $1/\text{poly}(L)$ error and requires only almost-linear time. See Appendix G.2 for a detailed proof. $\qquad \square$

**Remark 4.2.** We remark that Theorem 4.1 is dominated by the relation between $L$ and $d$, hence by the subspace dimension[3] $d_0 = dL$. A smaller $d_0$ makes Theorem 4.1 more likely to hold.

### 4.2 Computational Limits of Forward Inference

Since the inference of score-matching diffusion models is a forward pass of the trained score estimator $s_W$, the computational hardness of DiT ties to the transformer-based score network,

$$s_W(A_1, A_2, A_3) = W_A R^{-1}\big( \underbrace{W_{OV} A_3}_{d \times L} \underbrace{D^{-1}}_{L \times L} \exp\big( \underbrace{A_1^\top W_K^\top}_{L \times s} \underbrace{W_Q A_2}_{s \times L} \big)\big), \tag{4.5}$$

following notation in Definition 4.1. For inference, we study the following approximation problem. Notably, by Remark 4.1, (4.5) subsumes both conditional and unconditional DiT inferences.

**Problem 2** (Approximate DiT Inference ADITI($d, L, \Gamma, \delta_F$)). Let $\delta_F > 0$ and $B > 0$. Given $A_1, A_2, A_3 \in \mathbb{R}^{d \times L}$, and $W_{OV}, W_K, W_Q \in \mathbb{R}^{d \times d}$ with guarantees that $\|W_{OV} A_3\|_\infty \leq B$, $\|W_K A_1\|_\infty \leq B$ and $\|W_Q A_2\|_\infty \leq B$, we aim to study an approximation problem

---

[3]See Assumption 2.1.

AD1TI$(d, L, B, \delta_F)$, that approximates $s_W(A_1, A_2, A_3)$ with a vector $\widetilde{z} \in \mathbb{R}^{d_0}$ (with $d_0 = d \cdot L$) such that $\left\| \widetilde{z} - W_A R^{-1} \left( W_{OV} A_3 D^{-1} \exp\left( A_1^\top W_K^\top W_Q A_2 \right) \right) \right\|_{\max} \leq \delta_F$. Here, $\|A\|_{\max} := \max_{i,j} |A_{ij}|$ for any matrix $A$.

By (O2) and (O3), we make an observation that Problem 2 is just a special case of [Alman and Song, 2023]. Hence, we characterize the all possible efficient algorithms for AD1TI with next proposition.

**Proposition 4.1** (Norm-Based Efficiency Phase Transition). Let $\|W_Q A_2\|_\infty \leq B$, $\|W_K A_1\|_\infty \leq B$ and $\|W_{OV} A_3\|_\infty \leq B$ with $B = \mathcal{O}(\sqrt{\log L})$. Assuming SETH (Hypothesis 1), for every $q > 0$, there are constants $C, C_a, C_b > 0$ such that: there is no $O(n^{2-q})$-time (sub-quadratic) algorithm for the problem AD1TI$(L, d = C \log L, B = C_b \sqrt{\log L}, \delta_F = L^{-C_a})$.

**Remark 4.3.** Proposition 4.1 suggests an efficiency threshold for the upper bound of $\|W_K A_1\|_\infty$, $\|W_Q A_2\|_\infty$, $\|W_{OV} A_3\|_\infty$. Only below this threshold are efficient algorithms for Problem 2 possible.

Moreover, there exist almost-linear DiT inference algorithms following [Alman and Song, 2023].

**Proposition 4.2** (Almost-Linear Time DiT Inference). Assuming SETH, the DiT inference problem AD1TI$(L, d = \mathcal{O}(\log L), B = o(\sqrt{\log L}), \delta_F = 1/\mathrm{poly}(L))$ can be solved in $L^{1+o(1)}$ time.

**Remark 4.4.** Proposition 4.2 is a special case of Proposition 4.1 under the efficiency threshold.

**Remark 4.5.** Propositions 4.1 and 4.2 are dominated by the relation between $L$ and $d$, hence by the subspace dimension $d_0 = dL$. A smaller $d_0$ makes Propositions 4.1 and 4.2 more likely to hold.

## 5 Discussion and Concluding Remarks

We explore the fundamental limits of latent DiTs with 3 key contributions. First, we prove that transformers are universal approximators for the score functions in DiTs (Theorem 3.1), with approximation capacity and model size dependent only on the latent dimension, suggesting DiTs can handle high-dimensional data challenges. Second, we show that Transformer-based score estimators converge to the true score function (Theorem 3.2), ensuring the generated data distribution closely approximates the original (Corollary 3.2.1). Third, we provide provably efficient criteria (Proposition 4.1) and prove the existence of almost-linear time algorithms for forward inference (Proposition 4.2) and backward computation (Theorem 4.1). Our computational results hold for both unconditional and conditional generation of DiTs (Remark 4.1). These results highlight the potential of latent DiTs to achieve both computational efficiency and robust performance in practical scenarios.

**Practical Guidance from Computational Results.** Section 4 analyzes the computational feasibility and identifies all possible efficient DiT algorithms/methods for both forward inference and backward training. These results provide practical guidance for designing efficient methods:

- The latent dimension should be sufficiently small: $d = \mathcal{O}(\log L)$ (Theorem 4.1, Propositions 4.1 and 4.2).
- Normalization of $K, Q$, and $V$ in DiT attention heads enhances performance and efficiency:
  - For efficient inference: $\max\{\|W_K A_1\|, \|W_Q A_2\|, \|W_{OV} A_3\|\} \leq B$ with $B = o(\sqrt{\log L})$ (Proposition 4.2) and $A_1, A_2, A_3$ being the input data associated with $K, Q, V$.
  - For efficient training: $\max\{\|W_K A_1\|, \|W_Q A_2\|, \|W_{OV} A_3\|\} \leq \Gamma$ with $\Gamma = o(\sqrt{\log L})$ (Theorem 4.1).

We remark that these conditions are necessary but not sufficient; sufficient conditions depend on the specific design of the methods used. This is due to the best- or worst-case nature of hardness results.

**Limitations and Future Direction.** As discussed in Remark 3.4, the double exponential factor in our explicit sample complexity bound (Theorem 3.2) suggests a possible gap in our understanding of transformer universality and its interplay with DiT architecture. This motivates us to rethink transformer universality and explore new proof techniques for DiTs, which we leave for future work. Besides, due to its formal nature, this work does not provide immediate practical implementations. However, we expect that our findings provide valuable insights for future diffusion generative models.

**Post-Acceptance Note [October, 29, 2024].** A follow-up work by Hu et al. [2024f] alleviates the double exponential factor and achieves minimax optimal statistical rates for DiTs under Hölder smoothness data assumptions.

## Broader Impact

This theoretical work aims to shed light on the foundations of diffusion generative models and is not anticipated to have negative social impacts.

## Acknowledgments

JH would like to thank to Minshuo Chen, Sophia Pi, Yi-Chen Lee, Yu-Chao Huang, Yibo Wen, Damien Jian, Jialong Li, Zijia Li, Tim Tsz-Kit Lau, Chenwei Xu, Dino Feng and Andrew Chen for enlightening discussions on related topics; Ting-Chun Liu for pointing out typos; and the Red Maple Family for support. The authors would like to thank the anonymous reviewers and program chairs for constructive comments.

JH is partially supported by the Walter P. Murphy Fellowship. HL is partially supported by NIH R01LM1372201, AbbVie and Dolby. The content is solely the responsibility of the authors and does not necessarily represent the official views of the funding agencies.

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

# Appendix

# A   More Discussions on Low-Dimensional Linear Latent Space

Our analysis is based on the low-dimensional linear latent space assumption (Assumption 2.1). Here we further discuss this in light of our theoretical results

Our results are more general and extend beyond Assumption 2.1. In addition to the case where $d_0 < D$, our theoretical results apply to two other settings: $d_0 = D$ and $d_0 > D$. Especially, for $d_0 = D$, our results still hold by setting $B$ as the identity matrix $I_D$. Namely, our results hold after removing the linear subspace assumption.

- Statistically, for score approximation, score estimation, and distribution estimation, the upper bounds depend on the dimension of the latent variable $d_0$, other than $d$. A smaller $d_0$ allows for a reduced model size to achieve a specified approximation error compared to a larger one (Theorem 3.1). Additionally, with a smaller $d_0$, both score and distribution estimation errors are reduced relative to scenarios with larger ones (Theorem 3.2 and Corollary 3.2.1).

- Computationally, smaller $d_0$ benefits the provably efficient criteria (Proposition 4.1, almost-linear time algorithms for forward inference (Proposition 4.2) and backward computation (Theorem 4.1).

## B Notation Table

We summarize our notations in the following table for easy reference.

Table 1: Mathematical Notations and Symbols

| Symbol | Description |
| --- | --- |
| $\|z\|_2$ | Euclidean norm, where $z$ is a vector |
| $\|z\|_\infty$ | Infinite norm, where $z$ is a vector |
| $\|Z\|_2$ | 2-norm, where $Z$ is a matrix |
| $\|Z\|_{\mathrm{op}}$ | Operator norm, where $Z$ is a matrix |
| $\|Z\|_F$ | Frobenius norm, where $Z$ is a matrix |
| $\|Z\|_{p,q}$ | $p,q$-norm, where $Z$ is a matrix |
| $\|f(x)\|_{L^2}$ | $L^2$-norm, where $f$ is a function |
| $\|f(x)\|_{L^2(P)}$ | $L^2(P)$-norm, where $f$ is a function and $P$ is a distribution |
| $\|f(\cdot)\|_{Lip}$ | Lipschitz-norm, where $f$ is a function |
| $f_\sharp P$ | Pushforward measure, where $f$ is a function and $P$ is a distribution |
| $n$ | Sample size |
| $x$ | Data point in original data space, $x \in \mathbb{R}^D$ |
| $h$ | Latent variable in low-dimensional subspace, $h \in \mathbb{R}^{d_0}$ |
| $p_h$ | The destiny function of $h$ |
| $B$ | The matrix with orthonormal columns to transform $h$ to $x$, where $B \in \mathbb{R}^{D \times d_0}$ |
| $\bar{x}$ | Perturbed data variable at $t > 0$ |
| $\bar{h}$ | $\bar{h} = B^\top \bar{x}$ |
| $T$ | Stopping time in the forward process of Diffusion model |
| $T_0$ | Stopping time in the backward process of Diffusion model |
| $\mu$ | Discretized step size in backward process |
| $p_t(\cdot)$ | The density function of $x$ for at time $t$ |
| $p_t^h(\cdot)$ | The density function of $\bar{h}$ at time $t$ |
| $\psi$ | (Conditional) Gaussian density function |
| $d$ | Input dimension of each token in the transformer network of DiT |
| $L$ | Token length in the transformer network of DiT |
| $X$ | Sequence input of transformer network in DiT, where $X \in \mathbb{R}^{d \times L}$ |
| $E$ | Position encoding, where $E \in \mathbb{R}^{d \times L}$ |
| $R(\cdot)$ | Reshape layer in DiT, $R(\cdot) : \mathbb{R}^{d_0} \to \mathbb{R}^{d \times L}$ |
| $W_B$ | The orthonormal matrix to approximate $B$, where $W_B \in \mathbb{R}^{D \times d_0}$ |

## C  Related Works

**Organization.**  In the following, we first discuss recent developments in DiTs. Then, we discuss the main technique of our statistical results: the universality (universal approximation) of transformer. Next, we discuss recent theoretical developments in diffusion generative models. Lastly, we discuss other aspects of transformer in foundation models beyond diffusion models.

**Diffusion Transformers.**  Diffusion [Ho et al., 2020] and score-based generative models [Song and Ermon, 2019] have been particularly successful as generative models of images, video and biomedical data [Nichol et al., 2021, Ramesh et al., 2022, Liu et al., 2024, Zhou et al., 2024a,b, Wang et al., 2024a,b].  Recently, transformer-based diffusion models have garnered significant attention in research. The U-ViT model [Bao et al., 2022] incorporates transformer blocks into a U-net architecture, treating all inputs as tokens. In contrast, DiT [Peebles and Xie, 2023] utilizes a straightforward, non-hierarchical transformer structure. Empirically, diffusion transformers (DiTs) [Peebles and Xie, 2023] have emerged as a significant advancement (e.g., SoRA [OpenAI, 2024, Liu et al., 2024] from OpenAI), effectively combining the strengths of transformer architectures and diffusion-based approaches. Models like MDT [Gao et al., 2023a] and MaskDiT [Zheng et al., 2023] improve the training efficiency of DiT by applying a masking strategy.

**Universality and Memory Capacity of Transformers.**  The universality of transformers refers to their ability to serve as universal approximators. This means that transformers theoretically models any sequence-to-sequence function to a desired degree of accuracy. Yun et al. [2020] establish that transformers can universally approximate sequence-to-sequence functions by stacking numerous layers of feed-forward functions and self-attention functions. In a different approach, Jiang and Li [2023] affirm the universality of transformers by utilizing the Kolmogorov-Albert representation Theorem. Most recently, Kajitsuka and Sato [2023] show that transformers with one self-attention layer is a universal approximator.

The memory capacity of a transformer is a practical measure to test the theoretical results of the transformer's universality, by ensuring the model can handle necessary context and dependencies. By memory capacity, we refer to the minimal set of parameters such that the model (i.e., transformer) approximates all input-output pairs in the training dataset with a bounded error. Several works address the memory capacity of transformers. Kim et al. [2022] show that transformers with $\widetilde{O}(d+L+\sqrt{NL})$ parameters are sufficient to memorize $N$ length-$L$ and dimension-$d$ sequence-to-sequence data points by constructing a contextual mapping with $\mathcal{O}(L)$ attention layers. Mahdavi et al. [2023] show that a multi-head-attention with $h$ heads is able to memorize $\mathcal{O}(hL)$ examples under a linear independence data assumption. Kajitsuka and Sato [2023] show that a single layer transformer with $\mathcal{O}(NLd + d^2)$ parameters is able to memorize $N$ length-$L$ and dimension-$d$ sequence-to-sequence data points by utilizing the connection between the softmax function and Boltzmann operator. Hu et al. [2024d], Wang et al. [2023] extend the results of [Kajitsuka and Sato, 2023, Yun et al., 2020] to prompt tuning and discuss the memorization of the data sequences. Another line of research establishes a different kind of memory capacity for transformers by connecting transformer attention with dense associative memory models (modern Hopfield models) [Hu et al., 2024a,b,e, 2023, Wu et al., 2024a,b, Ramsauer et al., 2020]. Notably, they define memory capacity as the smallest number of (length-$L$ and dimension-$d$) data points the model (transformer attention) is able to store and derive exponential-in-$d$ high-probability capacity lower bounds. In particular, Hu et al. [2024e] report a tight exponential scaling of capacity with feature dimension from the perspective of spherical codes.

Our work is motivated by and builds on [Yun et al., 2020] to bridge the transformer's function approximation ability with data distribution estimation. While we do not address the memorization of DiTs (or diffusion models in general), recent studies on dense associative models suggest viewing pre-trained diffusion generative models as associative memory models [Achilli et al., 2024, Ambrogioni, 2023, Hoover et al., 2023]. We plan to explore this aspect in future work.

**Theories of Diffusion Models.**  In addition to empirical success, there has been several theoretical analysis about diffusion models [Chen et al., 2024b, Tang and Zhao, 2024]. Chen et al. [2023] studies score approximation, estimation, and distribution recovery of U-Net based diffusion models. Benton et al. [2024] provide convergence bounds linear in data dimensions, assuming accurate score function approximation. Zhu et al. [2023], Wibisono et al. [2024] provide statistical sample complexity bounds for score-matching under the similar assumptions. Oko et al. [2023] analyze the distribution

estimation under the assumption that the initial density is supported on $[-1,1]^D$ and smooth in the boundary.

Among these works, our work is built on and closest to [Chen et al., 2023], as both assume the data has a low-dimensional structure[4]. However, our work differs in three key aspects. First, beyond the simple ReLU networks considered in [Chen et al., 2023], we provide the first score approximation analysis for DiTs with a transformer-based score estimator. Second, our work is the first to provide the statistical rates of DiTs (score and distribution estimation) based on transformer universality [Yun et al., 2020] and norm-based converging number bound [Edelman et al., 2022], supporting the practical success of DiTs [Esser et al., 2024, Ma et al., 2024]. Lastly, our work provides the first comprehensive analysis of the computational limits and all possible efficient DiT algorithms/methods for both forward inference and backward training. This offers timely insights into the empirical computational inefficiency of DiTs [Liu et al., 2024] and guidance for future DiT architectures.

**Transformers in Foundation Models: Transformer-Based Pretrained Models.** Transformer-based pretrained models utilize attention mechanisms to process sequential data, enabling the learning of contextual relationships for tasks like natural language understanding and generation. These models encompass three types: encoder-based, decoder-based, and diffusion transformers. Encoder-based transformers, such as DNABERT [Zhou et al., 2024c, 2023, Ji et al., 2021], employ bidirectional attention to extract feature representations DNABERT shows great potential to capture complex patterns of genome sequences and improve tasks such as gene prediction. Decoder-based transformers generate output sequences from encoded information using unidirectional attention, such as ChatGPT [Radford et al., 2019, Floridi and Chiriatti, 2020, Brown et al., 2020] for natural language. The diffusion transformers generate a sequence toward a target distribution, such as SoRA [Liu et al., 2024] and Videofusion [Luo et al., 2023] for video generation and DecompDiff [Guan et al., 2024] for drug design. In our paper, we present an early exploration of the statistical and computational limits of diffusion transformer models.

---

[4]Recent work by Havrilla and Liao [2024] examines the generalization and approximation of transformers under Hölder smoothness and low-dimensional subspace assumptions.

# D    Supplementary Theoretical Background

In this section, we provide some further background. We show the details about the forward and backward process in Diffusion Models in Appendix D.1. Besides, we give the details of the proof about the score decomposition in Appendix D.2.

## D.1    Diffusion Models

**Forward Process.**    Diffusion models gradually add noise to the original data in the forward process. We describe the forward process as the following SDE

$$\mathrm{d}x_t = -\frac{1}{2}w(t)x_t\mathrm{d}t + \sqrt{w(t)}\mathrm{d}W_t, \ x_t \in \mathbb{R}^D, \tag{D.1}$$

where $x_0 \sim P_0$, $(W_t)_{t\geq 0}$ is a standard Brownian motion, and $w(t) > 0$ is a nondecreasing weighting function. Let $P_t$ and $p_t$ denote the marginal distribution and destiny of $x_t$. The conditional distribution $P(x_t|x_0)$ follows $N(\beta(t)x_0, \sigma(t)I_D)$, where $\beta(t) = \exp\left(-\int_0^t w(s)\mathrm{d}s/2\right)$ and $\sigma(t) = 1 - \beta^2(t)$. In practice, (D.1) terminates at a large enough $T$ such that $P_T$ is close to $N(0, I_D)$.

**Backward Process.**    We obtain the backward process $y_t := x_{T-t}$ by reversing (D.1). The backward process satisfies

$$\mathrm{d}y_t = \left[\frac{1}{2}w(T-t)y_t + w(T-t)\nabla\log p_{T-t}(y_t)\right]\mathrm{d}t + \sqrt{w(T-t)}\mathrm{d}\overline{W}_t, \tag{D.2}$$

where the score function $\nabla\log p_t(\cdot)$ is the gradient of log probability density function of $x_t$, and $\overline{W}_t$ is a reversed Brownian motion. However, $\nabla\log p_t(\cdot)$ and $P_T$ are both unknown in (D.2). To resolve this, we use a score estimator $s_W(\cdot, t)$ to replace $\nabla\log p_t(\cdot)$, where $s_W(\cdot, t)$ is usually a neural network with parameters $W$. Secondly, we replace $P_T$ by the standard Gaussian distribution. Consequently, we obtain the following SDE

$$\mathrm{d}\widetilde{y}_t = \left[\frac{1}{2}w(T-t)\widetilde{y}_t + w(T-t)s_W(\widetilde{y}_t, T-t)\right]\mathrm{d}t + \sqrt{w(T-t)}\mathrm{d}\overline{W}_t, \ \widetilde{y}_0 \sim N(0, I_D). \tag{D.3}$$

In practice, we use discrete schemes of (D.3) to generate data, following [Song and Ermon, 2019]. We use $\mu > 0$ to denote the discretization step size. For $t \in [k\mu, (k+1)\mu]$, we have

$$\mathrm{d}\widetilde{y}_t^\mu = \left[\frac{1}{2}w(T-t)\widetilde{y}_{k\mu}^\mu + w(T-t)s_W(\widetilde{y}_{k\mu}^\mu, T-k\mu)\right]\mathrm{d}t + \sqrt{w(T-t)}\mathrm{d}\overline{W}_t. \tag{D.4}$$

## D.2    Proof of Lemma 2.1

Here we restate the proof of [Chen et al., 2023, Lemma 1] for completeness.

*Proof.* Recall $x = Bh$ by Assumption 2.1 with $x \in \mathbb{R}^D$, $B \in \mathbb{R}^{D \times d_0}$ and $h \in \mathbb{R}^{d_0}$.

By the forward process (D.1), we have

$$p_t(\bar{x}) = \int \psi_t(\bar{x} \mid Bh)p_h(h)\mathrm{d}h, \tag{D.5}$$

where

$$\psi_t(\bar{x} \mid Bh) = [2\pi h(t)]^{-D/2}\exp\left(-\frac{\|\beta(t)Bh - \bar{x}\|_2^2}{2\sigma(t)}\right), \tag{D.6}$$

is the Gaussian transition kernel.

Then we write the score function as

$$\nabla \log p_t(\bar{x}) = \frac{\nabla p_t(\bar{x})}{p_t(\bar{x})} \tag{D.7}$$

$$= \frac{\nabla \int \psi_t(\bar{x} \mid Bh) p_h(h) \mathrm{d}h}{\int \psi_t(\bar{x} \mid Bh) p_h(h) \mathrm{d}h} \qquad \left(\text{By pluging in } p_t(\bar{x})\right)$$

$$= \frac{\int \nabla \psi_t(\bar{x} \mid Bh) p_h(h) \mathrm{d}h}{\int \psi_t(\bar{x} \mid Bh) p_h(h) \mathrm{d}h}, \qquad \left(\text{By interchanging } \int \text{ with } \nabla\right)$$

where the last equality holds since $\psi_t(\bar{x} \mid Bh)$ is continuously differentiable in $\bar{x}$.

Plugging (D.6) into (D.7), we have

$$\nabla \log p_t(\bar{x})$$
$$= \frac{[2\pi h(t)]^{-D/2}}{\int \psi_t(\bar{x} \mid Bh) p_h(h) \mathrm{d}h} \int \frac{1}{\sigma(t)} \left(\beta(t) Bh - \bar{x}\right) \exp\left(-\frac{\|\beta(t) Bh - \bar{x}\|_2^2}{2\sigma(t)}\right) p_h(h) \mathrm{d}h.$$

We then decompose above score function by projecting of $\bar{x}$ into $\mathrm{Span}(B)$, i.e., replacing $-\bar{x}$ with $-BB^\top \bar{x} - (I_D - BB^\top) \bar{x}$:

$$\nabla \log p_t(\bar{x})$$
$$= \frac{[2\pi h(t)]^{-D/2}}{\int \psi_t(\bar{x} \mid Bh) p_h(h) \mathrm{d}h}$$
$$\cdot \int \frac{1}{\sigma(t)} \left[\left(\beta(t) Bh - BB^\top \bar{x}\right) - \left(I_D - BB^\top\right) \bar{x}\right] \exp\left(-\frac{\|\beta(t) Bh - \bar{x}\|_2^2}{2\sigma(t)}\right) p_h(h) \mathrm{d}h.$$

Absorbing the factor of $[2\pi h(t)]^{-D/2}$ into the Gaussian kernel $\psi_t(\bar{x} \mid Bh)$, we have

$$\nabla \log p_t(\bar{x})$$
$$= \frac{[2\pi h(t)]^{-D/2}}{\int \psi_t(\bar{x} \mid Bh) p_h(h) \mathrm{d}h} \int \frac{1}{\sigma(t)} \left(\beta(t) Bh - BB^\top \bar{x}\right) \exp\left(-\frac{\|\beta(t) Bh - \bar{x}\|_2^2}{2\sigma(t)}\right) p_h(h) \mathrm{d}h$$
$$- \frac{1}{\int \psi_t(\bar{x} \mid Bh) p_h(h) \mathrm{d}h} \left(\frac{1}{\sigma(t)} \left(I_D - BB^\top\right) \bar{x}\right) \int \psi_t(\bar{x} \mid Bh) p_h(h) \mathrm{d}h$$
$$= \underbrace{\frac{1}{\int \psi_t(\bar{x} \mid Bh) p_h(h) \mathrm{d}h} \int \frac{1}{\sigma(t)} \left(\beta(t) Bh - BB^\top \bar{x}\right) \psi_t(\bar{x} \mid Bh) p_h(h) \mathrm{d}h}_{:=s_+} \underbrace{- \frac{1}{\sigma(t)} \left(I_D - BB^\top\right) \bar{x}}_{:=s_-}.$$

To further simplify $s_+$, we decompose $\psi_t(\bar{x} \mid Bh)$ as

$$\psi_t(\bar{x} \mid Bh)$$
$$= [2\pi h(t)]^{-D/2} \exp\left(-\frac{1}{2\sigma(t)} \|\beta(t) Bh - \bar{x}\|_2^2\right)$$
$$= [2\pi h(t)]^{-D/2} \exp\left(-\frac{1}{2\sigma(t)} \left\|\beta(t) Bh - BB^\top \bar{x} - \left(I_D - BB^\top\right) \bar{x}\right\|_2^2\right)$$
$$= [2\pi h(t)]^{-D/2} \exp\left(-\frac{1}{2\sigma(t)} \left(\left\|\beta(t) Bh - BB^\top \bar{x}\right\|_2^2 + \left\|\left(I_D - BB^\top\right) \bar{x}\right\|_2^2\right.\right.$$
$$\left.\left. - 2(B(\beta(t)h - B^\top \bar{x}))^\top (I_D - BB^\top) \bar{x}\right)\right)$$

$$= [2\pi h(t)]^{-D/2} \exp\left(-\frac{1}{2\sigma(t)}\left(\left\|\beta(t)Bh - BB^\top \bar{x}\right\|_2^2 + \left\|\left(I_D - BB^\top\right)\bar{x}\right\|_2^2\right)\right)$$

$$\left(B(\beta(t)h - B^\top\bar{x}) \text{ is in Span}(B) \text{ while } (I_D - BB^\top)\bar{x} \text{ is orthogonal to Span}(B)\right)$$

$$= \underbrace{[2\pi h(t)]^{-d_0/2} \exp\left(-\frac{\left\|\beta(t)h - B^\top\bar{x}\right\|_2^2}{2\sigma(t)}\right)}_{:=\psi_t(B^\top\bar{x}|h)} \cdot \underbrace{[2\pi h(t)]^{-(D-d_0)/2} \exp\left(-\frac{\left\|\left(I_D - BB^\top\right)\bar{x}\right\|_2^2}{2\sigma(t)}\right)}_{:=\psi_t((I_D - BB^\top)\bar{x})},$$

$$\left(\text{since } B \text{ has orthonormal columns}\right)$$

where both $\psi_t\left(B^\top\bar{x} \mid h\right)$ and $\psi_t\left((I_D - BB^\top)\bar{x}\right)$ are Gaussian.

Plugging $\psi_t(\bar{x} \mid Bh) = \psi_t\left(B^\top\bar{x} \mid h\right)\psi_t\left((I_D - BB^\top)\bar{x}\right)$ into $s_+$, we obtain

$$s_+(\bar{x}, t) = C \int \frac{1}{\sigma(t)}\left(\beta(t)Bh - BB^\top\bar{x}\right)\psi_t(B^\top\bar{x} \mid h)\psi_t((I_D - BB^\top)\bar{x})p_h(h)\mathrm{d}h$$

$$= C\psi_t((I_D - BB^\top)\bar{x}) \int \frac{1}{\sigma(t)}\left(\beta(t)Bh - BB^\top\bar{x}\right)\psi_t(B^\top\bar{x} \mid h)p_h(h)\mathrm{d}h$$

$$= \frac{1}{\int \psi_t(B^\top\bar{x} \mid h)p_h(h)\mathrm{d}h} \int \frac{1}{\sigma(t)}\left(\beta(t)Bh - BB^\top\bar{x}\right)\psi_t(B^\top\bar{x} \mid h)p_h(h)\mathrm{d}h,$$

where $C := [\psi_t((I_D - BB^\top)\bar{x}) \int \psi_t(B^\top\bar{x} \mid h)p_h(h)\mathrm{d}h]^{-1}$.

Notably, $s_+$ depends only on the projected data $B^\top\bar{x}$. Therefore, we are able to replace $s_+(\bar{x}, t)$ with $s_+(B^\top\bar{x}, t)$. The benefit is that the dimension $d_0$ of the first input in $s_+(B^\top\bar{x}, t)$ is much smaller.

Lastly, by denoting $\bar{h} = B^\top\bar{x}$ such that $\nabla_{\bar{h}}\psi_t(\bar{h} \mid h) = (\beta(t)h - \bar{h})\psi_t(\bar{h} \mid h)/\sigma(t)$, we arrive at

$$s_+(B^\top\bar{x}, t) = B \int \frac{\nabla_{\bar{h}}\psi_t(\bar{h} \mid h)p_h(h)}{\int \psi_t(\bar{h} \mid h)p_h(h)\mathrm{d}h}\mathrm{d}h$$

$$= B\nabla \log p_t^h(B^\top x). \qquad \left(p_t^h(\bar{h}) := \int \psi_t(\bar{h}|h)p_h(h)\mathrm{d}h\right)$$

This completes the proof. □

### D.3 Preliminaries: Strong Exponential Time Hypothesis (SETH) and Tensor Trick

Here we present the ideas we built upon for Section 4.

**Strong Exponential Time Hypothesis (SETH).** Impagliazzo and Paturi [2001] introduce the Strong Exponential Time Hypothesis (SETH) as a stronger form of the P ≠ NP conjecture. It suggests that our current best SAT algorithms are optimal and is a popular conjecture for proving fine-grained lower bounds for a wide variety of algorithmic problems [Cygan et al., 2016, Williams, 2018].

**Hypothesis 1** (SETH). For every $\epsilon > 0$, there is a positive integer $k \geq 3$ such that $k$-SAT on formulas with $n$ variables cannot be solved in $\mathcal{O}(2^{(1-\epsilon)n})$ time, even by a randomized algorithm.

**Tensor Trick for Computing Gradients.** The tensor trick [Diao et al., 2019, 2018] is an instrument to compute complicated gradients in a clean and tractable fashion. We start with some definitions.

**Definition D.1** (Vectorization). For any matrix $X \in \mathbb{R}^{L \times d}$, we define $\underline{X} := \mathrm{vec}(X) \in \mathbb{R}^{Ld}$ such that $X_{i,j} = \underline{X}_{(i-1)d+j}$ for all $i \in [L]$ and $j \in [d]$.

**Definition D.2** (Matrixization). For any vector $\underline{X} \in \mathbb{R}^{Ld}$, we define $\mathrm{mat}(\underline{X}) = X$ such that $X_{i,j} = \mathrm{mat}(\underline{X}) := \underline{X}_{(i-1)d+j}$ for all $i \in [L]$ and $j \in [d]$, namely $\mathrm{mat}(\cdot) = \mathrm{vec}^{-1}(\cdot)$.

**Definition D.3** (Kronecker Product). Let $A \in \mathbb{R}^{L_a \times d_a}$ and $B \in \mathbb{R}^{L_b \times d_b}$. We define the Kronecker product of $A$ and $B$ as $A \otimes B \in \mathbb{R}^{L_a L_b \times d_a d_b}$ such that $(A \otimes B)_{(i_a-1)L_b+i_b,(j_a-1)d_b+j_b}$, is equal to $A_{i_a,j_a}B_{i_b,j_b}$ with $i_a \in [L_a], j_a \in [d_a], i_b \in [L_b], j_b \in [d_b]$.

**Definition D.4** (Sub-Block of a Tensor)**.** For any $A \in \mathbb{R}^{L_a \times d_a}$ and $B \in \mathbb{R}^{L_b \times d_b}$, let $\mathsf{A} := A \otimes B \in \mathbb{R}^{L_a L_b \times d_a d_b}$. For any $\underline{j} \in [L_a]$, we define $\mathsf{A}_{\underline{j}} \in \mathbb{R}^{L_b \times d_a d_b}$ be the $\underline{j}$-th $L_b \times d_a d_b$ sub-block of $\mathsf{A}$.

**Lemma D.1** (Tensor Trick [Diao et al., 2019, 2018])**.** For any $A \in \mathbb{R}^{L_a \times d_a}$, $B \in \mathbb{R}^{L_b \times d_b}$ and $X \in \mathbb{R}^{d_a \times d_b}$, it holds $\mathrm{vec}\left(A^\top X B\right) = (A^\top \otimes B^\top)\underline{X} \in \mathbb{R}^{L_a L_b}$.

To showcase the tensor trick, let's consider a (single data point) attention following [Gao et al., 2023b,c]. Setting $D := \mathrm{diag}\left(\exp\left(X^\top W_K^\top W_Q X\right)\mathbb{1}_L\right)$ and $W := W_K W_Q^\top \in \mathbb{R}^{d \times d}$, we have

$$\mathcal{L}_0 := \big\| \underbrace{W_V}_{d \times d} \underbrace{X}_{\in \mathbb{R}^{d \times L}} \underbrace{D^{-1}}_{\in \mathbb{R}^{L \times L}} \underbrace{\exp\{X^\top W X\}}_{\in \mathbb{R}^{L \times L}} - \underbrace{Y}_{\in \mathbb{R}^{d \times L}} \big\|_2^2. \tag{D.8}$$

**Proposition D.1** (Definition 4.7 of [Gao et al., 2023b])**.** By Definition D.3 and Definition D.4, we identify $D_{\underline{j},\underline{j}} := \left\langle \exp\left(\mathsf{A}_{\underline{j}}\,\underline{W}\right), \mathbb{1}_L \right\rangle \in \mathbb{R}$ for all $\underline{j} \in [L]$, with $\mathsf{A} := X \otimes X \in \mathbb{R}^{L^2 \times d^2}$ and $\underline{W} \in \mathbb{R}^{d^2}$. Therefore, for each $\underline{j} \in [L]$ and $\underline{i} \in [d]$, it holds $\mathcal{L}_0 = \sum_{\underline{j}=1}^L \sum_{\underline{i}=1}^d \frac{1}{2}\left(\left\langle D_{\underline{j},\underline{j}}^{-1}\exp\left(\mathsf{A}_{\underline{j}}\,\underline{W}\right), X W_V[\cdot,\underline{i}]\right\rangle - Y_{\underline{j},\underline{i}}\right)^2$.

The elegance of Proposition D.1 emerges when we vectorize the weights into vectors $\underline{W}, \underline{W}_V$, making the gradient computations (e.g., $\mathrm{d}\mathcal{L}_0/\underline{W}$ and $\mathrm{d}\mathcal{L}_0/\underline{W}_V$) more tractable by avoiding complex matrix or tensor derivatives. This approach systematically simplifies the handling of chain-rule terms in the gradient computation of losses like $\mathcal{L}_0$.

**Fine-Grained Complexity for Transformer.** Many recent works also utilize similar techniques from fine-grained complexity to analyze transformer architectures. Alman and Song [2023, 2024b], Liang et al. [2024d], Alman and Song [2024a] explore the computational feasibility of inference and training for standard softmax and tensor attention. Liang et al. [2024c] extend the single-layer training results from [Alman and Song, 2024b] to deep transformer models. [Liang et al., 2024a] extend [Alman and Song, 2024b] to provide a fast attention gradient approximation based on Fourier transform. [Liang et al., 2024b] extend [Alman and Song, 2024b] to sparse attention matrix. Hu et al. [2024d] study the computational limits of inference and training in prompt-tuning for pretrained transformers. Hu et al. [2024c] study the computational limits of LoRA [Hu et al., 2021] in transformers, identifying norm-bound conditions for efficient LoRA training and proving the existence of nearly linear-time LoRA algorithms.

Our work is closest to [Alman and Song, 2024b, 2023]. Our forward inference computational results build on [Alman and Song, 2023]. Our backward training computational results are related to [Alman and Song, 2024b] but include additional analysis on reshaping and latent embedding.

# E    More Background and Auxiliary Lemmas: Universal Approximation of Transformers via Piecewise Approximation

Here, we review the universal approximation of transformers following [Yun et al., 2020].

Our goal is to reproduce the results of [Yun et al., 2020] and use or modify them as auxiliary lemmas for proofs of Section 3 (i.e., Appendix F.)

We start with their central result and prove it in the rest of the section.

**Lemma E.1** (Universal Approximation of Transformers, Theorem 3 of [Yun et al., 2020]). Let $\epsilon > 0$. For any given compact-supported continuous function $f : \mathbb{R}^{d \times L} \to \mathbb{R}^{d \times L}$, there exists a transformer network $f_{\mathcal{T}} \in \mathcal{T}_p^{2,1,4}$, such that

$$\left( \int \| f_{\mathcal{T}}(X) - f(X) \|_F^2 \, dX \right)^{1/2} \leq \epsilon.$$

**Proof Overview.**    We use the following proof strategy:

- **Step 1.** We show that the piecewise-constant function is able to approximate compact-supported continuous function in Appendix E.1.

- **Step 2.** We define modified self-attention and feed-forward layers to construct the modified transformer. We show that the modified transformer is able to approximate piecewise-constant function in Appendix E.2.

- **Step 3.** We show that the standard transformer in Appendix E.3 is able to approximate the modified transformer.

We provide details of **Step 1.** in Appendix E.1, **Step 2.** in Appendix E.2, and **Step 3.** in Appendix E.3. Then we summarize our results in Appendix E.4.

## E.1    Piecewise-Constant Function Approximates Compact-Supported Continuous Function

In this subsection, we show that the piecewise-constant function is able to approximate compact-supported continuous function.

We start with the definition of the compact-supported continuous functions of interest.

**Assumption E.1.** Without loss of generality, we assume that the target function in discussion is supported on $[0, 1]^{d \times L}$. We denote the set of $[0, 1]^{d \times L}$-supported continuous functions as $\mathcal{F}$.

We introduce the notion of grid and cube for the compact support $[0, 1]^{d \times L}$.

**Definition E.1** (Grid and Cube with Width $\delta$). Given a grid width $\delta$, let $\mathcal{G}_{\delta} := \{0, \delta, \ldots, 1 - \delta\}^{d \times L}$ denote the set of grids within $[0, 1]^{d \times L}$. For a grid point $G = (G_{j \in [d], k \in [L]}) \in \mathcal{G}_{\delta}$, we denote its associated cube as

$$\mathcal{S}_G := \otimes_{j=1}^{d} \otimes_{k=1}^{L} [G_{j,k}, G_{j,k} + \delta] \subset [0, 1]^{d \times L}.$$

Each cube $\mathcal{S}_G$ represents a hyper rectangular in the multi-dimensional space $[0, 1]^{d \times L}$, constructed to discretize the space into smaller subspaces.

We introduce the notion of piecewise-constant fucntion class w.r.t. the $[0, 1]^{d \times L}$-supported continuous function class $\mathcal{F}$.

**Definition E.2** (Piecewise-Constant Function Class). Let $f_{\delta}$ denote the piesewise constant function of grid width $\delta$, and $\mathbb{1}\{\cdot\}$ denote the indicator function. For each $G \in \mathcal{G}_{\delta}$, and any matrix $A_G \in \mathbb{R}^{d \times L}$, we define the piecewise-constant function class as

$$\mathcal{F}(\delta) := \left\{ f_{\delta} : X \to \sum_{G \in \mathcal{G}_{\delta}} A_G \cdot \mathbb{1}\{X \in \mathcal{S}_G\}, A_G \in \mathbb{R}^{d \times L} \right\}. \tag{E.1}$$

We recall that for a given sequence-to-sequence function $f$,

$$\|f\|_{L^2} := \left( \int \|f(X)\|_F^2 \mathrm{d}X \right)^{1/2}.$$

We approximate the compact-supported function with a piecewise-constant function in the next lemma.

**Lemma E.2.** (Lemma 8 of [Yun et al., 2020]) For any given $f \in \mathcal{F}$ and $\epsilon/3 > 0$, we can find a $\delta^\star > 0$, such that there exists a $f_{\delta^\star} \in \mathcal{F}(\delta^\star)$ satisfying $\|f - f_{\delta^\star}\|_{L^2} \le \epsilon/3$.

*Proof.* See Appendix E.5.2 for a detailed proof. □

### E.2 Modified Transformer Approximates Piecewise-constant Function

In this subsection, we define modified self-attention and feed-forward layers to construct the modified transformers. We use the modified transformers to approximate the piecewise-constant function.

**Definition E.3** (Modified Transformer Networks). The modified transformer network $\overline{\mathcal{T}}_p^{r,m,l}$ includes two modifications to the standard transformer network $\mathcal{T}_p^{r,m,l}$:

- Modified attention layer: Replace $\mathrm{Softmax}$ operator with $\mathrm{Hardmax}$ operator $\sigma_H(\cdot)$.
- Modified feed-forward layer: Replace $\mathrm{ReLU}(\cdot)$ with an activation function $\zeta \in \Psi$. Here, $\Psi$ denotes the set of all piecewise linear functions with at most three pieces and at least one constant.

We approximate $\mathcal{F}(\delta)$ with this modified transformer networks $\overline{\mathcal{T}}_p^{r,m,l}$.

**Lemma E.3** (Modified from Proposition 4 of [Yun et al., 2020]). For each $f_\delta \in \mathcal{F}(\delta)$, there exists a $f_{\mathcal{T},c} \in \overline{\mathcal{T}}_p^{2,1,1}$ such that $\|f_\delta - f_{\mathcal{T},c}\|_{L^2} = \mathcal{O}(\delta^{d/2})$.

*Proof Sketch.* Given $\delta$, and for any grid $G \in \mathcal{G}_\delta$, we have a grid set $\mathcal{G}_\delta$ and the cube $\mathcal{S}_G$.

Our proof follows two steps:

- **Quantization.** For all $X \in \mathbb{R}^{d \times L}$, we quantize it to a finite set:

    - If $X \in \mathcal{S}_G \subset [0,1]^{d \times L}$, we quantize it to the element $G \in \mathcal{G}_\delta$.
    - If $X \notin [0,1]^{d \times L}$, we quantize it to an element out of $\mathcal{G}_\delta$.

- **Mapping.** For any $G \in \mathcal{G}_\delta$, we map it to the desired output $A_G$.

For **Quantization**, we achieve this by a series of modified feed-forward layers. We show this in Appendix E.2.1.

For **Mapping**, we follow two steps:

- For any $G \neq G' \in \mathcal{G}_\delta$, we use a "contextual mapping" $q_c(\cdot)$ (defined as Definition E.4). The mapping maps all the elements in $q_c(G)$ and $q_c(G')$ to different values. Then, we use a series of modified self-attention layers to achieve "contextual mapping". We show this in Appendix E.2.2.

**Definition E.4** (Contextual Mapping). Consider a finite set $\mathcal{G}_\delta \in \mathbb{R}^{d \times L}$. A map $q_c : \mathcal{G}_\delta \to \mathbb{R}^{1 \times L}$ defines a contextual mapping if the map satisfies the following:

- For any $G \in \mathcal{G}_\delta$, the entries in $q_c(G)$ are all distinct.
- For any $G \neq G' \in \mathcal{G}_\delta$, all entries of $q_c(G)$ and $q_c(G')$ are distinct.

- For any $G \in \mathcal{G}_\delta$, we use a series of modified feed-forward layers to map $q_c(G)$ to $A_G$. We show this in Appendix E.2.3.

□

**Remark E.1.** Our proof differs from [Yun et al., 2020] in one aspect: Although [Yun et al., 2020, Proposition 4] outlines a proof for transformer networks without positional encoding and sketches the proof for networks with it, we provide a detailed proof for the latter to support our proof.

### E.2.1 Quantization by Modified Feed-forward Layers

We use a series of modified feed-forward layers in $\overline{\mathcal{T}}_p^{r,m,l}$ to quantize an input $X \in \mathbb{R}^{d \times L}$ to an element $G$ of the following grid:

$$\{-J, 0, \delta, \ldots, 1 - \delta\}^{d \times L},$$

where $J > L > 0$ is a large number to be determined later. We achieve this via two steps.

- **Step 1: Map the element out of $[0, 1)$ to $-J$.**
  We use $e_i$ to represent the standard unit vector where the $i$-th element is 1. For the $i$-th row of $X$, we define the following feed-forward layer to achieve our aim.

**Definition E.5** (Feed-forward Layer 1)**.** The vector $e_i$ acts as the weight parameters, and $\zeta_1(\cdot)$ acts as the activation function in the feed-forward layer

$$X \to X + e_i \zeta_1(e_i^\top X), \quad \zeta_1(t) = \begin{cases} -t - J, & \text{for } t < 0 \text{ or } t \geq 1, \\ 0, & \text{otherwise.} \end{cases} \tag{E.2}$$

We take $i = 1$ as an example to give the specific calculation. Let $X = (x_{i,j})_{d \times L}$, then we have

$$\mathrm{FF}(X) = X + \begin{pmatrix} 1 \\ 0 \\ \vdots \\ 0 \end{pmatrix} \begin{pmatrix} \zeta_1(x_{1,1}) & \zeta_1(x_{1,2}) & \cdots & \zeta_1(x_{1,L}) \end{pmatrix}$$

$$= X + \begin{pmatrix} \zeta_1(x_{1,1}) & \zeta_1(x_{1,2}) & \cdots & \zeta_1(x_{1,L}) \\ 0 & 0 & \cdots & 0 \\ \vdots & \vdots & \vdots & \vdots \\ 0 & 0 & \cdots & 0 \end{pmatrix}.$$

In the first row of $X$, the above layer transforms the element that is out of $[0, 1)$ to $-J$.
We stack the above layers together for $i = 1, 2, \ldots, d$. If the element of $X$ is out of $[0, 1)$, the series of layers maps it to $J$.

- **Step 2: Map the element in $[0, 1)$ to $\{0, \delta, 2\delta, \ldots, 1 - \delta\}$.**
  For the $i$-th row of $X$, we take $k = 0, 1, \ldots, 1/\delta - 1$ respectively. We define the following layer.

**Definition E.6** (Feed-forward Layer 2)**.** The vector $e_i$ acts as the weight parameters and $\zeta_2(\cdot)$ acts as the activation function in the feed-forward layer

$$X \to X + e_i \zeta_2(e_i^\top X - k\delta \mathbb{1}_n^\top), \quad \zeta_2(t) = \begin{cases} 0, & t < 0 \text{ or } t \geq \delta, \\ -t, & 0 \leq t < \delta. \end{cases} \tag{E.3}$$

We take $i = 1$ and $k = 1$ as an example. We give the following specific calculation

$$\mathrm{FF}(X) = X + \begin{pmatrix} 1 \\ 0 \\ \vdots \\ 0 \end{pmatrix} \begin{pmatrix} \zeta_2(x_{1,1} - \delta) & \zeta_2(x_{1,2} - \delta) & \cdots & \zeta_2(x_{1,L} - \delta) \end{pmatrix}$$

$$= X + \begin{pmatrix} \zeta_2(x_{1,1} - \delta) & \zeta_2(x_{1,2} - \delta) & \cdots & \zeta_2(x_{1,L} - \delta) \\ 0 & 0 & \cdots & 0 \\ \vdots & \vdots & \vdots & \vdots \\ 0 & 0 & \cdots & 0 \end{pmatrix}.$$

In the first row of $X$, the above layer transforms the element in $[\delta, 2\delta]$ to $\delta$.

We stack the above layers together for $i = 1, 2, \ldots, d$ and $k = 0, 1, \ldots, 1/\delta - 1$. If the element of $X$ is in $[k\delta, (k+1)\delta]$, the series layers maps it to $k\delta$.

Combining the above two parts, we achieve our goal with $d/\delta + d$ feed-forward layers. We denote the $d/\delta + d$ series layers as $f_{\mathcal{T},c1}$.

### E.2.2 Contextual Mapping by Modified Self-attention Layers

In our attention layers, we use the following positional encoding $E \in \mathbb{R}^{d \times L}$

$$
E = \begin{pmatrix}
0 & 1 & 2 & \cdots & L-1 \\
0 & 1 & 2 & \cdots & L-1 \\
\vdots & \vdots & \vdots & & \vdots \\
0 & 1 & 2 & \cdots & L-1
\end{pmatrix}.
\tag{E.4}
$$

According to Appendix E.2.1, the output of $f_{\mathcal{T},c1}$ is in the grid $\{-J, 0, \delta, \ldots, 1 - \delta\}^{d \times L}$. For any $X$ in this grid, the first column of $X + E$ is in

$$
\{-J, 0, \delta, \ldots, 1 - \delta\}^d,
$$

and the second column is in

$$
\{-J + 1, 1, 1 + \delta, \ldots, 2 - \delta\}^d.
$$

The results are similar in the other columns.

For $i = 0, 1, \ldots, L - 1$, we use the following notation:

$$
[i : \delta : i + 1 - \delta]_J := \{i - J, i, i + \delta, \ldots, i + 1 - \delta\}.
$$

Then, we define the grid $\mathcal{G}_\delta^+$ as the following.

**Definition E.7** (Grid $\mathcal{G}_\delta^+$). We add $E$ to all the grid points in $\mathcal{G}_\delta$ to generate the modified grid $\mathcal{G}_\delta^+$, defined as follows:

$$
\mathcal{G}_\delta^+ := [0 : \delta : 1 - \delta]_J^d \times [1 : \delta : 2 - \delta]_J^d \times \cdots \times [L - 1 : \delta : L - \delta]_J^d.
$$

Next, we show that the modified attention layer computes contextual mapping (Definition E.4) for $\mathcal{G}_\delta^+$. For $i = 1, 2, \ldots, L - 1$, we use the following notation:

$$
[i : \delta : i + 1 - \delta] := \{i, i + \delta, i + 2\delta, \ldots, i + 1 - \delta\}.
$$

**Lemma E.4** (Modified from Lemma 6 of [Yun et al., 2020]). We consider the following subset of $\mathcal{G}_\delta^+$:

$$
\widetilde{\mathcal{G}}_\delta := \underbrace{[0 : \delta : 1 - \delta]^d \times [1 : \delta : 2 - \delta]^d \times \cdots \times [L - 1 : \delta : L - \delta]^d}_{L}.
$$

Assume that $L \geq 2$ and $\delta^{-1} \geq 2$. Then, there exist a function $f_{\mathcal{T},c2} : \mathbb{R}^{d \times L} \to \mathbb{R}^{d \times L}$ composed of $\delta^{-d} + 1$ modified attention layers (Definition E.3), a vector $u \in \mathbb{R}^d$, and two constants $t_l, t_r \in \mathbb{R}$ $(0 < t_l < t_r)$, such that $q_c(G) := u^\top f_{\mathcal{T},c2}(G), G \in \mathcal{G}_\delta^+$ satisfies the following properties:

1. For any $G \in \widetilde{\mathcal{G}}_\delta$, all the entries of $q_c(G)$ are distinct.

2. For any different $G, G' \in \widetilde{\mathcal{G}}_\delta$, all the entries of $q_c(G), q_c(G')$ are distinct.

3. For any $G \in \widetilde{\mathcal{G}}_\delta$, all the entries of $q_c(G)$ are in $[t_l, t_r]$.

4. For any $G \in \mathcal{G}_\delta^+ \setminus \widetilde{\mathcal{G}}_\delta$, all the entries of $q_c(G)$ are outside $[t_l, t_r]$.

*Proof.* See Appendix E.5.3 for a detailed proof. ☐

**Remark E.2.** Our proof differs from [Yun et al., 2020] in one aspect: The original [Yun et al., 2020, Lemma 6] does not include positional encoding (E.4). Although Yun et al. [2020] sketches the proof for networks with (E.4) in the attention layer input, we detail the proof.

### E.2.3 Map to the Desired Output by Modified Feed-forward Layers

Next, we show that a series of feed-forward layers map the output of modified attention layers $f_{\mathcal{T},c2}$ to the desired output of function $f_{\delta^\star}$.

**Lemma E.5** (Lemma 7 of [Yun et al., 2020]). There exists a function $f_{\mathcal{T},c3} : \mathbb{R}^{d \times L} \to \mathbb{R}^{d \times L}$ composed of $\mathcal{O}(L(1/\delta)^{dL}/L!)$ modified feed-forward layers, such that

$$
f_{\mathcal{T},c3} \circ f_{\mathcal{T},c2}(G) = \begin{cases} A_G & \text{if } G \in \widetilde{\mathcal{G}}_\delta, \\ \mathbf{0}_{d \times L} & \text{if } G \in \mathcal{G}_\delta^+ \setminus \widetilde{\mathcal{G}}_\delta. \end{cases}
$$

*Proof.* See Appendix E.5.4 for a detailed proof. ☐

In conclusion, we have the following lemma for the required number of layers in the modified transformer.

**Lemma E.6** (Total Number of Layers). From the proof of Lemma E.3, if we want to achieve a approximation error $\mathcal{O}(\delta^{d/2})$ by the modified transformer, we need $\mathcal{O}(\delta^{-1})$ modified feed-forward layers in $f_{\mathcal{T},c1}$, $\mathcal{O}(\delta^{-d})$ modified self-attention layers in $f_{\mathcal{T},c2}$, and $\mathcal{O}(\delta^{-dL})$ modified feed-forward layers in $f_{\mathcal{T},c3}$.

*Proof.* By the proof of Lemma E.3, we complete the proof. ☐

### E.3 Standard Transformers Approximate Modified Transformers

In this subsection, we show that standard neural network layers are able to approximate the modified self-attention layers and the modified feed-forward layers (Definition E.3). We have the following Lemma E.7.

**Lemma E.7** (Lemma 9 of [Yun et al., 2020]). For each $f_{\mathcal{T},c} \in \overline{\mathcal{T}}_p^{2,1,1}$ and any $\epsilon > 0$, there exists $f_{\mathcal{T}} \in \mathcal{T}_p^{2,1,4}$ such that $\|f_{\mathcal{T}} - f_{\mathcal{T},c}\|_{L^2} \le \epsilon/3$.

*Proof.* See Appendix E.5.5 for a detailed proof. ☐

### E.4 All Together: Standard Transformers Approximate Compact-supported Continuous Functions

We summarize the results of Lemmas E.2, E.3 and E.7. Then we prove Lemma E.1.

Furthermore, to achieve the $\epsilon$ approximation error in Lemma E.1, we take $\delta = \mathcal{O}(\epsilon^{2/d})$ in Lemma E.3.

### E.5 Supplementary Proofs

We first present two preliminary concepts: selective shift operation and bijective column ID mapping in Appendix E.5.1.

Then we show

- Proof of Lemma E.2 in Appendix E.5.2
- Proof of Lemma E.4 in Appendix E.5.3
- Proof of Lemma E.5 in Appendix E.5.4
- Proof of Lemma E.7 in Appendix E.5.5

#### E.5.1 Preliminaries

Here, we give the definition of two preliminary concepts: selective shift operation and bijective column ID mapping.

**Selective Shift Operation.** This operation refers to shifting certain entries of the input selectively. To achieve this, we consider the following function $\xi(\cdot;\cdot) : \mathbb{R}^{d \times L} \to \mathbb{R}^{d \times L}$

$$\xi(X; b_Q) = e_1 u^\top X \sigma_H \left[ (u^\top X)^\top (u^\top X - b_Q \mathbb{1}_n^\top) \right], \tag{E.5}$$

where $X \in \mathbb{R}^{d \times L}$, $e_1 = (1, 0, 0, \cdots, 0)^\top \in \mathbb{R}^d$, and $b_Q \in \mathbb{R}$. $u \in \mathbb{R}^d$ is a vector to be determined.

To see the output, we consider the $j$-th column of $u^\top X \sigma_H \left[ (u^\top X)^\top (u^\top X - b_Q \mathbb{1}_n^\top) \right]$:

- If $u^\top X_{:,j} > b_Q$, it calculates argmax of $u^\top X$;
- If $u^\top X_{:,j} < b_Q$, it calculates argmin of $u^\top X$.

All rows of $\xi(X; b_Q)$ except the first row are zero. We consider the $j$-th entry of the first row in $\xi(X; b_Q)$, which is denoted as $\xi(X; b_Q)_{1,j}$. Then for all $j \in [L]$, we have

$$\xi(X; b_Q)_{1,j} = u^\top X \sigma_H \left[ (u^\top X)^\top (u^\top X_{:,j} - b_Q) \right] = \begin{cases} \max_k u^\top X_{:,k} & \text{if } u^\top X_{:,j} > b_Q, \\ \min_k u^\top X_{:,k} & \text{if } u^\top X_{:,j} < b_Q. \end{cases}$$

From this observation, we define a function parametrized by $b_Q$ and $b_Q'$ (with $b_Q < b_Q'$)

$$\xi(X; b_Q, b_Q') := \xi(X; b_Q) - \xi(X; b_Q'). \tag{E.6}$$

Then we have

$$\xi(X; b_Q, b_Q')_{1,j} = \begin{cases} \max_k u^\top X_{:,k} - \min_k u^\top X_{:,k}, & \text{if } b_Q < u^\top X_{:,j} < b_Q', \\ 0, & \text{others.} \end{cases}$$

We define an attention layer of the form $X \to X + \xi(X; b_Q, b_Q')$. For any column $X_{:,j}$, if $b_Q < u^\top X_{:,j} < b_Q'$, its first coordinate $X_{1,j}$ is shifted up by $\max_k u^\top X_{:,k} - \min_k u^\top X_{:,k}$, while all the other coordinates stay untouched. We call this the selective shift operation because we can choose $b_Q$ and $b_Q'$ to shift certain entries of the input selectively.

**Bijective Column ID Mapping.** We consider the input $G \in \mathcal{G}_\delta^+$ (Definition E.7). We use

$$J = L + 3L\delta^{-dL}, \text{ and } u = (1, \delta^{-1}, \delta^{-2}, \ldots, \delta^{-d+1}). \tag{E.7}$$

For any $j \in [L]$, we have the following two conclusions:

- If $G_{i,j} \geq 0$ for all $i \in [d]$, i.e., $G_{:,j} \in [j-1:\delta:j-\delta]^d$, then we have

$$u^\top G_{:,j} \in \left[\delta_j : \delta : \delta_j + \delta^{-d+1} - \delta\right], \text{ where } \delta_j = (j-1) \cdot \left(\frac{\delta - \delta^{-d+1}}{\delta - 1}\right). \quad \text{(E.8)}$$

The mapping $G_{:,j} \to u^\top G_{:,j}$ maps the elements in $[j-1:\delta:j-\delta]^d$ to $\left[\delta_j : \delta : \delta_j + \delta^{-d+1} - \delta\right]$. This is a bijection.

- If there exists $i \in [d]$ such that $G_{i,j} = -J + j$, then

$$u^\top G_{:,j} \leq -3L\delta^{-dL} + (j-1) \cdot \left(\frac{\delta^{-d+1} - \delta}{1 - \delta}\right) + \delta^{-d+1} < 0. \quad \text{(E.9)}$$

We say that $u^\top G_{:,j}$ gives the "column ID" for each possible value of $G_{:,j} \in [j-1:\delta:j-\delta]^d$.

**Remark E.3** (Illustration of Bijection Property). For the bijection property, we give the following illustration. Let $G_{:j} = (g_{1j}, g_{2j}, \cdots, g_{dj})^\top$ and $\overline{G}_{:j} = (\overline{g}_{1j}, \overline{g}_{2j}, \cdots, \overline{g}_{dj})^\top$. If $u^\top G_{:j} = u^\top \overline{G}_{:j}$ and $G_{:j} \neq \overline{G}_{:j}$, we deduce

$$(g_{1j} - \overline{g}_{1j}) + \delta^{-1}(g_{2j} - \overline{g}_{2j}) + \cdots + \delta^{-d+1}(g_{dj} - \overline{g}_{dj}) = 0. \quad \text{(E.10)}$$

Because $G_{:j} \neq \overline{G}_{:j}$, then there exists a $k$ $(k < d)$, such that $g_{kj} \neq \overline{g}_{kj}$ and $g_{ij} = \overline{g}_{ij} (i > k)$. We have

$$\left|\delta^{-k+1}(g_{kj} - \overline{g}_{kj})\right| \geq \delta^{-k+2}.$$

However,

$$\begin{aligned}
&\left|(g_{1j} - \overline{g}_{1j}) + \cdots + \delta^{-k+2}(g_{k-1,j} - \overline{g}_{k-1,j})\right| \\
&\leq |g_{1j} - \overline{g}_{1j}| + \cdots + \left|\delta^{-k+2}(g_{k-1,j} - \overline{g}_{k-1,j})\right| \\
&\leq (1 - \delta) + \cdots + \delta^{-k+2}(1 - \delta) \\
&< \delta^{-k+2}.
\end{aligned}$$

This contradicts with (E.10). Thus we prove the property of bijection.

### E.5.2 Proof of Lemma E.2

*Proof of Lemma E.2.* We restate the proof from [Yun et al., 2020] for completeness.

By the nature of the compact-supported continuous function, $f$ is uniformly continuous.

Because $\|\cdot\|_\infty$ is equivalent to $\|\cdot\|_F$ when the number of entries are finite, we have the following by the definition of uniform continuity.

For any $\epsilon/3 > 0$, there exists a $\delta^\star > 0$, such that for any $X, Y \in \mathbb{R}^{d \times L}$, and $\|X - Y\|_\infty < \delta^\star$, we have $\|f(X) - f(Y)\|_F < \epsilon/3$.

Then we perform the following steps following Definitions E.1 and E.2:

- We create a grid $\mathcal{G}_{\delta^\star}$ by choosing grid width $\delta^\star$. We also create cube $\mathcal{S}_G$ with respect to $G \in \mathcal{G}_{\delta^\star}$.

- For any grid point $G \in \mathcal{G}_{\delta^\star}$, we define $C_G \in \mathcal{S}_G$ as the center point of the cube $\mathcal{S}_G$.

- We define a piecewise-constant function $f_{\delta^\star}(X) = \sum_{L \in \mathcal{G}_{\delta^\star}} f(C_G) \mathbb{1}\{X \in \mathcal{S}_G\}$.

For any $X \in \mathcal{S}_G$, we have $\|X - C_G\|_\infty < \delta^\star$. According to the uniform continuity, we drive

$$\|f(X) - f_{\delta^\star}(X)\|_F = \|f(X) - f(C_G)\|_F < \epsilon/3.$$

This implies that $\|f - f_{\delta^\star}\|_{L^2} < \epsilon/3$ and completes the proof. $\qquad\square$

### E.5.3 Proof of Lemma E.4

We give the proof of Lemma E.4 by constructing the network to satisfy the requirements.

*Proof of Lemma E.4.* Recall the selective shift operation in Appendix E.5.1. The overall idea of the construction includes two steps:

- **Step 1:** For each $j \in [L]$, we stack $\delta^{-d}$ attention layers. For $g \in [\delta_j : \delta : \delta_j + \delta^{-d+1} - \delta]$ (E.8) in the increasing order, we use the attention layer as

$$\delta^{-d}\xi(\cdot; g - \delta/2, g + \delta/2). \tag{E.11}$$

  The total number of layers is $L\delta^{-d}$. These layers cast $G \in \widetilde{\mathcal{G}}_\delta$ to $L$ different entries required by Property 1 of Lemma E.4.

- **Step 2:** We add an extra single-head attention layer with the following attention part

$$L\delta^{-(L+1)d-1}\xi(\cdot; 0). \tag{E.12}$$

  This layer achieves a global shifting and casts different $G \in \widetilde{\mathcal{G}}_\delta$ to unique elements required by the Property 2 of Lemma E.4.

The two operations map $\widetilde{\mathcal{G}}_\delta$ and $\mathcal{G}_\delta^+ \setminus \widetilde{\mathcal{G}}_\delta$ to different sets, as required by Property 3 and Property 4 of Lemma E.4. The bounds $t_l$ and $t_r$ are calculated then.

Then, we give a detailed proof by showing the impact of the two steps and verifying the four properties of Lemma E.4. We achieve this by making a category division of $\mathcal{G}_\delta^+$:

- **Category 1:** $G \in \widetilde{\mathcal{G}}_\delta$, all entries in the point $G$ are between $0$ and $L - \delta$.

- **Category 2:** $G \in \mathcal{G}_\delta^+ \setminus \widetilde{\mathcal{G}}_\delta$, the point $G$ has at least one entry that equals to $-J$.

Let $u = (1, \delta^{-1}, \delta^{-2}, \ldots, \delta^{-d+1})$. Recall that $\delta_j = (j-1)(\delta - \delta^{-d+1})/(\delta - 1)$ for any $j \in [L]$ in (E.8).

**Category 1.** We denote $g_j := u^\top G_{:,j}$, then we have $g_1 < g_2 < \cdots < g_L$. The first $\delta^{-d}$ layers sweep the set $[\delta_j : \delta : \delta_j + \delta^{-d+1} - \delta], j \in [L]$ and apply selective shift operation on each element in the set. This means that selective shift operation will be applied to $g_1$ first, then $g_2$, followed by $g_3$, and so on.

- **The First Shift Operation.** In the first selective shift operation with $g$ going through $[\delta_1 : \delta : \delta_1 + \delta^{-d+1} - \delta]$, the $(1,1)$-th entry of $G$ (i.e., $G_{1,1}$) is shifted by the operation, while the other entries are left untouched. The updated value $\widetilde{G}_{1,1}$ is

$$\widetilde{G}_{1,1} = G_{1,1} + \delta^{-d}\left[\max_k \left(u^\top G_{:,k}\right) - \min_k \left(u^\top G_{:,k}\right)\right] = G_{1,1} + \delta^{-d}(g_L - g_1).$$

Therefore, the output of the layer after the operation is

$$\begin{pmatrix} \widetilde{G}_{:,1} & G_{:,2} & \cdots & G_{:,L} \end{pmatrix}.$$

Let $\widetilde{g}_1 := u^T \widetilde{G}_{:,1}$. We have

$$\widetilde{g}_1 = \widetilde{G}_{1,1} + \sum_{i=2}^{d} \delta^{-i+1} G_{i,1}$$

$$= G_{1,1} + \delta^{-d}(g_L - g_1) + \sum_{i=2}^{d} \delta^{-i+1} G_{i,1}$$

$$= g_1 + \delta^{-d}(g_L - g_1).$$

Then we deduce $g_L < \widetilde{g}_1$, because

$$\widetilde{g}_1 = g_1 + \delta^{-d}(g_L - g_1)$$

$$\geq 0 + \delta^{-d}\left[(L-1)\cdot\frac{\delta - \delta^{-d+1}}{\delta - 1} - \delta^{-d+1} + \delta\right] \qquad \left(\text{By (E.8)}\right)$$

$$= \delta^{-d}\left[(L-1)\frac{\delta}{1-\delta} + \delta + (L-1)\frac{\delta^{-d+1}}{1-\delta} - \delta^{-d+1}\right]$$

$$\geq \delta^{-d}\cdot\left((L-1)\frac{\delta}{1-\delta} + \delta\right)$$

$$= (L-1)\frac{\delta^{-d+1}}{1-\delta} + \delta^{-d+1}$$

$$> g_L. \qquad \left(\text{By } \delta < 1 \text{ and (E.8)}\right)$$

Thus, after updating, we have

$$\max u^\top \begin{pmatrix} \widetilde{G}_{:,1} & G_{:,2} & \cdots & G_{:,L} \end{pmatrix} = \max\{\widetilde{g}_1, g_2, \ldots, g_L\} = \widetilde{g}_1,$$

and the new minimum is $g_2$.

- **The Second Shift Operation.** In the second selective shift operation with $g$ going through $[\delta_2 : \delta : \delta_2 + \delta^{-d+1} - \delta]$, the $(1,2)$-th entry of $G$ (i.e., $G_{1,2}$) is shifted by the operation, while the other entries are left untouched. The updated value $\widetilde{G}_{1,2}$ is

$$\widetilde{G}_{1,2} = G_{1,2} + \delta^{-d}(\widetilde{g}_1 - g_2)$$

$$= G_{1,2} + \delta^{-d}(g_1 - g_2) + \delta^{-2d}(g_L - g_1).$$

Therefore, the output of the layer after the operation is

$$\begin{pmatrix} \widetilde{G}_{:,1} & \widetilde{G}_{:,2} & \cdots & G_{:,L} \end{pmatrix}.$$

We have

$$\widetilde{g}_2 := u^\top \widetilde{G}_{:,2}$$
$$= g_2 + \delta^{-d}(g_1 - g_2) + \delta^{-2d}(g_L - g_1).$$

Then we deduce $\widetilde{g}_1 < \widetilde{g}_2$, because

$$g_1 + \delta^{-d}(g_L - g_1) < g_2 + \delta^{-d}(g_1 - g_2) + \delta^{-2d}(g_L - g_1)$$
$$\iff (\delta^{-d} - 1)(g_2 - g_1) < \delta^{-d}(\delta^{-d} - 1)(g_L - g_1). \qquad \left(\text{By } \delta^{-d} > 1 \text{ and } g_L > g_2\right)$$

Thus, after updating, we have

$$\max u^\top \begin{pmatrix} \widetilde{G}_{:,1} & \widetilde{G}_{:,2} & \cdots & G_{:,L} \end{pmatrix} = \max\{\widetilde{g}_1, \widetilde{g}_2, \ldots, g_L\} = \widetilde{g}_2,$$

and the new minimum is $g_3$.

- **Repeating the Process.** By repeating this process, we show that the $j$-th shift operation shifts $G_{1,j}$ by $\delta^{-d}(\widetilde{g}_{j-1} - g_j)$. Then we have

$$\widetilde{g}_j := u^\top \widetilde{G}_{:,j}$$
$$= g_j + \sum_{k=1}^{j-1} \delta^{-kd}(g_{j-k} - g_{j-k+1}) + \delta^{-jd}(g_L - g_1).$$

We deduce $\widetilde{g}_{j-1} < \widetilde{g}_j$ holds for all $2 \leq j \leq L$, because

$$\widetilde{g}_{j-1} < \widetilde{g}_j$$
$$\iff g_{j-1} + \sum_{k=2}^{j-1} \delta^{-kd+d}(g_{j-k} - g_{j-k+1}) + \delta^{-(j-1)d}(g_L - g_1)$$
$$< g_j + \sum_{k=1}^{j-1} \delta^{-kd}(g_{j-k} - g_{j-k+1}) + \delta^{-jd}(g_L - g_1)$$
$$\iff \sum_{k=1}^{j-1} \delta^{-kd+d}(\delta^{-d} - 1)(g_{j-k+1} - g_{j-k}) < \delta^{-(j-1)d}(\delta^{-d} - 1)(g_L - g_1),$$

where the last inequality holds because

$$\sum_{k=1}^{j-1} \delta^{-kd+d}(g_{j-k+1} - g_{j-k})$$
$$< \delta^{-(j-1)d} \sum_{k=1}^{j-1}(g_{j-k+1} - g_{j-k})$$
$$< \delta^{-(j-1)d}(g_L - g_1).$$

Therefore, after the $j$-th selective shift operation, $\widetilde{g}_j$ is the new maximum among $\{\widetilde{g}_1, \ldots, \widetilde{g}_j, g_{j+1}, \ldots, g_L\}$ and $g_{j+1}$ is the new minimum.

- **After $L$ Shift Operations.** After the whole $L$ shift operations, the input $G$ is mapped to a new point $\widetilde{G}$, where $u^\top \widetilde{G} = \begin{pmatrix} \widetilde{g}_1 & \widetilde{g}_2 & \cdots & \widetilde{g}_L \end{pmatrix}$ and $\widetilde{g}_1 < \widetilde{g}_2 < \cdots < \widetilde{g}_L$. For the lower and upper bound of $\widetilde{g}_L$, we have the following lemma.

**Lemma E.8** (Lemma 10 of [Yun et al., 2020]). $\widetilde{g}_L = u^\top \widetilde{G}_{:,L}$ satisfies the following bounds:

$$\delta^{-(L-1)d+1}(\delta^{-d} - 1) \leq \widetilde{g}_L \leq L\delta^{-(L+1)d}.$$

- **Global Shifting by the Last Layer.** We note that after the above $L$ shift operations, there is another attention layer with attention part $L\delta^{-(L+1)d-1}\xi(\cdot;0)$. Since $0 < \widetilde{g}_1 < \cdots < \widetilde{g}_L$, it adds the following to each entry in the first row of $\widetilde{G}$:

$$L\delta^{-(L+1)d-1} \max_k u^\top \widetilde{G}_{:,k} = L\delta^{-(L+1)d-1}\widetilde{g}_L.$$

The output of this layer is defined to be the function $f_{\mathcal{T},c2}(G)$.

In summary, for any $G \in \widetilde{\mathcal{G}}_\delta$, $i \in [d]$, and $j \in [L]$, we have

$$f_{\mathcal{T},c2}(G)_{i,j} = \begin{cases} G_{1,j} + \delta_j^+ & \text{if } i = 1, \\ G_{i,j} & \text{if } 2 \le i \le d, \end{cases}$$

where $\delta_j^+ = \sum_{k=1}^{j-1} \delta^{-kd}(g_{j-k} - g_{j-k+1}) + \delta^{-jd}(g_L - g_1) + L\delta^{-(L+1)d-1}\widetilde{g}_L$.
For any $G \in \widetilde{\mathcal{G}}_\delta$ and $j \in [L]$,

$$u^\top f_{\mathcal{T},c2}(G)_{:,j} = \widetilde{g}_j + L\delta^{-(L+1)d-1}\widetilde{g}_L.$$

Next, we check the Property 1, Property 2 and Property 3 of Lemma E.4.

- **Checking Property 1 of Lemma E.4.** Given any $G \in \widetilde{\mathcal{G}}_\delta$, we already prove that

$$\widetilde{g}_1 < \widetilde{g}_2 < \cdots < \widetilde{g}_L,$$

All of them are distinct.

- **Checking Property 2 of Lemma E.4.** Note that the upper bound on $\widetilde{g}_L$ from Lemma E.8 also holds for other $\widetilde{g}_j$ ($j \in [L-1]$). For all $j \in [L]$, we have

$$L\delta^{-(L+1)d-1}\widetilde{g}_L \le u^\top f_{\mathcal{T},c2}(G)_{:,j} < L\delta^{-(L+1)d-1}\widetilde{g}_L + L\delta^{-(L+1)d}.$$

By Lemma E.8, two different $G, G' \in \widetilde{\mathcal{G}}_\delta$ are mapped to different $\widetilde{g}_L$ and $\widetilde{g}_L'$, and they differ at least by $\delta$. This means that the following two intervals are guaranteed to be disjoint:

$$[L\delta^{-(L+1)d-1}\widetilde{g}_L, L\delta^{-(L+1)d-1}\widetilde{g}_L + L\delta^{-(L+1)d}),$$
$$[L\delta^{-(L+1)d-1}\widetilde{g}_L', L\delta^{-(L+1)d-1}\widetilde{g}_L' + L\delta^{-(L+1)d}).$$

Thus, the entries of $u^\top f_{\mathcal{T},c2}(G)$ and $u^\top f_{\mathcal{T},c2}(G')$ are all distinct.

Now, we finish showing that the mapping $f_{\mathcal{T},c2}(\cdot)$ uses $(1/\delta)^d + 1$ attention layers to implement a contextual mapping on $\widetilde{\mathcal{G}}_\delta$.

- **Checking Property 3 of Lemma E.4.** Given Lemma E.8 and $u^\top f_{\mathcal{T},c2}(G)_{:,j} \in [L\delta^{-(L+1)d-1}\widetilde{g}_L, L\delta^{-(L+1)d-1}\widetilde{g}_L + L\delta^{-(L+1)d})$, for any $G \in \widetilde{\mathcal{G}}_\delta$, we have

$$u^\top f_{\mathcal{T},c2}(G)_{:,j} \ge L\delta^{-2(L+1)d}(\delta^{-d} - 1),$$
$$u^\top f_{\mathcal{T},c2}(G)_{:,j} < L^2\delta^{-2(L+1)d-1} + L\delta^{-(L+1)d}.$$

This proves that all $u^\top f_{\mathcal{T},c2}(L)_{:,j}$ are between $t_l$ and $t_r$, where

$$t_l = L\delta^{-2(L+1)d}(\delta^{-d} - 1),$$

$$t_r = L^2 \delta^{-2(L+1)d-1} + L\delta^{-(L+1)d}.$$

**Category 2.** Now we check the Property 4 of Lemma E.4. For the input points $G \in \mathcal{G}_\delta^+ \setminus \widetilde{\mathcal{G}}_\delta$, note that the point $G$ has at least one entry that equals to $-J + k, k \in [L-1]$. Let $g_j \coloneqq u^\top G_{:,j}$. Recall that whenever a column $G_{:,j}$ has an entry that equals to $-J + k, k \in [L-1]$, we have $g_j < 0$. Without loss of generality, assume that $g_1 < 0$.

Because the selective shift operation is applied to each element of $[0 : \delta : \delta_L + \delta^{-d+1} - \delta]$ and is not applied to negative values, thus we have $\min_k u^\top G_{:,k} = g_1 < 0$. $g_1$ never gets shifted upwards and remains the minimum for the whole time.

- **All $g_j$'s are Negative.** When all $g_j$'s are negative, selective shift operation never shifts the input $G$. Thus $\widetilde{G} = G$. Recall that $u^\top \widetilde{G}_{:,j} < 0$ for all $j \in [L]$. The last layer with attention part $L\delta^{-(L+1)d-1}\xi(\cdot; 0)$ adds $L\delta^{-(L+1)d-1} \min_k u^\top \widetilde{G}_{:,k} < 0$ to each entry in the first row of $\widetilde{G}$. This makes $\widetilde{G}$ remain negative. Therefore, $f_{\mathcal{T},c2}(G)$ satisfies $u^\top f_{\mathcal{T},c2}(G)_{:,j} < 0 < t_l$ for all $j \in [L]$.

- **Not All $g_j$'s are Negative.** Now consider the case where at least one $g_j$ is positive. Suppose that there are $k$ positive elements and they satisfy $g_{i_1} < g_{i_2} < \cdots < g_{i_k}$. Thus selective shift operation does not affect $g_i$, where $i \in [L] \setminus \{i_1, \ldots, i_k\}$. It shifts $g_{i_1}$ by

$$
\begin{aligned}
&\delta^{-d}(\max_k u^\top G_{:,k} - \min_k u^\top G_{:,k}) \\
&\geq \delta^{-d}(2L\delta^{-dL} - (L-1)\frac{\delta^{-d+1}-\delta}{1-\delta} - \delta^{-d+1} + (i_k - 1)\frac{\delta^{-d+1}-\delta}{1-\delta}) && \text{(By (E.9))} \\
&= \delta^{-d}(3L\delta^{-dL} - \delta^{-d+1} - (L - i_k)\frac{\delta^{-d+1}-\delta}{1-\delta}) \\
&\geq \delta^{-d} \cdot 2L\delta^{-dL} && \left(\text{By } \delta^{-1} \geq 2\right) \\
&= 2L\delta^{-(L+1)d}.
\end{aligned}
$$

The next shift operations shift $g_{i_2}, \ldots, g_{i_k}$ by an even larger amount. Therefore, at the end of the first $L(1/\delta)^d$ layers, we have $L\delta^{-(L+1)d} \leq \widetilde{g}_{i_1} \leq \cdots \leq \widetilde{g}_{i_k}$, and $\widetilde{g}_j < 0$ for all $j \in [L] \setminus \{i_1, \ldots, i_k\}$.

Then, we shift $G$ by the last layer. The last layer with attention part $L\delta^{-(L+1)d-1}\xi(\cdot; 0)$ acts differently for negative and positive $\widetilde{g}_j$'s. (i). For negative $\widetilde{g}_j$'s, it adds the following to $\widetilde{g}_j, j \in [L] \setminus \{i_1, \ldots, i_k\}$:

$$L\delta^{-(L+1)d-1} \min_k u^\top \widetilde{G}_{:,k} = L\delta^{-(L+1)d-1}g_1 < 0.$$

This term pushes them further to the negative side. (ii). For positive $\widetilde{g}_i$'s, it adds

$$L\delta^{-(L+1)d-1} \max_k u^\top \widetilde{G}_k = L\delta^{-(L+1)d-1}\widetilde{g}_{i_k} \geq 2L^2\delta^{-2(L+1)d-1}.$$

Thus they are all greater than or equal to $2L^2\delta^{-2(L+1)d+1}$. Note that

$$2L^2\delta^{-2(L+1)d-1} > t_r, \text{ where } t_r = L^2\delta^{-2(L+1)d-1} + L\delta^{-(L+1)d}.$$

Then the final output $f_{\mathcal{T},c2}(G)$ satisfies $u^\top f_{\mathcal{T},c2}(G)_{:,j} \notin [t_l, t_r]$, for all $j \in [L]$. This completes the verification of Property 4 of Lemma E.4.

In conclusion, we need $\mathcal{O}(L\delta^{-d})$ layers of modified self-attention layer to obtain our approximation. This completes the proof. □

### E.5.4 Proof of Lemma E.5

*Proof of Lemma E.5.* We restate the proof from [Yun et al., 2020] for completeness.

Note that $|\mathcal{G}_\delta^+| = (1/\delta + 1)^{dL} < \infty$, so the output of $f_{\mathcal{T},c2}(\mathcal{G}_\delta^+)$ has finite number of distinct real values. Let $M$ be the upper bound of all these possible values. By the construction of $f_{\mathcal{T},c2}$, $M > 0$.

**Construct the Layers:** $f_{\mathcal{T},c3}(f_{\mathcal{T},c2}(G)) = \mathbf{0}_{d \times L}$ **if** $G \in \mathcal{G}_\delta^+ \setminus \widetilde{\mathcal{G}}_\delta$. According to Lemma E.4, for all $j \in [L]$, we have $u^\top f_{\mathcal{T},c2}(G)_{:,j} \in [t_l, t_r]$ if $G \in \widetilde{\mathcal{G}}_\delta$, and $u^\top f_{\mathcal{T},c2}(G)_{:,j} \notin [t_l, t_r]$ if $G \in \mathcal{G}_\delta^+ \setminus \widetilde{\mathcal{G}}_\delta$. Due to this property, we add the following feed-forward layer.

**Definition E.8** (Feed-forward Layer 3). The vectors $u$ and $\mathbb{1}_L$ act as the weight parameters, and $\zeta_3(\cdot)$ acts as the activation function in the feed-forward layer.

$$X \to X - (M+1)\mathbb{1}_L\zeta_3(u^\top X), \quad \zeta_3(t) = \begin{cases} 0 & \text{if } t \in [t_l, t_r] \\ 1 & \text{if } t \notin [t_l, t_r]. \end{cases} \tag{E.13}$$

- **Case for** $G \in \mathcal{G}_\delta^+ \setminus \widetilde{\mathcal{G}}_\delta$**.** We have $\zeta_3(u^\top f_{\mathcal{T},c2}(G)) = \mathbb{1}_L^\top$. Thus, all the entries of the input are shifted by $-M-1$ and become strictly negative.

- **Case for** $G \in \widetilde{\mathcal{G}}_\delta$**.** We have $\zeta_3(u^\top f_{\mathcal{T},c2}(G)) = \mathbf{0}_L^\top$, so the output stays the same as the $f_{\mathcal{T},c2}(G)$.

With the input $f_{\mathcal{T},c2}(G)$, if $G \in \widetilde{\mathcal{G}}_\delta$, then $\zeta_3(u^\top f_{\mathcal{T},c2}(G)) = \mathbf{0}_L^\top$. Thus, the output stays the same as the input. If $G \in \mathcal{G}_\delta^+ \setminus \widetilde{\mathcal{G}}_\delta$, then $\zeta_3(u^\top f_{\mathcal{T},c2}(G)) = \mathbb{1}_L^\top$. Thus, all the entries of the input are shifted by $-M-1$ and become strictly negative.

Next, we map those negative entries to zero. For $i = 1, 2, \cdots, d$, we add the following layer:

**Definition E.9** (Feed-forward Layer 4). The vectors $u$ and $e_i$ act as the weight parameters and $\zeta_4(\cdot)$ acts as the activation function in the feed-forward layer.

$$X \to X + e_i\zeta_4((e_i)^\top X), \quad \zeta_4(t) = \begin{cases} -t & \text{if } t < 0 \\ 0 & \text{if } t \geq 0. \end{cases} \tag{E.14}$$

After these $d$ layers, the output for $G \in \mathcal{G}_\delta^+ \setminus \widetilde{\mathcal{G}}_\delta$ is a zero matrix, while the output for $G \in \widetilde{\mathcal{G}}_\delta$ remains $f_{\mathcal{T},c2}(G)$.

**Construct the Layers:** $f_{\mathcal{T},c3}(f_{\mathcal{T},c2}(G)) = A_G$ **if** $G \in \widetilde{\mathcal{G}}_\delta$**.** Each different $G$ is mapped to $L$ unique numbers $u^\top f_{\mathcal{T},c2}(G)$, which are at least $\delta$ apart from each other. We map each unique number to the corresponding output column as follows. We choose one $\overline{G} \in \widetilde{\mathcal{G}}_\delta$. For each $u^\top f_{\mathcal{T},c2}(\overline{G})_{:,j}, j \in [L]$, we add the following feed-forward layer.

**Definition E.10** (Feed-forward Layer 5). The vectors $u$ and $e_i$ act as the weight parameters, and $\zeta_4(\cdot)$ acts as the activation function in the feed-forward layer.

$$X \to X + \left((A_{\overline{G}})_{:,j} - f_{\mathcal{T},c2}(\overline{G})_{:,j}\right)\zeta_5(u^\top X - u^\top f_{\mathcal{T},c2}(\overline{G})_{:,j}\mathbb{1}_L^\top), \tag{E.15}$$

$$\zeta_5(t) = \begin{cases} 1 & -\delta/2 \leq t < \delta/2, \\ 0 & \text{others.} \end{cases} \tag{E.16}$$

- **Case for** $G \in \mathcal{G}_\delta^+ \setminus \widetilde{\mathcal{G}}_\delta$**.** Recall that the input $X$ of this layer is $f_{\mathcal{T},c2}(G)$. If $X$ is a zero matrix, which is the case for $G \in \mathcal{G}_\delta^+ \setminus \widetilde{\mathcal{G}}_\delta$, we have $u^\top X = \mathbf{0}_L^\top$. Then $u^\top X - u^\top f_{\mathcal{T},c2}(\overline{G})_{:,j}\mathbb{1}_L^\top < -t_l\mathbb{1}_L$. Since $t_l > \delta/2$, the output remains the same as $X$.

- **Case for** $G \in \widetilde{\mathcal{G}}_\delta$. Let the input $X$ be $f_{\mathcal{T},c2}(G)$, where $G \in \widetilde{\mathcal{G}}_\delta$ is not equal to $\overline{G}$. According to the Property 2 of Lemma E.4 and given a $j \in [L]$, $u^\top f_{\mathcal{T},c2}(G)_{:,k}, (k \in [L])$ differs from $u^\top f_{\mathcal{T},c2}(\overline{G})_{:,j}$ by at least $\delta$. Then we have

$$\zeta_5(u^\top f_{\mathcal{T},c2}(G) - u^\top f_{\mathcal{T},c2}(\overline{G})_{:,j}\mathbb{1}_L^\top) = \mathbf{0}_L^\top.$$

Thus the input is left untouched.

If $G = \overline{G}$, then

$$\zeta_5(u^\top f_{\mathcal{T},c2}(G) - u^\top f_{\mathcal{T},c2}(\overline{G})_{:,j}\mathbb{1}_L^\top) = (e_j)^\top.$$

Thus we shift the $j$-th column of $f_{\mathcal{T},c2}(G)$ to

$$f_{\mathcal{T},c2}(G)_{:,j} + ((A_{\overline{G}})_{:,j} - f_{\mathcal{T},c2}(\overline{G})_{:,j}) = f_{\mathcal{T},c2}(G)_{:,j} + ((A_G)_{:,j} - f_{\mathcal{T},c2}(G)_{:,j}) = (A_G)_{:,j}.$$

In other word, this layer maps the column $f_{\mathcal{T},c2}(G)_{:,j}$ to $(A_G)_{:,j}$, without affecting any other columns.

For each $G \in \widetilde{\mathcal{G}}_\delta$, we defer that we need one layer for each unique value of $u^\top f_{\mathcal{T},c2}(G)_{:,j}$. Note that there are $\mathcal{O}(\delta^{-dL})$ such numbers, so we use $\mathcal{O}(\delta^{-dL})$ layers to finish our construction.

This completes the proof. $\qquad\square$

### E.5.5 Proof of Lemma E.7

*Proof of Lemma E.7.* We restate the proof from [Yun et al., 2020] for completeness.

The proof follows two steps: (i) Approximate the modified self-attention layers. (ii) Approximate the modified feed-forward layers.

- **Step 1: Approximate the Modified Self-Attention Layers.**

  We achieve this by approximating the $\mathrm{Softmax}$ operator $\sigma_S$ with the $\mathrm{Hardmax}$ operator $\sigma_H$. Given a matrix $X \in \mathbb{R}^{d \times L}$, we have

  $$\sigma_S(\lambda X) \to \sigma_H(X), \quad \text{as} \quad \lambda \to \infty.$$

  The operator is the only difference between the normal and the modified self-attention layers. We approximate the modified self-attention layer in $\overline{\mathcal{T}}_p^{r,m,l}$ by the normal self-attention layer with the same number of heads $r$ and head size $m$.

- **Step2: Approximate the Modified Feed-Forward Layers.**

  We achieve this by approximating the activation function in $\Psi$ with four $\mathrm{ReLU}$ functions. From Definition E.3, we recall that $\Psi$ denotes three-piecewise functions with at least a constant piece. We consider the following $\zeta \in \Psi$:

  $$\zeta(x) = \begin{cases} b_1 & \text{if } x < c_1, \\ a_2 x + b_2 & \text{if } c_1 \leq x < c_2, \\ a_3 x + b_3 & \text{if } c_2 \leq x, \end{cases}$$

  where $a_2, a_3, b_1, b_2, b_3, c_1, c_2 \in \mathbb{R}$, and $c_1 < c_2$.

  We approximate $\zeta(x)$ by $\widetilde{\zeta}(x)$ composed of four $\mathrm{ReLU}$ functions:

$$
\begin{aligned}
\widetilde{\zeta}(x) =& b_1 + \frac{a_2 c_1 + b_2 - b_1}{\epsilon} \mathrm{ReLU}(x - c_1 + \epsilon) + \left( a_2 - \frac{a_2 c_1 + b_2 - b_1}{\epsilon} \right) \mathrm{ReLU}(x - c_1) \\
& + \left( \frac{a_3 c_2 + b_3 - a_2(c_2 - \epsilon) - b_2}{\epsilon} - a_2 \right) \mathrm{ReLU}(x - c_2 + \epsilon) \\
& + \left( a_3 - \frac{a_3 c_2 + b_3 - a_2(c_2 - \epsilon) - b_2}{\epsilon} \right) \mathrm{ReLU}(x - c_2) \\
=& \begin{cases} b_1 & \text{if } x < c_1 - \epsilon, \\ (a_2 c_1 + b_2 - b_1)(x - c_1)/\epsilon + a_2 c_1 + b_2 & \text{if } c_1 - \epsilon \leq x < c_1, \\ a_2 x + b_2 & \text{if } c_1 \leq x < c_2 - \epsilon, \\ (a_3 c_2 + b_3 - a_2(c_2 - \epsilon) - b_2)(x - c_2)/\epsilon + a_3 c_2 + b_3 & \text{if } c_2 - \epsilon \leq x < c_2, \\ a_3 x + b_3 & \text{if } c_2 \leq x. \end{cases}
\end{aligned}
$$

As $\epsilon \to 0$, we approximate $\zeta(x)$ by $\widetilde{\zeta}(x)$. The activation function is the only difference between the normal and modified feed-forward layers. We approximate the modified feed-forward layer in $\overline{\mathcal{T}}_p^{r,m,l}$ by the normal one.

Thus, for any $f_{\mathcal{T},c} \in \overline{\mathcal{T}}_p^{2,1,1}$, there exists a function $f_{\mathcal{T}} \in \mathcal{T}_p^{2,1,4}$ to approximate $f_{\mathcal{T},c}$.

This completes the proof. $\qquad\qquad\square$

# F    Proofs of Section 3

Our proofs are motivated by the approximation and estimation theory of U-Net-based diffusion models in [Chen et al., 2023]. We use transformer networks' universal approximation theory in Appendix E and the covering number to proceed with our proof. Specifically, we derive the approximation error bound in Appendix F.1 and the corresponding sample complexity bound in Appendix F.2. Then we show that the data distribution generated from the estimated score function converges toward a proximate area of the original one in Appendix F.3.

## F.1    Proof of Theorem 3.1

Here we present some auxiliary theoretical results in Appendix F.1.1 to prepare for our main proof of Theorem 3.1. Then we derive the approximation error bound of DiTs (i.e., the proof of Theorem 3.1) in Appendix F.1.2.

### F.1.1    Auxiliary Lemmas for Theorem 3.1.

We restate some auxiliary lemmas and their proofs from [Chen et al., 2023] for later convenience.

**Lemma F.1** (Lemma 16 of [Chen et al., 2023]). Consider a probability density function $p_h(h) = \exp\left(-C\|h\|_2^2/2\right)$ for $h \in \mathbb{R}^{d_0}$ and constant $C > 0$. Let $r_h > 0$ be a fixed radius. Then it holds

$$\int_{\|h\|_2 > r_h} p_h(h)\mathrm{d}h \leq \frac{2d_0\pi^{d_0/2}}{C\Gamma(d_0/2+1)}r_h^{d_0-2}\exp\left(-Cr_h^2/2\right),$$

$$\int_{\|h\|_2 > r_h} \|h\|_2^2 p_h(h)\mathrm{d}h \leq \frac{2d_0\pi^{d_0/2}}{C\Gamma(d_0/2+1)}r_h^{d_0}\exp\left(-Cr_h^2/2\right).$$

**Lemma F.2** (Lemma 2 of [Chen et al., 2023]). Suppose Assumption 2.2 holds and $g$ is defined as:

$$q(\bar{h}, t) = \int \frac{h\psi_t(\bar{h}|h)p_h(h)}{\int \psi_t(\bar{h}|h)p_h(h)\mathrm{d}h}\mathrm{d}h, \quad \bar{h} = B^\top \bar{x}.$$

Given $\epsilon > 0$, with $r_h = c\left(\sqrt{d_0\log(d_0/T_0) + \log(1/\epsilon)}\right)$ for an absolute constant $c$, it holds

$$\left\|q(\bar{h}, t)\mathbb{1}\{\|\bar{h}\|_2 \geq r_h\}\right\|_{L^2(P_t)} \leq \epsilon, \text{ for } t \in [T_0, T].$$

**Lemma F.3** (Theorem 1 of [Chen et al., 2023]). We denote

$$\tau(r_h) = \sup_{t\in[T_0,T]} \sup_{\bar{h}\in[0,r_h]^d} \left\|\frac{\partial}{\partial t}q(\bar{h}, t)\right\|_2.$$

With $q(\bar{h}, t) = \int h\psi_t(\bar{h}|h)p_h(h)/(\int \psi_t(\bar{h}|h)p_h(h)\mathrm{d}h)\mathrm{d}h$ and $p_h$ satisfies Assumption 2.2, we have a coarse upper bound for $\tau(r_h)$:

$$\tau(r_h) = \mathcal{O}\left(\frac{1+\beta^2(t)}{\beta(t)}\left(L_{s_+} + \frac{1}{\sigma(t)}\right)\sqrt{d_0}r_h\right) = \mathcal{O}\left(e^{T/2}L_{s_+}r_h\sqrt{d_0}\right).$$

**Lemma F.4** (Lemma 10 of [Chen et al., 2020b]). For any given $\epsilon > 0$, and $L$-Lipschitz function $g$ defined on $[0, 1]^{d_0}$, there exists a continuous function $\bar{f}$ constructed by trapezoid function, such that

$$\left\|g - \bar{f}\right\|_\infty \leq \epsilon.$$

Moreover, the Lipschitz continuity of $\bar{f}$ is bounded:

$$\left|\bar{f}(x) - \bar{f}(y)\right| \leq 10d_0 L \|x - y\|_2 \quad \text{for any} \quad x, y \in [0, 1]^{d_0}.$$

### F.1.2 Main Proof of Theorem 3.1

*Proof of Theorem 3.1.* With $\nabla \log p_t^h (\bar{h}) = B^\top s_+(\bar{h}, t)$, we have the following in (2.4)

$$q(\bar{h}, t) = \sigma(t) \nabla \log p_t^h (\bar{h}) + B^\top \bar{x} = \sigma(t) B^\top (s_+(\bar{h}, t) + \bar{x}). \tag{F.1}$$

We proceed as follows:

- **Step 1.** Approximate $q(\bar{h}, t)$ with a compact-supported continuous function $\bar{f}(\bar{h}, t)$.

- **Step 2.** Approximate $\bar{f}(\bar{h}, t)$ with a transformer network.

**Step 1. Approximate $q(\bar{h}, t)$ with a Compact-supported Continuous Function $\bar{f}(\bar{h}, t)$.** We partition $\mathbb{R}^{d_0}$ into a compact subset $H_1 := \{\bar{h} | \|\bar{h}\|_2 \leq r_h\}$ and its complement $H_2$, where $r_h$ is to be determined later. We approximate $q(\bar{h}, t)$ on the two subsets respectively and then prove $\bar{f}$'s continuity. Such a step achieves an estimation error of $\sqrt{d_0}\epsilon$ between $q(\bar{h}, t)$ and $\bar{f}(\bar{h}, t)$. We show the main proof here.

- **Approximation on $H_2 \times [T_0, T]$.** For any $\epsilon > 0$, we take $r_h = c(\sqrt{d_0 \log(d_0/T_0) - \log \epsilon})$. From Lemma F.2, we have

$$\left\|q(\bar{h}, t)\mathbb{1}\{\|\bar{h}\|_2 \geq r_h\}\right\|_{L^2(P_t)} \leq \epsilon \quad \text{for} \quad t \in [T_0, T].$$

So we set $\bar{f}(\bar{h}, t) = 0$ on $H_2 \times [T_0, T]$.

- **Approximation on $H_1 \times [T_0, T]$.** On $H_1 \times [T_0, T]$, we approximate $q(\bar{h}, t)$ by approximating each coordinate $q_k(\bar{h}, t)$ respectively, where $q(\bar{h}, t) = [q_1(\bar{h}, t), q_2(\bar{h}, t), \cdots, q_{d_0}(\bar{h}, t)]$. We rescale the input by $y' = (\bar{h} + r_h \mathbb{1})/2r_h$ and $t' = t/T$. Then the transformed input space is $[0, 1]^{d_0} \times [T_0/T, 1]$. We implement such a transformation by a single feed-forward layer.

By Assumption 2.3, on-support score $s_+(\bar{h}, t)$ is $L_{s_+}$-Lipschitz in $\bar{h}$. This implies $q(\bar{h}, t)$ is $(1 + L_{s_+})$-Lipschitz in $\bar{h}$. When taking the transformed inputs, $g(y', t') = q(2r_h y' - r_h \mathbb{1}, Tt')$ becomes $2r_h(1 + L_{s_+})$-Lipschitz in $y'$. Similarly, each coordinate $g_k(y', t)$ is also $2r_h(1 + L_{s_+})$-Lipschitz in $y'$. Here we take $L_h = 1 + L_{s_+}$.

Besides, $g(y', t')$ is $T\tau(r_h)$-Lipsichitz with respect to $t$, where

$$\tau(r_h) = \sup_{t \in [T_0, T]} \sup_{\bar{h} \in [0, r_h]^d} \left\|\frac{\partial}{\partial t} q(\bar{h}, t)\right\|_2.$$

We have a coarse upper bound for $\tau(r_h)$ in Lemma F.3. We restate it here for convenience

$$\tau(r_h) = \mathcal{O}\left(\frac{1 + \beta^2(t)}{\beta(t)}\left(L_{s_+} + \frac{1}{\sigma(t)}\right)\sqrt{d_0}r_h\right) = \mathcal{O}\left(e^{T/2}L_{s_+}r_h\sqrt{d_0}\right).$$

In conclusion, each $g_k(y', t)$ is Lipsichitz continuous. So we can apply Lemma F.4 to determine $\bar{f}_k(y', t)$ for approximating each coordinate. We concatenate $\bar{f}_i$'s together and construct $\bar{f} = [\bar{f}_1, \ldots, \bar{f}_{d_0}]^\top$. According to the construction in Lemma F.4 and for any given $\epsilon$, we achieve

$$\sup_{y', t' \in [0, 1]^d \times [T_0/T, 1]} \left\|\bar{f}(y', t') - g(y', t')\right\|_\infty \leq \epsilon,$$

Considering the input rescaling (i.e., $\bar{h} \to y'$ and $t \to t'$), we obtain:

– The constructed function is Lipschitz continuous in $\bar{h}$. For any $\bar{h}_1, \bar{h}_2 \in H_1$ and $t \in [T_0, T]$, it holds

$$\left\| \bar{f}(\bar{h}_1, t) - \bar{f}(\bar{h}_2, t) \right\|_\infty \leq 10 d_0 L_h \left\| \bar{h}_1 - \bar{h}_2 \right\|_2. \tag{F.2}$$

– The function is also Lipschitz in $t$. For any $t_1, t_2 \in [T_0, T]$ and $\left\| \bar{h} \right\|_2 \leq r_h$, it holds

$$\left\| \bar{f}(\bar{h}, t_1) - \bar{f}(\bar{h}, t_2) \right\|_\infty \leq 10 \tau(r_h) \| t_1 - t_2 \|_2.$$

Due to the fact that the construction of $\bar{f}(\bar{h}, t)$ is based on trapezoid function, we have $\bar{f}(\bar{h}, t) = 0$ for $\left\| \bar{h} \right\|_2 = r_h$ and any $t \in [T_0, T]$. Thus, the two parts of $\bar{f}(\bar{h}, t)$ can be joined together. To be more specific, the above Lipschitz continuity in $\bar{h}$ extends to the whole $\mathbb{R}^{d_0}$.

• **Approximation Error Analysis under $L^2$ Norm.** The $L^2$ approximation error of $\bar{f}$ can be decomposed into two terms:

$$\left\| q(\bar{h}, t) - \bar{f}(\bar{h}, t) \right\|_{L^2(P_t^h)}$$
$$= \left\| (q(\bar{h}, t) - \bar{f}(\bar{h}, t)) \mathbb{1}\{ \left\| \bar{h} \right\|_2 < r_h \} \right\|_{L^2(P_t^h)} + \left\| q(\bar{h}, t) \mathbb{1}\{ \left\| \bar{h} \right\|_2 > r_h \} \right\|_{L^2(P_t^h)}.$$

The second term in the RHS above has already been bounded with the selection of $r_h$:

$$\left\| g(\bar{h}, t) \mathbb{1}\{ \left\| \bar{h} \right\|_2 > r_h \} \right\|_{L^2(P_t^h)} \leq \epsilon.$$

The first term is bounded by:

$$\left\| (q(\bar{h}, t) - \bar{f}(\bar{h}, t)) \mathbb{1}\{ \left\| \bar{h} \right\|_2 < r_h \} \right\|_{L^2(P_t^h)}$$
$$\leq \sqrt{d_0} \sup_{y', t' \in [0,1]^d \times [T_0/T, 1]} \left\| \bar{f}(y', t') - g(y', t') \right\|_\infty$$
$$\leq \sqrt{d_0} \epsilon.$$

Then we obtain

$$\left\| q(\bar{h}, t) - \bar{f}(\bar{h}, t) \right\|_{L^2(P_t^h)} \leq (\sqrt{d_0} + 1)\epsilon.$$

If we substitute $\epsilon$ with $\epsilon/2$, we obtain that the approximation error of $\bar{f}(\bar{h}, t)$ is $\sqrt{d_0}\epsilon$.

**Step 2. Approximate $\bar{f}(\bar{h}, t)$ by a Transformer.** This step is based on the universal approximation of transformers for the compact-supported continuous function in Lemma E.1. DiT uses time point $t$ to calculate the scale and shift value in the transformer backbone [Peebles and Xie, 2023]. It also transforms an input picture into a sequential version. We ignore time point $t$ in the notation of the transformer network in DiT. Recall the reshape layer $R(\cdot)$ in Definition 3.1, we consider using $f(\cdot) := R^{-1} \circ f_\mathcal{T} \circ R(\cdot)$ to approximate $\bar{f}_t(\cdot) := \bar{f}(\cdot, t)$, where $f_\mathcal{T} \in \mathcal{T}_p^{2,1,4}$.

• **Overall Approximation Error.** With Lemma E.1, we approximate $\bar{f}_t(\cdot)$ with $\widehat{f}(\cdot) := R^{-1} \circ \widehat{f}_\mathcal{T} \circ R(\cdot)$. We denote

$$H = R(\bar{h}).$$

We have

$$\left\| \bar{f}_t(\bar{h}) - \widehat{f}(\bar{h}) \right\|_{L^2(P_t^h)} = \left( \int_{P_t^h} \left\| \bar{f}_t(\bar{h}) - \widehat{f}(\bar{h}) \right\|_2^2 \mathrm{d}h \right)^{1/2}$$

$$= \left( \int_{P_t^h} \left\| R \circ \bar{f}_t \circ R^{-1}(H) - R \circ \widehat{f} \circ R^{-1}(H) \right\|_F^2 dh \right)^{1/2}$$

$$= \left( \int_{P_t^h} \left\| R \circ \bar{f}_t \circ R^{-1}(H) - \widehat{f}_{\mathcal{T}}(H) \right\|_F^2 dh \right)^{1/2}$$

$$\leq \epsilon. \tag{F.3}$$

Along with Step 1, we obtain

$$\left\| q(\bar{h}, t) - \widehat{f}(\bar{h}) \right\|_{L^2(P_t^h)} \leq \left\| q(\bar{h}, t) - \bar{f}(\bar{h}, t) \right\|_{L^2(P_t^h)} + \left\| \bar{f}(\bar{h}, t) - \widehat{f}(\bar{h}) \right\|_{L^2(P_t^h)} \leq (1 + \sqrt{d_0})\epsilon.$$

The constructed approximator to $\nabla \log p_t(x)$ is $s_{\widehat{W}} = (B\widehat{f}(B^\top x, t) - x)/\sigma(t)$, and the approximation error is

$$\left\| \nabla \log p_t(\cdot) - s_{\widehat{W}}(\cdot, t) \right\|_{L^2(P_t)} \leq \frac{1 + \sqrt{d_0}}{\sigma(t)} \epsilon \quad \text{for any} \quad t \in [T_0, T].$$

- **Settling-down of Hyperparameters.** We settle down the hyperparameters to configure our network here. We refer to Appendix E.2 for some of the following calculations.

  1. **Model Architecture Depth $K$.**
     From Lemma E.6, we have $K = \mathcal{O}((1/\delta)^{dL})$. To achieve $\epsilon$-error approximation, we set $\delta = \mathcal{O}\left(\epsilon^{2/d}\right)$ according to Lemma E.3. Thus we obtain

     $$K = \mathcal{O}\left(\epsilon^{-2L}\right). \tag{F.4}$$

  2. **Lipchitz Upperbound for Transformer: $L_{\mathcal{T}}$.**
     We denote $\bar{f}_{t,R}(\cdot) = R \circ \bar{f}_t \circ R^{-1}(\cdot)$. We get the Lipshitz upper bound for $\widehat{f}_{\mathcal{T}} \in \mathcal{T}_p^{2.1.4}$ in the following way

     $$\left\| \widehat{f}_{\mathcal{T}}(H_1) - \widehat{f}_{\mathcal{T}}(H_2) \right\|_F \leq \left\| \widehat{f}_{\mathcal{T}}(H_1) - \bar{f}_{t,R}(H_1) \right\|_F + \left\| \bar{f}_{t,R}(H_1) - \bar{f}_{t,R}(H_2) \right\|_F$$
     $$+ \left\| \bar{f}_{t,R}(H_2) - \widehat{f}_{\mathcal{T}}(H_2) \right\|_F$$
     $$\leq 2\epsilon + \left\| \bar{f}_{t,R}(H_1) - \bar{f}_{t,R}(H_2) \right\|_F \qquad \text{(By (F.3))}$$
     $$\leq 2\epsilon + 10d_0 L_{s_+} \|H_1 - H_2\|_F. \qquad \text{(By (F.2))}$$

     Then we get

     $$L_{\mathcal{T}} = \mathcal{O}\left(d_0 L_{s_+}\right). \tag{F.5}$$

  3. **Model Output Bound for $\mathcal{S}_{\mathcal{T}_p^{2,1,4}}$.**
     For the output of the constructed transformer $\widehat{f}_{\mathcal{T}}(\cdot)$, according to Lemma E.5, we have $\widehat{f}_{\mathcal{T}}(O) = O$, where $O = \mathbf{0}_{d \times L}$. Thus, with the Lipschitz upperbound $\mathcal{O}(d_0 L_{s_+})$, we have $\|\widehat{f}_{\mathcal{T}}(H)\|_F = \mathcal{O}(d_0 L_{s_+} r_h)$, where $\|H\|_F \leq r_h$. With $r_h = c(\sqrt{d_0 \log(d_0/T_0) + \log(1/\epsilon)})$, we obtain

     $$C_{\mathcal{T}} = \mathcal{O}\left(d_0 L_{s_+} \cdot \sqrt{d_0 \log(d_0/T_0) + \log(1/\epsilon)}\right). \tag{F.6}$$

  4. **Model Parameters Bound: $C_{OV}^{2,\infty}, C_{OV}, C_{KQ}^{2,\infty}, C_{KQ}, C_E$.**
     By definition, we have:

     $$\left\| (W_{OV}^i)^\top \right\|_{2,\infty} \leq C_{OV}^{2,\infty}, \ \left\| (W_{OV}^i)^\top \right\|_2 \leq C_{OV}, \ \left\| W_{KQ}^i \right\|_{2,\infty} \leq C_{KQ}^{2,\infty}, \ \left\| W_{KQ}^i \right\|_2 \leq C_{KQ},$$

where $i = 1, 2$. For simplicity, we omit $i$ hereafter, which does not affect our discussion.

Recall that $\|Z\|_{2,\infty}$ denotes the $2, \infty$-norm, where the 2-norm is over columns and $\infty$-norm is over rows. By the construction of modified attention layers (E.11) and (E.12) in Appendix E.5.3, we consider $W_{OV}$ to have the largest norm, i.e.,

$$W_{OV} = L\delta^{-(L+1)d-1} \cdot \begin{pmatrix} 1 & \delta^{-1} & \cdots & \delta^{-d+1} \\ 0 & 0 & \cdots & 0 \\ \vdots & \vdots & \cdots & \vdots \\ 0 & 0 & \cdots & 0 \end{pmatrix}.$$

We give the following upper bounds

$$\left\|W_{OV}^\top\right\|_{2,\infty} = Ld\delta^{-(L+2)d} = \mathcal{O}\left(\delta^{-Ld}\right), \tag{F.7}$$

$$\left\|W_{OV}^\top\right\|_2 = \sup_{\|x\|_2=1} \left\|W_{OV}^\top x\right\|_2 = L\delta^{-(L+1)d-1} \cdot \sqrt{\sum_{i=0}^{d-1} \delta^{-2i}} = \mathcal{O}\left(\delta^{-Ld}\right). \tag{F.8}$$

By (E.11) and (E.12) in Appendix E.5.3, and the self-attention layers in Appendix E.5.5, we consider $W_{KQ}$ to have the largest norm, i.e.,

$$W_{KQ} := \begin{pmatrix} 1 \\ \delta^{-1} \\ \vdots \\ \delta^{-d+1} \end{pmatrix} \left(1, \delta^{-1}, \cdots, \delta^{-d+1}\right) = \begin{pmatrix} 1 & \delta^{-1} & \cdots & \delta^{-d+1} \\ \delta^{-1} & \delta^{-2} & \cdots & \delta^{-d} \\ \vdots & \vdots & \cdots & \vdots \\ \delta^{-d+1} & \delta^{-d} & \cdots & \delta^{-2d+2} \end{pmatrix}.$$

Then we have

$$\|W_{KQ}\|_{2,\infty} = \sqrt{\sum_{i=0}^{d-1} \delta^{-2i-2d+2}} = \mathcal{O}(\delta^{-2d}), \tag{F.9}$$

$$\|W_{KQ}\|_2 = \sup_{\|x\|_2=1} \|W_{KQ}x\|_2 = \delta^{-2d+2} = \mathcal{O}(\delta^{-2d}). \tag{F.10}$$

We substitute $\delta$ with $\mathcal{O}\left(\epsilon^{2/d}\right)$ (according to Appendix E.4) and get:

$$C_{OV}^{2,\infty} = (1/\epsilon)^{\mathcal{O}(1)},$$
$$C_{OV} = (1/\epsilon)^{\mathcal{O}(1)},$$
$$C_{KQ}^{2,\infty} = (1/\epsilon)^{\mathcal{O}(1)},$$
$$C_{KQ} = (1/\epsilon)^{\mathcal{O}(1)}.$$

From the construction of positional encoder (E.4) in Appendix E.2, we have

$$E = \begin{pmatrix} 0 & 1 & \cdots & L-1 \\ 0 & 1 & \cdots & L-1 \\ \vdots & \vdots & \vdots & \vdots \\ \vdots & \vdots & \vdots & \vdots \\ 0 & 1 & \cdots & L-1 \end{pmatrix}.$$

We deduce

$$\left\|E^\top\right\|_{2,\infty} = \sqrt{L}(L-1) = \mathcal{O}(L^{3/2}).$$

Thus we have

$$C_E = \mathcal{O}(L^{3/2}). \tag{F.11}$$

5. **Parameters Bound in Feed Forward Layers:** $C_F^{2,\infty}, C_F$.
   Recall the construction of modified feed-forward layers in the proof of Lemma E.4, which includes Definitions E.5, E.6 and E.8 to E.10. With the approximation by normal feed-forward layers in Appendix E.5.5, we consider the weight parameters with the largest norm in the feed-forward layers, i.e.,

$$W_1 := \begin{pmatrix} 1 \\ 1 \\ 1 \\ 1 \end{pmatrix} \left(1, \delta^{-1}, \cdots, \delta^{-d+1}\right) = \begin{pmatrix} 1 & \delta^{-1} & \cdots & \delta^{-d+1} \\ 1 & \delta^{-1} & \cdots & \delta^{-d+1} \\ 1 & \delta^{-1} & \cdots & \delta^{-d+1} \\ 1 & \delta^{-1} & \cdots & \delta^{-d+1} \end{pmatrix} \in \mathbb{R}^{4 \times d}.$$

Then we have

$$C_F^{2,\infty} = \mathcal{O}\left(\sqrt{\sum_{i=0}^{d-1} \delta^{-2i}}\right) = \mathcal{O}\left(\delta^{-d}\right) \tag{F.12}$$

$$= (1/\epsilon)^{\mathcal{O}(1)}. \qquad \left(\text{By setting } \delta = \mathcal{O}(\epsilon^{2/d}) \text{ according to Appendix E.4}\right)$$

and

$$C_F = \sup_{\|x\|_2 = 1} \|W_1 x\|_2 = \mathcal{O}\left(\delta^{-d}\right) \tag{F.13}$$

$$= (1/\epsilon)^{\mathcal{O}(1)}. \qquad \left(\text{By setting } \delta = \mathcal{O}(\epsilon^{2/d}) \text{ according to Appendix E.4}\right)$$

This completes the proof. $\qquad\qquad\qquad\qquad\qquad\qquad\qquad\qquad\qquad\qquad\qquad\qquad\square$

## F.2 Proof of Theorem 3.2

Here we present the auxiliary theoretical results about the covering number of transformer networks in Appendix F.2.1. The results are based on [Edelman et al., 2022, Theorem A.17]. Then we derive the sample complexity bound of DiTs (i.e., the proof of Theorem 3.2) in Appendix F.2.

### F.2.1 Auxiliary Lemmas for Theorem 3.2

**Lemma F.5** (Lemma 15 of [Chen et al., 2023]). Let $\mathcal{G}$ be a bounded function class. Then there exists a constant $b$ such that the output of any $g \in \mathcal{G} : \mathbb{R}^{d_0} \mapsto [0, b]$ is bounded by $b$. Let $z_1, z_2, \cdots, z_n \in \mathbb{R}^{d_0}$ be i.i.d. random variables. For any $\delta \in (0, 1), a \leq 1$, and $c > 0$, we have

$$P\left(\sup_{g \in \mathcal{G}} \frac{1}{n} \sum_{i=1}^{n} g(z_i) - (1 + a)\mathbb{E}\left[g(z)\right] > \frac{(1 + 3/a)B}{3n} \log \frac{\mathcal{N}(c, \mathcal{G}, \|\cdot\|_\infty)}{\delta} + (2 + a)c\right) \leq \delta,$$

$$P\left(\sup_{g \in \mathcal{G}} \mathbb{E}\left[g(z)\right] - \frac{1 + a}{n} \sum_{i=1}^{n} g(z_i) > \frac{(1 + 6/a)B}{3n} \log \frac{\mathcal{N}(c, \mathcal{G}, \|\cdot\|_\infty)}{\delta} + (2 + a)c\right) \leq \delta.$$

Now, we give the definition of the covering number as follows.

**Definition F.1** (Covering Number). Given a function class $\mathcal{F}$ and a data distribution $P$. Sample n data points $\{X_i\}_{i=1}^{n}$ from $P$. For any $\epsilon > 0$, the covering number $\mathcal{N}(\epsilon, \mathcal{F}, \{X_i\}_{i=1}^{n}, \|\cdot\|)$ is the smallest size of a collection (a cover) $\mathcal{C} \in \mathcal{F}$, such that for any $f \in \mathcal{F}$, there exists a $\widehat{f} \in \mathcal{C}$ satisfying

$$\max_i \left\| f(X_i) - \widehat{f}(X_i) \right\| \leq \epsilon.$$

Furthermore, we define the covering number with respect to the data distribution as

$$\mathcal{N}(\epsilon, \mathcal{F}, \|\cdot\|) = \sup_{\{X_i\}_{i=1}^{n} \sim P} \mathcal{N}(\epsilon, \mathcal{F}, \{X_i\}_{i=1}^{n}, \|\cdot\|).$$

Then we give the covering number of the transformer networks.

**Lemma F.6** (Modified from Theorem A.17 of [Edelman et al., 2022]). Let $\mathcal{T}_p^{r,m,l}(K, C_{\mathcal{T}}, C_{OV}^{2,\infty}, C_{OV}, C_{KQ}^{2,\infty}, C_{KQ}, C_F^{2,\infty}, C_F, C_E, L_{\mathcal{T}})$ represent the class of functions of $K$-layer transformer blocks satisfying the norm bound for matrix and Lipsichitz property for feed-forward layers. Then for all data point $\|X\|_{2,\infty} \leq C_X$, we have

$$\log \mathcal{N}(\epsilon_c, \mathcal{T}_p^{r,m,l}(K, C_{\mathcal{T}}, C_{OV}^{2,\infty}, C_{OV}, C_{KQ}^{2,\infty}, C_{KQ}, C_F^{2,\infty}, C_F, C_E, L_{\mathcal{T}}), \|\cdot\|_2)$$

$$\leq \frac{\log(nL)}{\epsilon_c^2} \cdot \left( \sum_{i=1}^{K} \alpha^{\frac{2}{3}} \left( d^{\frac{2}{3}} \left( C_F^{2,\infty} \right)^{\frac{4}{3}} + d^{\frac{2}{3}} \left( 2(C_F)^2 C_{OV} C_{KQ}^{2,\infty} \right)^{\frac{2}{3}} + \tau m^{\frac{2}{3}} \left( (C_F)^2 C_{OV}^{2,\infty} \right)^{\frac{2}{3}} \right) \right)^3,$$

where $\alpha := \prod_{j<i}(C_F)^2 C_{OV}(1 + 4C_{KQ})(C_X + C_E)$.

**Remark F.1.** We modify [Edelman et al., 2022, Theorem A.17] in seven aspects:

1. We do not consider the last linear layer in the model, which converts each column vector of the transformer output to a scalar. Therefore, we ignore the item related to the last linear layer in [Edelman et al., 2022, Theorem A.17].

2. We do not consider the normalization layer in our model. Because the normalization layer $\prod_{\text{norm}}(\cdot)$ in the original proof only ensures that $\|\prod_{\text{norm}}(X_1) - \prod_{\text{norm}}(X_2)\|_{2,\infty} \leq \|X_1 - X_2\|_{2,\infty}$, ignoring this layer does not change the result.

3. Our activation function is $\mathrm{ReLU}$. Thus, we replace the Lipschitz upperbound of activate function by 1.

4. We consider the positional encoding (E.4). Then we need to replace the upperbound $C_X$ for the inputs with the upperbound $C_X + C_E$. Besides, for multi-layer transformer, the original conclusion in [Edelman et al., 2022, Theorem A.17] uses 1 as the upperbound for the $2, \infty$-norm of inputs. We incorporate the upperbound for the inputs into the result stated in Lemma F.6.

5. We use (2.7) as the feed-forward layer, including two linear layers and a residual layer. Thus, we replace the original upperbound for the norm of weight matrix with the upperbound for the norm of $I_d + W_2 W_1$ in Lemma F.6. In the following, we use $\mathcal{O}$ to estimate the log-covering number, thus we ignore the item for $I_d$ here for convenience. This is the same for the self-attention layer.

6. We use multi-head attention, and incorporate the number of heads $\tau$ into our result, which is similar to [Edelman et al., 2022, Theorem A.12].

7. In our work, we use the transformer $\mathcal{T}_p^{2,1,4}$, i.e., $\tau = 2, m = 1$.

### F.2.2 Proof of Theorem 3.2

*Proof of Theorem 3.2.* Our proof is built on [Chen et al., 2023, Appendix B.2]. For one data sample, we define the empirical score matching loss objective (2.1) as follows

$$\ell(x; s_{\widehat{W}}) = \frac{1}{T - T_0} \int_{T_0}^{T} \mathbb{E}_{x_t|x_0=x}[\left\| \nabla_{x_t} \log \psi_t(x_t|x_0) - s_{\widehat{W}}(x_t, t) \right\|_2^2] \mathrm{d}t.$$

Then we define $\mathcal{L}(s_{\widehat{W}}) = \mathbb{E}_{x \sim P_0}\left[ \ell(x; s_{\widehat{W}}) \right]$.

Following [Chen et al., 2023, Appendix B.2], for any $a \in (0, 1)$, we have

$$\mathcal{L}(s_{\widehat{W}}) \\
\leq \underbrace{\mathcal{L}^{\mathrm{trunc}}(s_{\widehat{W}}) - (1+a)\widehat{\mathcal{L}}^{\mathrm{trunc}}(s_{\widehat{W}})}_{(I)} + \underbrace{\mathcal{L}(s_{\widehat{W}}) - \mathcal{L}^{\mathrm{trunc}}(s_{\widehat{W}})}_{(II)} + (1+a) \underbrace{\inf_{s_W \in \mathcal{S}_{\mathrm{NN}}} \widehat{\mathcal{L}}(s_W)}_{(III)},$$

where

$$\mathcal{L}^{\mathrm{trunc}}(s_{\widehat{W}}) := \mathbb{E}_{x \sim P_0}\left[ \ell^{\mathrm{trunc}}(x; s_{\widehat{W}}) \right] = \mathbb{E}_{x \sim P_0}\left[ \ell(x; s_{\widehat{W}}) \mathbb{1}\{\|x\|_2 \leq r_x\} \right], \quad r_x > B.$$

We denote

$$\eta := 4C_{\mathcal{T}}(C_{\mathcal{T}} + r_x)(r_x/D)^{D-2} \exp\left(-r_x^2/\sigma(t)\right)/(T_0(T - T_0)),$$
$$r_x := \mathcal{O}\left( \sqrt{d_0 \log d_0 + \log C_{\mathcal{T}} + \log(n/\bar{\delta})} \right).$$

Then we have

$$\eta \leq \frac{1}{nT_0(T - T_0)}. \tag{F.14}$$

For any $\bar{\delta} > 0$, according to Lemma F.5, the following holds for term $(I)$ with probability $1 - \bar{\delta}$,

$$(I) = \mathcal{O}\left( \frac{(1 + 6/a)(C_{\mathcal{T}}^2 + r_x^2)}{nT_0(T - T_0)} \log \frac{\mathcal{N}\left( \frac{(T-T_0)(\iota-\eta)}{(C_{\mathcal{T}}+r_x)\log(T/T_0)}, \mathcal{S}_{\mathcal{T}_p^{2,1,4}}, \|\cdot\|_2 \right)}{\bar{\delta}} + (2+a)\iota \right),$$

where $c \leq 0$ is a constant, and $\iota > 0$ will be determined later.

We set

$$\iota := \frac{2}{n^b T_0 (T - T_0)},$$

where $0 < b \le 1$ is a constant to be determined later.

**Remark F.2** (Selection Criteria of $\tau$). We have two criteria:

- Recall that the covering number used in our setting is $\mathcal{N}\left(\frac{(T-T_0)(\iota-\eta)}{(C_T+r_x)\log(T/T_0)}, \mathcal{S}_{\mathcal{T}_p^{2,1,4}}, \|\cdot\|_2\right)$. Thus, we must ensure $\iota \ge \eta$. According to (F.14), we consider $\iota$ satisfying the condition $\iota \ge (nT_0(T - T_0))^{-1}$. Therefore, we consider $0 < b \le 1$.

- For the exponent of $(T - T_0)$, although selecting a value smaller than 1 is possible, we find that the convergence rate with respect to $T$ is dominated by the $1/T$ term appearing later in the second term of (F.18). Therefore, we continue to consider the exponent to be 1.

Then we have

$$(I) = \mathcal{O}\left(\frac{(1 + 6/a)\left(C_T^2 + r_x^2\right)}{nT_0(T - T_0)} \log \frac{\mathcal{N}\left((n^b(C_T + r_x)T_0 \log(T/T_0))^{-1}, \mathcal{S}_{\mathcal{T}_p^{2,1,4}}, \|\cdot\|_2\right)}{\bar{\delta}} + \frac{4 + 2a}{n^b T_0 (T - T_0)}\right),$$

with probability $1 - \bar{\delta}$.

Following the proof structure of term $(II)$ in [Chen et al., 2023, Appendix B.2], we have

$$(II) = \mathcal{O}\left(\frac{1}{T_0} C_T^2 r_x^2 \exp\{-A_2 r_x^2/2\}\right).$$

For any $\epsilon > 0$, let $s_{\overline{W}}$ be the transformer network approximator to the score function in Theorem 3.1. For the term $(III)$, we have

$$(III) \le \underbrace{\widehat{\mathcal{L}}(s_{\overline{W}}) - (1 + a)\mathcal{L}^{\mathrm{trunc}}(s_{\overline{W}})}_{(III)_1} + (1 + a)\underbrace{\mathcal{L}^{\mathrm{trunc}}(s_{\overline{W}})}_{(III)_2}.$$

For any $\bar{\delta} > 0$, according to Lemma F.5 and given that $s_{\overline{W}}$ is a fixed function, the following holds for term $(III)_1$ with probability $1 - \bar{\delta}$,

$$(III)_1 = \mathcal{O}\left(\frac{(1 + 3/a)\left(C_T^2 + r_x^2\right)}{nT_0(T - T_0)} \log \frac{1}{\bar{\delta}}\right).$$

Following the proof structure of term $(III)_2$ in [Chen et al., 2023, Appendix B.2], we have

$$(III)_2 = \mathcal{O}\left(\frac{d\epsilon^2}{T_0(T - T_0)}\right) + C_3,$$

where $C_3$ is a constant.

Putting $(I)$, $(II)$, and $(III)$ together and setting $a = \epsilon^2$, then we have

$$\frac{1}{T - T_0} \int_{T_0}^{T} \left\|s_{\widehat{W}}(\cdot, t) - \nabla \log p_t(\cdot)\right\|_{L^2(P_t)}^2 \mathrm{d}t$$

$$= \mathcal{O}\left(\frac{\left(C_T^2 + r_x^2\right)}{\epsilon^2 n T_0(T - T_0)} \log \frac{\mathcal{N}\left((n^b(C_T + r_x)T_0 \log(T/T_0))^{-1}, \mathcal{S}_{\mathcal{T}_p^{2,1,4}}, \|\cdot\|_2\right)}{\bar{\delta}} + \frac{n^{-b} + d_0\epsilon^2}{T_0(T - T_0)}\right), \quad \text{(F.15)}$$

with probability $1 - 3\bar{\delta}$.

**Covering Number of $\mathcal{S}_{\mathcal{T}_p^{2,1,4}}$.** The next step is to calculate the covering number of $\mathcal{S}_{\mathcal{T}_p^{2,1,4}}$. $\mathcal{S}_{\mathcal{T}_p^{2,1,4}}$ consists of two components: (i) Matrix $W_B$ with orthonormal columns; (ii) Network function $\bar{f}_{\mathcal{T}}$.

Suppose we have $W_{B1}, W_{B2}$ and $f_1, f_2$, such that $\|W_{B1} - W_{B2}\|_F \leq \delta_1$ and $\sup_{\|x\|_2 \leq 3r_x + \sqrt{D \log D}, t \in [T_0, T]} \|f_1(x,t) - f_2(x,t)\|_2 \leq \delta_2$, where $f_1 = R^{-1} \circ f_{\mathcal{T}1} \circ R, f_2 = R^{-1} \circ f_{\mathcal{T}2} \circ R$. Then we have

$$\sup_{\|\bar{x}\|_2 \leq 3r_x + \sqrt{D \log D}, t \in [T_0, T]} \|s_{W_{B1}, f_{\mathcal{T}1}}(\bar{x}, t) - s_{W_{B2}, f_{\mathcal{T}2}}(\bar{x}, t)\|_2$$

$$= \frac{1}{\sigma(t)} \sup_{\|\bar{x}\|_2 \leq 3r_x + \sqrt{D \log D}, t \in [T_0, T]} \left\| W_{B1} f_1(W_{B1}^\top \bar{x}, t) - W_{B2} f_2(W_{B2}^\top \bar{x}, t) \right\|_2$$

$$\leq \frac{1}{\sigma(t)} \sup_{\|\bar{x}\|_2 \leq 3r_x + \sqrt{D \log D}, t \in [T_0, T]} \left( \left\| W_{B1} f_1(W_{B1}^\top \bar{x}, t) - W_{B1} f_1(W_{B2}^\top \bar{x}, t) \right\|_2 \right.$$

$$\left. + \left\| W_{B1} f_1(W_{B2}^\top \bar{x}, t) - W_{B1} f_2(W_{B2}^\top \bar{x}, t) \right\|_2 + \left\| W_{B1} f_2(W_{B2}^\top \bar{x}, t) - W_{B2} f_2(W_{B2}^\top \bar{x}, t) \right\|_2 \right)$$

$$\leq \frac{1}{\sigma(t)} \left( L_{\mathcal{T}} \delta_1 \sqrt{d_0} (3r_x + \sqrt{D \log D}) + \delta_2 + \delta_1 K \right), \tag{F.16}$$

where $L_{\mathcal{T}}$ upper bounds the Lipschitz constant of $f_{\mathcal{T}}$.

For the set $\{W_B \in \mathbb{R}^{D \times d_0} : \|W_B\|_2 \leq 1\}$, its $\delta_1$-covering number is $\left(1 + 2\sqrt{d_0}/\delta_1\right)^{Dd_0}$ [Chen et al., 2020a, Lemma 8]. The $\delta_2$-covering number of $f$ needs further discussion as there is a reshaping process in our network. The input is reshaped from $\bar{h} \in \mathbb{R}^{d_0}$ to $H \in \mathbb{R}^{d \times L}$, and

$$\left\| \bar{h} \right\|_2 \leq r_x \iff \|H\|_F \leq r_x.$$

Thus we have

$$\sup_{\left\| \bar{h} \right\|_2 \leq 3r_x + \sqrt{D \log D}, t \in [T_0, T]} \left\| f_1(\bar{h}, t) - f_2(\bar{h}, t) \right\|_2 \leq \delta_2$$

$$\iff \sup_{\|H\|_F \leq 3r_x + \sqrt{D \log D}, t \in [T_0, T]} \|f_{\mathcal{T}1}(H) - f_{\mathcal{T}2}(H)\|_2 \leq \delta_2.$$

Then we follow the covering number of sequence-to-sequence transformer $\mathcal{T}_p^{2,1,4}$ in Lemma F.6. We get the following $\delta_2$-covering number

$$\frac{\log(nL)}{\delta_2^2} \cdot \left( \sum_{i=1}^{K} \alpha_i^{\frac{2}{3}} \left( d^{\frac{2}{3}} \left( C_F^{2,\infty} \right)^{\frac{4}{3}} + d^{\frac{2}{3}} \left( 2(C_F)^2 C_{OV} C_{KQ}^{2,\infty} \right)^{\frac{2}{3}} + \tau m^{\frac{2}{3}} \left( (C_F)^2 C_{OV}^{2,\infty} \right)^{\frac{2}{3}} \right) \right)^3,$$

where

$$\alpha_i := \prod_{j < i} (C_F)^2 C_{OV} (1 + 4C_{KQ})(C_X + C_E).$$

According to the (F.4), (F.5), (F.7), (F.8), (F.9), (F.10), (F.12), (F.13), (F.11) and (F.6) in Appendix F.1.2, we derive the following with $\delta = \mathcal{O}(\epsilon^{2/d})$ (Appendix E.4) and $d = 4$ (Theorem 3.1):

$$K = \mathcal{O}\left(\epsilon^{-2L}\right), L_{\mathcal{T}} = \mathcal{O}\left(d_0 L_{s_+}\right), C_{OV}^{2,\infty} = \mathcal{O}(d\epsilon^{-4L}), C_{OV} = \mathcal{O}(\epsilon^{-4L}),$$

$$C_{KQ}^{2,\infty} = \mathcal{O}(\epsilon^{-4}), C_{KQ} = \mathcal{O}(\epsilon^{-4}), C_F^{2,\infty} = \mathcal{O}(\epsilon^{-4}), C_F = \mathcal{O}(\epsilon^{-2}), C_E = \mathcal{O}(L^{3/2}), \tag{F.17}$$

$$C_{\mathcal{T}} = \mathcal{O}\left(d_0 L_{s_+} \cdot \sqrt{d_0 \log(d_0/T_0) + \log(1/\epsilon)}\right), r_x = \mathcal{O}\left(\sqrt{d_0 \log d_0 + \log C_{\mathcal{T}} + \log(n/\bar{\delta})}\right).$$

Each element of the input data is within $[0, 1]$, as shown in Appendix E.

For any $\delta_3 > 0$, we get the log-covering number of $\mathcal{T}_p^{2,1,4}$,

$$\log \mathcal{N}\left(\delta_3, \mathcal{T}_p^{2,1,4}, \|\cdot\|_2\right) = \mathcal{O}\left(\frac{\epsilon^{-8K} \cdot L^K d^2 \log(nL)}{\delta_3}\right)$$

$$= \mathcal{O}(1) \cdot \left(\frac{2^{8K \log(L/\epsilon)} d^2 \log(nL)}{\delta_3}\right).$$

According to (F.15), we adopt the following value for $\delta_3$ in our setting

$$\delta_3 = \frac{1}{n^b(C_{\mathcal{T}} + r_x)T_0 \log(T/T_0)}.$$

According to [Chen et al., 2023, Appendix B.2], the log-covering number of $\mathcal{S}_{\mathcal{T}_p^{2,1,4}}$ is

$$\log \mathcal{N}\left(\delta_3, \mathcal{S}_{\mathcal{T}_p^{2,1,4}}, \|\cdot\|_2\right)$$

$$= \mathcal{O}\left(2Dd_0 \cdot \log\left(1 + \frac{6C_{\mathcal{T}} L_{\mathcal{T}} \sqrt{d_0}(3r_x + \sqrt{D \log D})}{T_0 \delta_3}\right) + \frac{2^{8K \log(L/\epsilon)} d^2 \log(nL)}{T_0^2 \delta_3^2}\right)$$
$$\left(\text{By (F.16)}\right)$$

$$= \mathcal{O}\left(n^{2b} 2^{8(1/\epsilon)^L \log(L/\epsilon)} D d^2 d_0^6 L_{s_+}^2 \cdot \log(nL)\right) \qquad\qquad\qquad \left(\text{By (F.17)}\right)$$

$$= \mathcal{O}\left(n^{2b} 2^{(1/\epsilon)^{2L}} D d^2 d_0^6 L_{s_+}^2 \cdot \log(nL)\right) \qquad\qquad \left(\text{By } (1/\epsilon)^L \geq 8 \log(L/\epsilon)\right)$$

$$= \widetilde{\mathcal{O}}\left(n^{2b} 2^{(1/\epsilon)^{2L}} D d^2 d_0^6 L_{s_+}^2\right) \qquad\qquad\qquad \left(\text{By ignoring the log factors}\right)$$

$$= \widetilde{\mathcal{O}}\left(n^{2b} 2^{(1/\epsilon)^{2L}} D d^2 d_0^6 L_{s_+}^2\right).$$

Substituting the log-covering number into (F.15), we have

$$\frac{1}{T - T_0} \int_{T_0}^T \left\|s_{\widehat{W}}(\cdot, t) - \nabla \log p_t(\cdot)\right\|_{L^2(P_t)}^2 \mathrm{d}t$$

$$= \mathcal{O}\left(\frac{C_{\mathcal{T}}^2 + r_x^2}{\epsilon^2 n T_0 (T - T_0)}\left(\log \mathcal{N}(\delta_3, \mathcal{S}_{\mathcal{T}_p^{2,1,4}}, \|\cdot\|_2) + \log(1/\bar{\delta})\right) + \frac{1}{n^b T_0 (T - T_0)} + \frac{d_0}{T_0(T - T_0)}\epsilon^2\right)$$

$$= \mathcal{O}\left(\underbrace{\frac{C_{\mathcal{T}}^2 + r_x^2}{\epsilon^2 n T_0 T}\left(\log \mathcal{N}(\delta_3, \mathcal{S}_{\mathcal{T}_p^{2,1,4}}, \|\cdot\|_2) + \log(1/\bar{\delta})\right)}_{\text{1st term}} + \frac{1}{n^b T_0 T} + \underbrace{\frac{d_0}{T_0 T}\epsilon^2}_{\text{2nd term}}\right). \qquad \text{(F.18)}$$

Recall the following parameters:

- $C_{\mathcal{T}}^2 = \mathcal{O}(d_0^2 L_{s_+}^2 d_0 \log(d_0/T_0) + \log(1/\epsilon))$,

- $r_x^2 = \mathcal{O}(d_0 \log d_0 + \log C_{\mathcal{T}} + \log(n/\bar{\delta}))$,

- $\bar{\delta}$: probability error,

- $\epsilon$: approximation error,

- $n$: sample size,

- $T_0 < T/2$,

- $D, d, d_0 > 1$: feature dimension,

- $L > 1$: sequence length,

- $d_0 = L \cdot d$,

- $L_{s_+}$: Lipschitz coefficient.

Ignoring the $\log$ factors and $\mathrm{poly}(D, d, d_0, L_{S_+})$, the first term in (F.18) becomes

$$\frac{1}{n^{1-2b}} \cdot \frac{1}{T_0 T} \cdot 2^{(1/\epsilon)^{2L}}.$$

The second term is simplified to

$$\frac{1}{T_0 T} \epsilon^2.$$

Thus, the final bound is

$$\widetilde{O}\left( \frac{1}{n^{1-2b}} \cdot \frac{1}{T_0 T} \cdot 2^{(1/\epsilon)^{2L}} + \frac{1}{n^b T_0 T} + \frac{1}{T_0 T} \epsilon^2 \right).$$

To balance the first and second terms with respect to $n$, we select $b = 1/3$. Therefore, we give the final bound as

$$\widetilde{O}\left( \frac{1}{n^{1/3}} \cdot \frac{1}{T_0 T} \cdot 2^{(1/\epsilon)^{2L}} + \frac{1}{n^{1/3} T_0 T} + \frac{1}{T_0 T} \epsilon^2 \right).$$

This completes the proof. $\qquad\square$

### F.3 Proof of Corollary 3.2.1

Our proof is built on [Chen et al., 2023, Appendix C]. The main difference between our work and [Chen et al., 2023] is our score estimation error in Theorem 3.2. Consequently, only the subspace error and the total variation distance differ from [Chen et al., 2023, Theorem 3].

First, we introduce the ground truth backward SDE and the learned backward SDE of the latent variable. Recall from (D.2), $y_t$ denotes the backward process. We denote the backward latent variable by $h_t^\leftarrow = B^\top y_t$. Since we write the time index explicitly, we drop the $\bar{y}, \bar{h}$ notation for $t > 0$.

Following [Chen et al., 2023, Appendix C.2], we have the following ground truth backward process

$$\mathrm{d}h_t^\leftarrow = \left[\frac{1}{2}h_t^\leftarrow + \nabla \log p_{T-t}^h(h_t^\leftarrow)\right]\mathrm{d}t + \mathrm{d}\left(B^\top \overline{W}_t\right),$$

where $\overline{W}_t$ denotes the reversed Wiener process (standard Brownian motion) at time $t$ (see Section 2 for more details).

We define $P_{T_0}^h$ as the *ground truth* marginal distribution of $h_{T_0}^\leftarrow$.

For the learned process $\widetilde{y}_t$, we consider $\widetilde{h}_t^\leftarrow = W_B^\top \widetilde{y}_t$. For any orthogonal matrix $U \in \mathbb{R}^{d_0 \times d_0}$, we define the $U$ transformed version of $\widetilde{h}_t^\leftarrow$ as $\widetilde{h}_t^{\leftarrow,U} = U^\top \widetilde{h}_t^\leftarrow$. Then the backward SDE for $\widetilde{h}_t^{\leftarrow,U}$ is

$$\mathrm{d}\widetilde{h}_t^{\leftarrow,U} = \left[\widetilde{h}_t^{\leftarrow,U} + \widetilde{s}_{U,f}^h(\widetilde{h}_t^{\leftarrow,U}, T-t)\right]\mathrm{d}t + \mathrm{d}\left(U^\top W_B^\top \overline{W}_t\right),$$

where

$$\widetilde{s}_{U,f}^h\left(\widetilde{h}_t^{\leftarrow,U}, t\right) := \frac{1}{\sigma(t)}[-\widetilde{h}_t^{\leftarrow,U} + U^\top f(U\widetilde{h}_t^{\leftarrow,U}, t)].$$

We define $\widehat{P}_{T_0}^h$ as the *estimated* marginal distribution of $\widetilde{h}_{T_0}^{\leftarrow,U}$ from above continuous SDE.

The discretized backward SDE of $\widetilde{h}_{T_0}^{\leftarrow,U}$ is

$$\mathrm{d}\widetilde{h}_t^{\Leftarrow,U} = \left[\widetilde{h}_{k\mu}^{\Leftarrow,U} + \widetilde{s}_{U,f}^h(\widetilde{h}_{k\mu}^{\Leftarrow,U}, T-k\mu)\right]\mathrm{d}t + \mathrm{d}\left(U^\top W_B^\top \overline{W}_t\right), t \in [k\mu, (k+1)\mu).$$

We define $\widehat{P}_{T_0}^{h,\mathrm{dis}}$ as the *estimated* marginal distribution of $\widetilde{h}_{T_0}^{\Leftarrow,U}$ from above discrete SDE.

Next, we present the auxiliary theoretical results in Appendix F.3.1 to prepare our main proof of Corollary 3.2.1. Then we give a detailed proof of Corollary 3.2.1 in Appendix F.3.2.

#### F.3.1 Auxiliary Lemmas

Here we include a few auxiliary lemmas from [Chen et al., 2023] without proofs. Recall the definition of Lipschitz norm: for a given function $f$, $\|f(\cdot)\|_{Lip} = \sup_{x \neq y}(\|f(x) - f(y)\|_2 / \|x - y\|_2)$.

**Lemma F.7** (Lemma 3 of [Chen et al., 2023]). Assume that the following holds

$$\mathbb{E}_{h \sim P_h}\|\nabla \log p_h(h)\|_2^2 \leq C_{sh}, \quad \lambda_{\min}\mathbb{E}_{h \sim P_h}[hh^\top] \geq c_0, \quad \mathbb{E}_{h \sim P_h}\|h\|_2^2 \leq C_h,$$

where $\lambda_{\min}$ denotes the smallest eigenvalue. We denote

$$\overline{\mathbb{E}}[\phi(\bar{h}, t)] = \int_{T_0}^T \frac{1}{\sigma^2(t)}\mathbb{E}_{\bar{x} \sim P_t}[\phi(B^\top \bar{x}, t)]dt. \tag{F.19}$$

Let $T_0 \leq \min\{2\log(d_0/C_{sh}), 1, 2\log(c_0), c_0\}$ and $T \geq \max\{2\log(C_h/d_0), 1\}$. Suppose we have

$$\overline{\mathbb{E}}\left\|W_B f(W_B^\top \bar{x}, t) - Bq(B^\top \bar{x}, t)\right\|_2^2 \leq \epsilon. \tag{F.20}$$

Then we have

$$\left\|W_B W_B^\top - BB^\top\right\|_F^2 = \mathcal{O}(\epsilon T_0 / c_0),$$

and there exists an orthogonal matrix $U \in \mathbb{R}^{d_0 \times d_0}$, such that:

$$\bar{\mathbb{E}}\left\|U^\top f(U\bar{h}, t) - q(\bar{h}, t)\right\|_2^2$$
$$= \epsilon \cdot \mathcal{O}\left(1 + \frac{T_0}{c_0}\left[(T - \log T_0)d_0 \cdot \max_t \|f(\cdot, t)\|_{\text{Lip}}^2 + C_{sh}\right] + \frac{\max_t \|f(\cdot, t)\|_{\text{Lip}}^2 \cdot C_h}{c_0}\right).$$

**Lemma F.8** (Lemma 4 of [Chen et al., 2023]). Assume that $P_h$ is sub-Gaussian and that $f(\bar{h}, t)$ and $\nabla \log p_t^h(\bar{h})$ are Lipschitz continuous with respect to $\bar{h}$ and $t$. For any orthogonal matrix $U \in \mathbb{R}^{d_0 \times d_0}$, we define

$$\widetilde{s}_{U,f}^h(\bar{h}, t) := \frac{1}{\sigma(t)}[-\bar{h} + U^\top f(U\bar{h}, t)].$$

Assume that we have the latent score matching error-bound

$$\int_{T_0}^T \mathbb{E}_{\bar{h} \sim P_t^h}\left\|\widetilde{s}_{U,f}^h(\bar{h}, t) - \nabla \log p_t^h(\bar{h})\right\|_2^2 \, dt \le \epsilon_{\text{latent}}(T - T_0),$$

where $\epsilon_{\text{latent}} > 0$. Then we have the following latent distribution estimation error for the continuous backward SDE:

$$\text{TV}\left(P_{T_0}^h, \widehat{P}_{T_0}^h\right) \lesssim \sqrt{\epsilon_{\text{latent}}(T - T_0)} + \sqrt{\text{KL}\left(P_h \| N(0, I_{d_0})\right)} \cdot \exp(-T),$$

where $\widehat{P}_{T_0}^h$ is the marginal distribution of the generated $h_{T_0}$ using the continuous backward SDE. Furthermore, let $\widehat{P}_{T_0}^{h,\text{dis}}$ denote the marginal distribution of the generated $h_{T_0}$ using the discretized backward SDE. Then we have the following latent distribution estimation error for the discretized backward SDE

$$\text{TV}\left(P_{T_0}^h, \widehat{P}_{T_0}^{h,\text{dis}}\right) \lesssim \sqrt{\epsilon_{\text{latent}}(T - T_0)} + \sqrt{\text{KL}\left(P_h \| N(0, I_{d_0})\right)} \cdot \exp(-T) + \sqrt{\epsilon_{\text{dis}}(T - T_0)},$$

where

$$\epsilon_{\text{dis}} = \left(\frac{\max_{\bar{h}} \left\|f(\bar{h}, \cdot)\right\|_{\text{Lip}}}{\sigma(T_0)} + \frac{\max_{\bar{h}, t} \left\|f(\bar{h}, t)\right\|_2}{T_0^2}\right)^2 \eta^2$$
$$+ \left(\frac{\max_t \|f(\cdot, t)\|_{\text{Lip}}}{\sigma(T_0)}\right)^2 \eta^2 \max\left\{\mathbb{E}\|h_0\|^2, d_0\right\} + \eta d_0,$$

and $\eta$ is the step size in the backward process.

**Lemma F.9** (Lemma 6 of [Chen et al., 2023]). Consider the following discretized SDE with step size $\mu$ satisfying $T - T_0 = K_T \mu$ for some $K_T \in \mathbb{N}_+$,

$$dy_t = \left[\frac{1}{2} - \frac{1}{\sigma(T - k\mu)}\right] y_{k\mu} dt + dB_t, \quad \text{for} \quad t \in [k\mu, (k+1)\mu),$$

where $y_0 \sim N(0, I)$. Then, for $T > 1$ and $T_0 + \mu \le 1$, we have $y_{T-T_0} \sim N(0, \sigma^2 I)$ with $\sigma^2 \le e(T_0 + \mu)$.

**Lemma F.10** (Lemma 10 in [Chen et al., 2023]). Assume that $\nabla \log p_h(h)$ is $L_h$-Lipschitz. Then we have $\mathbb{E}_{h \sim P_h} \|\nabla \log p_h(h)\|_2^2 \leq d_0 L_h$.

### F.3.2 Main Proof of Corollary 3.2.1

*Proof.* Recall the estimation error in Theorem 3.2 is $\xi(n, \epsilon, L)/(TT_0)$, where

$$\xi(n, \epsilon, L) := \frac{1}{n^{1/3}} \cdot 2^{(1/\epsilon)^{2L}} + \frac{1}{n^{1/3}} + \epsilon^2.$$

- **Proof of (i).** By the definition of (F.19) and the estimation error in Theorem 3.2, the error bound in (F.20) equals to $\xi(n, \epsilon, L)(T - T_0)/(TT_0)$ in Lemma F.7. By Lemma F.10, we set $C_{sh} = d_0 L_h$. Then, we have

$$\|W_B W_B^\top - BB^\top\|_F^2 = \mathcal{O}\left(\frac{\xi(n, \epsilon, L)}{c_0}\right).$$

By substituting the value of $\xi(n, \epsilon, L)$ and $T = \mathcal{O}(\log n)$ into the bound above, we deduce

$$\|W_B W_B^\top - BB^\top\|_F^2 = \mathcal{O}\left(\frac{1}{c_0 n^{1/3}} 2^{(1/\epsilon)^{2L}} + \frac{1}{c_0 n^{1/3}} + \frac{\epsilon^2}{c_0}\right).$$

- **Proof of (ii).** Recall that $\max_t \|f(\cdot, t)\|_{\text{Lip}} \leq L_\mathcal{T}$. Furthermore, according to Lemma F.7 and Lemma F.10, we have

$$\mathbb{E}\|U^\top f(U\bar{h}, t) - q(\bar{h}, t)\|_2^2 = \mathcal{O}(\epsilon_{\text{latent}}(T - T_0)),$$

where

$$\epsilon_{\text{latent}} = \frac{\xi(n, \epsilon, L)}{TT_0} \cdot \mathcal{O}\left(\frac{T_0}{c_0}\left[(T - \log T_0)d_0 \cdot L_\mathcal{T}^2 + d_0 L_h\right] + \frac{L_\mathcal{T}^2 \cdot C_h}{c_0}\right).$$

Following the proof structure in [Chen et al., 2023, Appendix C.4], we get

$$\mathbb{E}\|U^\top f(U\bar{h}, t) - q(\bar{h}, t)\|_2^2 = \int_{T_0}^{T} \mathbb{E}_{\bar{h} \sim P_t^h} \left\|\frac{U^\top f(U\bar{h}, t) - \bar{h}}{\sigma(t)} - \nabla \log p_t^h(\bar{h})\right\|_2^2 \mathrm{d}t$$
$$\leq \epsilon_{\text{latent}}(T - T_0).$$

Following the proof structure in [Chen et al., 2023, Appendix C.4] and setting $T = \mathcal{O}(\log n)$, we obtain

$$\mathsf{TV}(P_{T_0}^h, \widehat{P}_{T_0}^{h, \text{dis}}) = \widetilde{\mathcal{O}}\left(\sqrt{\epsilon_{\text{latent}}(T - T_0)}\right)$$
$$= \widetilde{\mathcal{O}}\left(\sqrt{\left(\frac{1}{n^{1/3}} 2^{(1/\epsilon)^{2L}} + \frac{1}{n^{1/3}} + \epsilon^2\right) \cdot \log n}\right),$$

where $\widetilde{\mathcal{O}}$ hides the factor about $D, d_0, d, L_{s_+}, \log n$, and $T - T_0$

By definition, $\widehat{P}_{T_0}^{h, \text{dis}} = (UW_B)_\sharp^\top \widehat{P}_{T_0}$, where $\widehat{P}_{T_0}$ is the distribution generated by $s_{\widehat{W}}$ using the discretized backward process. This completes the proof of the total variation distance.

- **Proof of (iii).** We apply Lemma F.9 due to our score decomposition. With the marginal distribution at time $T - T_0$ and observing $\mu \ll T_0$, we obtain the last property.

This completes the proof. □

# G Proofs of Section 4

Our proofs are motivated by the observation of low-rank gradient decomposition in transformer-like models [Alman and Song, 2024b, Gu et al., 2024]. With our simplifications and observations made in Section 4, we utilize the fine-grained complexity results of transformer and attention [Hu et al., 2024b, Alman and Song, 2024a,b] and tensor trick (Lemma D.1 and [Diao et al., 2019, 2018]) to proceed our proofs. Specifically, we approximate DiT training gradients with a series of low-rank approximations in Appendices G.1.1 to G.1.3, and carefully match the multiplication dimensions so that the computation of $\frac{\mathrm{d}g_2}{\mathrm{d}\underline{W}}$ forms a chained low-rank approximation in Appendix G.2.

## G.1 Auxiliary Theoretical Results for Theorem 4.1

Here we present some auxiliary theoretical results to prepare our main proof of the Existence of almost-linear Time Algorithms for ADITGC Theorem 4.1.

### G.1.1 Low-Rank Decomposition of DiT Gradients

We start by some definitions. Recall that $W \in \mathbb{R}^{d \times d}$ and $\underline{W} \in \mathbb{R}^{d^2}$ denotes the vectorization of $W \in \mathbb{R}^{d \times d}$ following Definition D.1.

**Definition G.1.** Let $A_1, A_2 \in \mathbb{R}^{d \times L}$ be two matrices. Suppose $\mathsf{A} = A_1^\top \otimes A_2^\top \in \mathbb{R}^{L^2 \times d^2}$. Define $\mathsf{A}_{j_0} \in \mathbb{R}^{L \times d^2}$ as an $L \times d^2$ sub-block of $\mathsf{A}$. There are $L$ such sub-blocks in total. For each $j_0 \in [L]$, define the function $u(\underline{W})_{j_0} : \mathbb{R}^{d^2} \to \mathbb{R}^L$ by $u(\underline{W})_{j_0} := \exp(\mathsf{A}_{j_0} \underline{W}) \in \mathbb{R}^L$.

**Definition G.2.** Let $A_1, A_2 \in \mathbb{R}^{d \times L}$ be two matrices. Suppose $\mathsf{A} = A_1^\top \otimes A_2^\top \in \mathbb{R}^{L^2 \times d^2}$. Define $\mathsf{A}_{j_0} \in \mathbb{R}^{L \times d^2}$ as an $L \times d^2$ sub-block of $\mathsf{A}$. There are $L$ such sub-blocks in total. For every index $j_0 \in [L]$, consider the function $\alpha(\underline{W})_{j_0} : \mathbb{R}^{d^2} \to \mathbb{R}$ defined by $\alpha(\underline{W})_{j_0} := \langle \underbrace{\exp(\mathsf{A}_{j_0} \underline{W})}_{L \times 1}, \underbrace{\mathbb{1}_L}_{L \times 1} \rangle$.

**Definition G.3.** Suppose that $\alpha(\underline{W})_{j_0} \in \mathbb{R}$ and $u(\underline{W})_{j_0} \in \mathbb{R}^L$ are defined as in Definitions G.1 and G.2, respectively. For a fixed $j_0 \in [L]$, consider the function $f(\underline{W})_{j_0} : \mathbb{R}^{d^2} \to \mathbb{R}^L$ defined by

$$f(\underline{W})_{j_0} := \underbrace{\alpha(\underline{W})_{j_0}^{-1}}_{\text{scalar}} \underbrace{u(\underline{W})_{j_0}}_{L \times 1}.$$

Define $f(\underline{W}) \in \mathbb{R}^{L \times L}$ as the matrix where the $j_0$-th row is $(f(\underline{W})_{j_0})^\top$.

**Definition G.4.** For every $i_0 \in [d]$, define the function $h(\underline{W}_{OV})_{i_0} : \mathbb{R}^{d^2} \to \mathbb{R}^L$ by

$$h(\underline{W}_{OV})_{i_0} := \underbrace{A_3^\top}_{L \times d} \underbrace{(W_{OV}^\top)_{*,i_0}}_{d \times 1}.$$

Here, $W_{OV} \in \mathbb{R}^{d \times d}$ denotes the matrix representation of $\underline{W}_{OV} \in \mathbb{R}^{d^2}$, and $(W_{OV})_{*,i_0}^\top$ represents the $i_0$-th column of $W_{OV}^\top$. Define $h(\underline{W}_{OV}) \in \mathbb{R}^{L \times d}$ as the matrix where the $i_0$-th column is $h(\underline{W}_{OV})_{i_0}$.

**Definition G.5.** For each $j_0 \in [L]$, we denote $f(\underline{W})_{j_0} \in \mathbb{R}^L$ as the normalized vector defined by Definition G.3. For each $i_0 \in [d]$, $h(\underline{W}_{OV})_{i_0}$ is defined as per Definition G.4. For every pair $(j_0, i_0) \in [L] \times [d]$, define the function $c(\underline{W})_{j_0,i_0} : \mathbb{R}^{d^2} \times \mathbb{R}^{d^2} \to \mathbb{R}$ by

$$c(\underline{W})_{j_0,i_0} := \langle f(\underline{W})_{j_0}, h(\underline{W}_{OV})_{i_0} \rangle - Y_{j_0,i_0}^\top,$$

where $(W_{OV})_{j_0,i_0}$ is the element at the $(j_0, i_0)$ position of the matrix $W_{OV} \in \mathbb{R}^{L \times d}$. $c(\cdot)$ has matrix form

$$\underbrace{c(\underline{W})}_{L \times d} = \underbrace{f(\underline{W})}_{L \times L} \underbrace{h(\underline{W}_{OV})}_{L \times d} - \underbrace{Y^\top}_{L \times d}.$$

With the tensor trick (Appendix D.3), we compute the gradient $\frac{\mathrm{d}g_2}{\mathrm{d}\underline{W}}$ of the DiT loss as follows:

$$\frac{\mathrm{d}g_2}{\mathrm{d}\underline{W}} = \frac{\mathrm{d}}{\mathrm{d}\underline{W}} \left[ \frac{1}{2} \sum_{j_0=1}^{L} \sum_{i_0=1}^{d} c_{j_0,i_0}^2(\underline{W}) \right]. \tag{G.1}$$

(G.1) presents a neat decomposition of $\frac{\mathrm{d}g_2}{\mathrm{d}\underline{W}}$. Each term is easy enough to handle. Thus, we arrive at the following lemma. Let $Z[i, \cdot]$ and $Z[\cdot, j]$ be the $i$-th row and $j$-th column of matrix $Z$.

**Lemma G.1** (Low-Rank Decomposition of DiT Gradient). Let matrix $A_1, A_2, A_3, W, W_{OV}, Y$ and loss function $\mathcal{L}$ follow Definition 4.1, and $\mathsf{A} := A_1^\top \otimes A_2^\top$. It holds

$$\frac{\mathrm{d}g_2}{\mathrm{d}\underline{W}} = \sum_{j_0=1}^{L} \sum_{i_0=1}^{d} c(\underline{W})_{j_0,i_0} \, \mathsf{A}_{j_0}^\top \underbrace{\left( \overbrace{\mathrm{diag}\left(f(\underline{W})_j\right)}^{(II)} - \overbrace{f(\underline{W})_{j_0} f(\underline{W})_{j_0}^\top}^{(III)} \right)}_{(I)} h(\underline{W}_{OV})_{i_0}. \tag{G.2}$$

*Proof.* Let $Z[i, \cdot]$ and $Z[\cdot, j]$ be the $i$-th row and $j$-th column of matrix $Z$.

With DiT loss Definition 4.1, we have

$$\frac{\mathrm{d}g_2}{\mathrm{d}\underline{W}} = \frac{1}{2} \sum_{j_0=1}^{L} \sum_{i=1}^{d} \frac{\mathrm{d}}{\mathrm{d}\underline{W}} c_{j_0,i_0}^2(\underline{W})$$

$$= \sum_{j_0=1}^{L} \sum_{i=1}^{d} \frac{\mathrm{d}}{\mathrm{d}\underline{W}} c_{j_0,i_0}^2 c(\underline{W})_{j_0,i_0} \cdot \frac{\mathrm{d}c(\underline{W})_{j_0,i_0}}{\mathrm{d}\underline{W}_{i_0}}$$

$$= \sum_{j_0=1}^{L} \sum_{i=1}^{d} \frac{\mathrm{d}}{\mathrm{d}\underline{W}} c_{j_0,i_0}^2 c(\underline{W})_{j_0,i_0} \cdot \frac{\mathrm{d}\langle f(\underline{W})_{j_0}, h(\underline{W}_{OV})_{i_0}\rangle}{\mathrm{d}\underline{W}_{i_0}} \qquad \text{(By Definition G.5)}$$

$$= \sum_{j_0=1}^{L} \sum_{i=1}^{d} \frac{\mathrm{d}}{\mathrm{d}\underline{W}} c_{j_0,i_0}^2 c(\underline{W})_{j_0,i_0} \cdot \left\langle \frac{\mathrm{d}f(\underline{W})_{j_0}}{\mathrm{d}\underline{W}_i}, h(\underline{W}_{OV})_{i_0} \right\rangle$$

$$= \sum_{j_0=1}^{L} \sum_{i=1}^{d} \frac{\mathrm{d}}{\mathrm{d}\underline{W}} c_{j_0,i_0}^2 c(\underline{W})_{j_0,i_0} \cdot \left\langle \frac{\mathrm{d}\alpha^{-1}(\underline{W})_{j_0} u(\underline{W})_{j_0}}{\mathrm{d}\underline{W}_i}, h(\underline{W}_{OV})_{i_0} \right\rangle \qquad \text{(By Definition G.3)}$$

$$= \sum_{j_0=1}^{L} \sum_{i=1}^{d} \frac{\mathrm{d}}{\mathrm{d}\underline{W}} c_{j_0,i_0}^2 c(\underline{W})_{j_0,i_0} \cdot \left\langle \alpha(\underline{W})_{j_0}^{-1} \cdot \frac{\mathrm{d}u(\underline{W})_{j_0}}{\mathrm{d}\underline{W}_{i_0}} + \frac{\mathrm{d}\alpha(\underline{W})_{j_0}^{-1}}{\mathrm{d}\underline{W}_{i_0}} \cdot u(\underline{W})_{j_0}, h(\underline{W}_{OV})_{i_0} \right\rangle$$

$$= \sum_{j_0=1}^{L} \sum_{i=1}^{d} \frac{\mathrm{d}}{\mathrm{d}\underline{W}} c_{j_0,i_0}^2 c(\underline{W})_{j_0,i_0} \cdot \left\langle \alpha(\underline{W})_{j_0}^{-1} \cdot \frac{\mathrm{d}u(\underline{W})_{j_0}}{\mathrm{d}\underline{W}_{i_0}} - \alpha(\underline{W})_{j_0}^{-2} \frac{\mathrm{d}\alpha(\underline{W})_{j_0}}{\mathrm{d}\underline{W}_{i_0}} \cdot u(\underline{W})_{j_0}, h(\underline{W}_{OV})_{i_0} \right\rangle.$$
$$\text{(By chain rule)}$$

For each $j_0 \in [L]$, we have

$$\frac{\mathrm{d}\left(\mathsf{A}_{j_0} \underline{W}\right)}{\mathrm{d}\underline{W}_{i_0}} = \mathsf{A}_{j_0} \cdot \frac{\mathrm{d}\underline{W}}{\mathrm{d}\underline{W}_{i_0}} = \left(\mathsf{A}_{j_0}\right)[\cdot, i].$$

Therefore, for each $j_0 \in [L]$, we have

$$
\frac{\mathrm{d}u(\underline{W})_{j_0}}{\mathrm{d}\underline{W}_{i_0}} = \frac{\mathrm{d}\exp\left(\mathsf{A}_{j_0}\,\underline{W}\right)}{\mathrm{d}\underline{W}_{i_0}} \qquad\qquad \left(\text{By Definition G.1}\right)
$$

$$
= \exp\left(\mathsf{A}_{j_0}\,\underline{W}\right) \odot \frac{\mathrm{d}\,\mathsf{A}_{j_0}\,\underline{W}}{\mathrm{d}\underline{W}_{i_0}} \qquad \left(\text{By entry-wise product rule}\right)
$$

$$
= \mathsf{A}_{j_0}[\cdot,i] \odot u(\underline{W})_{j_0}. \qquad\qquad \left(\text{By Definition G.1 again}\right)
$$

Similarly,

$$
\frac{\mathrm{d}\alpha(\underline{W})_{j_0}}{\mathrm{d}\underline{W}_{i_0}} = \frac{\mathrm{d}\left\langle u(\underline{W})_{j_0}, \mathbb{1}_L \right\rangle}{\mathrm{d}\underline{W}_{i_0}} \qquad\qquad \left(\text{By Definition G.2}\right)
$$

$$
= \left\langle \mathsf{A}_{j_0}[\cdot,i] \odot u(\underline{W})_{j_0}, \mathbb{1}_L \right\rangle \qquad \left(\text{By entry-wise product rule}\right)
$$

$$
= \left\langle \mathsf{A}_{j_0}[\cdot,i], u(\underline{W})_{j_0} \right\rangle. \qquad\qquad \left(\text{By Definition G.1 again}\right)
$$

Putting all together, we have

$$
\frac{\mathrm{d}g_2(\underline{W})_{j_0,i_0}}{\mathrm{d}\underline{W}_{i_0}}
$$
$$
= \left[ \left\langle h(\underline{W}_{OV})_{i_0}, \mathsf{A}_{j_0}[\cdot,i] \odot f(\underline{W})_{j_0} \right\rangle - \left\langle h(\underline{W}_{OV})_{i_0}, f(\underline{W})_{j_0} \right\rangle \cdot \left\langle \mathsf{A}_{j_0}[\cdot,i], f(\underline{W})_{j_0} \right\rangle \right] \cdot c(\underline{W})_{j_0,i_0},
$$

where

$$
\left\langle h(\underline{W}_{OV})_{i_0}, \mathsf{A}_{j_0}[\cdot,i] \odot f(\underline{W})_{j_0} \right\rangle - \left\langle h(\underline{W}_{OV})_{i_0}, f(\underline{W})_{j_0} \right\rangle \cdot \left\langle \mathsf{A}_{j_0}[\cdot,i], f(\underline{W})_{j_0} \right\rangle
$$
$$
= \mathsf{A}_{j_0}^\top \left( \mathrm{diag}\left(f(\underline{W})_{j_0}\right) - f(\underline{W})_{j_0} f(\underline{W})_{j_0}^\top \right) h(\underline{W}_{OV})_{i_0}.
$$

This completes the proof. $\qquad\qquad\qquad\qquad\qquad\qquad\qquad\qquad\qquad\qquad\qquad\square$

Observe (G.2) carefully. We see that (I) is diagonal and (II) is low-rank. This provides a hint for algorithmic speedup through low-rank approximation: If we approximate the other parts with low-rank approximation and carefully match the multiplication dimensions, we might formulate the computation of $\frac{\mathrm{d}g_2}{\mathrm{d}\underline{W}}$ as a chained low-rank approximation.

Surprisingly, such an approach makes computing (G.2) as fast as in almost-linear time. To proceed, we further decompose (G.2) according to the chain-rule in the next lemma, and then conduct the approximation term-by-term.

To facilitate our proof, it's convenient to introduce the following notations.

**Definition G.6** ($q(\cdot)$)**.** Define $c(\underline{W}) \in \mathbb{R}^{L \times d}$ as specified in Definition G.5 and $h(\underline{W}_{OV}) \in \mathbb{R}^{L \times d}$ as described in Definition G.4. Define $q(\underline{W}) \in \mathbb{R}^{L \times L}$ by

$$
q(\underline{W}) := \underbrace{c(\underline{W})}_{L \times d} \underbrace{h(\underline{W}_{OV})^\top}_{d \times L}.
$$

In addition, $q(\underline{W})_{j_0}^\top$ denotes the $j_0$-th row of $q(\underline{W})$, transposed, making it an $L \times 1$ vector.

**Definition G.7** ($p(\cdot), p_1(\cdot), p_2(\cdot)$)**.** For each index $j_0 \in [L]$, we define $p(\underline{W})_{j_0} \in \mathbb{R}^n$ as follows:

$$
p(\underline{W})_{j_0} := \left( \mathrm{diag}(f(\underline{W})_{j_0}) - f(\underline{W})_{j_0} f(\underline{W})_{j_0}^\top \right) q(\underline{W})_{j_0}.
$$

We define $p(\underline{W}) \in \mathbb{R}^{L \times L}$ such that $p(\underline{W})_{j_0}^{\top}$ forms the $j_0$-th row of $p(\underline{W})$. In addition, for every index $j_0 \in [L]$, we define $p_1(\underline{W})_{j_0}, p_2(\underline{W})_{j_0} \in \mathbb{R}^L$ as

$$p_1(\underline{W})_{j_0} := \operatorname{diag}\left(f(\underline{W})_{j_0}\right) q(\underline{W})_{j_0}, \quad p_2(\underline{W})_{j_0} := f(\underline{W})_{j_0} f(\underline{W})_{j_0}^{\top} q(\underline{W})_{j_0},$$

such that $p(\underline{W}) = p_1(\underline{W}) - p_2(\underline{W})$.

$p(\cdot)$ allows us to express $\frac{\mathrm{d}g_2}{\mathrm{d}\underline{W}}$ in a neat form:

**Lemma G.2.** Define the functions $f(\underline{W}) \in \mathbb{R}^{L \times L}$, $c(\underline{W}) \in \mathbb{R}^{d \times L}$, $h(\underline{W}_{OV}) \in \mathbb{R}^{d \times L}$, $q(\underline{W}) \in \mathbb{R}^{L \times L}$, and $p(\underline{W}) \in \mathbb{R}^{L \times L}$ as specified in Definitions G.3 to G.7, respectively. Let $A_1, A_2 \in \mathbb{R}^{d \times L}$ be two given matrices, and define $\mathsf{A} = A_1^{\top} \otimes A_2^{\top}$. Define $g_2$ according to (O1), and let $g_2(\underline{W})_{j_0, i_0}$ be as described in (G.1). It holds

$$\frac{\mathrm{d}g_2}{\mathrm{d}\underline{W}} = \operatorname{vec}\left(A_1 p(\underline{W}) A_2^{\top}\right). \tag{G.3}$$

*Proof.* By definitions, (G.1) gives

$$\frac{\mathrm{d}(g_2)_{j_0, i_0}}{\mathrm{d}\underline{W}_{i_0}} \tag{G.4}$$
$$= c_{j_0, i_0} \cdot (\underbrace{\langle f(\underline{W})_{j_0} \odot \mathsf{A}_{j_0, i_0}, h(\underline{W}_{OV})_{i_0} \rangle}_{= \mathsf{A}_{j_0, i}^{\top} \operatorname{diag}(f(\underline{W})_{j_0}) h(\underline{W}_{OV})_{i_0}} - \underbrace{\langle f(\underline{W})_{j_0}, h(\underline{W}_{OV})_{i_0} \rangle \cdot \langle f(\underline{W})_{j_0}, \mathsf{A}_{j_0, i_0} \rangle}_{= \mathsf{A}_{j_0, i}^{\top} f(\underline{W})_{j_0} f(\underline{W})_{j_0}^{\top} h(\underline{W}_{OV})_{i_0}}).$$
$$\left(\text{By } \langle a \odot b, c \rangle = a^{\top} \operatorname{diag}(b) c \text{ for } a, b, c \in \mathbb{R}^L\right)$$

Therefore, (G.4) becomes

$$\frac{\mathrm{d}(g_2)_{j_0, i_0}}{\mathrm{d}\underline{W}_{i_0}} = c_{j_0, i_0} \cdot (\mathsf{A}_{j_0, i}^{\top} \operatorname{diag}(f(\underline{W})_{j_0}) h(\underline{W}_{OV})_{i_0} - \mathsf{A}_{j_0, i}^{\top} f(\underline{W})_{j_0} f(\underline{W})_{j_0}^{\top} h(\underline{W}_{OV})_{i_0})$$
$$= c_{j_0, i_0} \cdot \mathsf{A}_{j_0, i}^{\top}(\operatorname{diag}(f(\underline{W})_{j_0}) - f(\underline{W})_{j_0} f(\underline{W})_{j_0}^{\top}) h(\underline{W}_{OV})_{i_0}. \tag{G.5}$$

Then, by definitions of $q(\cdot), p(\cdot)$, we complete the proof. $\qquad\square$

### G.1.2 Low-Rank Approximations of Building Blocks Part I: $f(\cdot), q(\cdot)$, and $c(\cdot)$

The definitions of $p$, $p_1$, $p_2$, and Lemma G.2 show that the DiT training gradient $\frac{\mathrm{d}g_2}{\mathrm{d}\underline{W}}$ involves entry-wise products of $f$, $q$, and $c$. Therefore, if we approximate these with inner-dimension-matched low-rank approximations, computing $\frac{\mathrm{d}g_2}{\mathrm{d}\underline{W}}$ itself becomes a low-rank approximation. In the following sections, we present low-rank approximations for $f$, $q$, and $c$.

**Lemma G.3** (Approximate $f(\cdot)$, Modified from [Alman and Song, 2023]). Let $\Gamma = o(\sqrt{\log L})$ and $k_1 = L^{o(1)}$. Let $A_1, A_2, \in \mathbb{R}^{d \times L}$, $W \in \mathbb{R}^{d \times d}$ and $f(\underline{W}) = D^{-1} \exp(A_1^{\top} \mathbf{X} A_2)$ with $D = \operatorname{diag}\left(\exp\left(A_1^{\top} W A_2\right) \mathbb{1}_L\right)$ follows Definitions G.1 to G.3 and G.5. If $\max\left(\left\|A_1^{\top} W\right\|_{\max} \leq \Gamma, \|A_2\|_{\max}\right) \leq \Gamma$, then there exist two matrices $U_1, V_1 \in \mathbb{R}^{L \times k_1}$ such that $\left\|U_1 V_1^{\top} - f(\underline{W})\right\|_{\max} \leq \epsilon/\operatorname{poly}(L)$. In addition, it takes $L^{1+o(1)}$ time to construct $U_1$ and $V_1$.

*Proof.* By [Alman and Song, 2023, Theorem 3], we complete the proof. $\qquad\square$

**Lemma G.4** (Approximate $c(\cdot)$). Assume all numerical values are in $O(\log L)$ bits. Let $d = O(\log L)$ and $c(\underline{W}) \in \mathbb{R}^{L \times d}$ follows Definition G.5. There exist two matrices $U_1, V_1 \in \mathbb{R}^{L \times k_1}$ such that $\left\|U_1 V_1^{\top} h(W_{OV}) - Y^{\top} - c(\underline{W})\right\|_{\max} \leq \epsilon/\operatorname{poly}(L)$.

*Proof of Lemma G.4.*

$$\left\|U_1 V_1^\top h(W_{OV}) - Y^\top - c(\underline{W})\right\|_{\max} = \left\|U_1 V_1^\top h(W_{OV}) - Y^\top - (f(\underline{W})h(W_{OV}) - Y^\top)\right\|_{\max}$$
$$\text{(By Definition G.5)}$$
$$= \left\|\left[U_1 V_1^\top - f(\underline{W})\right] h(W_{OV})\right\|_{\max}$$
$$\le \epsilon/\text{poly}(L). \qquad \text{(By [Alman and Song, 2023, Theorem 3])}$$

$\square$

**Lemma G.5** (Approximate $q(\cdot)$). Let $k_2 = L^{o(1)}$, $c(\cdot) \in \mathbb{R}^{L \times d}$ follow Definition G.5 and let $q(\underline{W}) := c(\underline{W})h(W_{OV})^\top \in \mathbb{R}^{L \times L}$ (follow Definition G.6). There exist two matrices $U_2, V_2 \in \mathbb{R}^{L \times k_2}$ such that $\left\|U_2 V_2^\top - q(\underline{W})\right\|_{\max} \le \epsilon/\text{poly}(L)$. In addition, it takes $L^{1+o(1)}$ time to construct $U_2, V_2$.

*Proof of Lemma G.5.* Our proof is built on [Alman and Song, 2023, Lemma D.3].

Let $\widetilde{q}(\cdot)$ denote an approximation to $q(\cdot)$.

By Lemma G.4, $U_1 V_1^\top h(W_{OV}) - Y$ approximates $c(\underline{W})$ up to accuracy $\epsilon = 1/\text{poly}(L)$.

Thus, by setting $\widetilde{q}(\underline{W}) = h(W_{OV})\left(U_1 V_1^\top h(W_{OV}) - Y\right)^\top$, we find a low-rank form for $\widetilde{q}(\cdot)$:

$$\widetilde{q}(\underline{W}) = h(W_{OV})\left(h(W_{OV})\right)^\top V_1 U_1^\top - h(W_{OV})Y^\top,$$

such that

$$\|\widetilde{q}(\underline{W}) - q(\underline{W})\|_{\max} = \left\|h(W_{OV})\left(U_1 V_1^\top h(W_{OV}) - Y\right)^\top - h(W_{OV})Y^\top\right\|_{\max}$$
$$\le d\,\|h(W_{OV})\|_{\max}\left\|U_1 V_1^\top h(W_{OV}) - Y - c(\underline{W})\right\|_{\max}$$
$$\le \epsilon/\text{poly}(L).$$

By $k_1, d = L^{o(1)}$, compute $\underbrace{\left(h(W_{OV})\right)^\top}_{d \times L} \underbrace{V_1}_{L \times k_1} \underbrace{U_1^\top}_{k_1 \times L}$ takes only $L^{1+o(1)}$ time. This completes the proof.

$\square$

### G.1.3 Low-Rank Approximations of Building Blocks Part II: $p(\cdot)$

Now, we use the low-rank approximations of $f, q, c$ to construct low-rank approximations for $p_1(\cdot), p_2(\cdot), p(\cdot)$.

**Lemma G.6** (Approximate $p_1(\cdot)$). Let $k_1, k_2 = L^{o(1)}$. Suppose $U_1, V_1 \in \mathbb{R}^{L \times k_1}$ approximates $f(\underline{W}) \in \mathbb{R}^{L \times L}$ such that $\left\|U_1 V_1^\top - f(\underline{W})\right\|_{\max} \le \epsilon/\text{poly}(L)$, and $U_2, V_2 \in \mathbb{R}^{L \times k_2}$ approximates the $q(\underline{W}) \in \mathbb{R}^{L \times L}$ such that $\left\|U_2 V_2^\top - q(\underline{W})\right\|_{\max} \le \epsilon/\text{poly}(L)$. Then there exist two matrices $U_3, V_3 \in \mathbb{R}^{L \times k_3}$ such that $\left\|U_3 V_3^\top - p_1(\underline{W})\right\|_{\max} \le \epsilon/\text{poly}(L)$. In addition, it takes $L^{1+o(1)}$ time to construct $U_3, V_3$.

*Proof of Lemma G.6.* By tensor trick, we construct $U_3, V_3$ as tensor products of $U_1, V_1$ and $U_2, V_2$, respectively, while preserving their low-rank structures. Then, we show the low-rank approximation of $p_1(\cdot)$ with bounded error by Lemma G.3 and Lemma G.5.

Let $\oslash$ be *column-wise* Kronecker product such that $A \oslash B := [A[\cdot, 1] \otimes B[\cdot, 1] \mid \ldots \mid A[\cdot, k_1] \otimes B[\cdot, k_1]] \in \mathbb{R}^{L \times k_1 k_2}$ for $A \in \mathbb{R}^{L \times k_1}, B \in \mathbb{R}^{L \times k_2}$.

Let $\widetilde{f}(\underline{W}) := U_1 V_1^\top$ and $\widetilde{q}(\underline{W}) := U_2 V_2^\top$ denote matrix-multiplication approximations to $f(\underline{W})$ and $q(\underline{W})$, respectively.

For the case of presentation, let $U_3 = \overbrace{U_1}^{L \times k_1} \oslash \overbrace{U_2}^{L \times k_2}$ and $V_3 = \overbrace{V_1}^{L \times k_1} \oslash \overbrace{V_2}^{L \times k_2}$. It holds

$$
\begin{aligned}
&\left\| U_3 V_3^\top - p_1(\underline{W}) \right\|_{\max} \\
&= \left\| U_3 V_3^\top - f(\underline{W}) \odot q(\underline{W}) \right\|_{\max} && \left( \text{By } p_1(\underline{W}) = f(\underline{W}) \odot q(\underline{W}) \right) \\
&= \left\| (U_1 \oslash U_2)(V_1 \oslash V_2)^\top - f(\underline{W}) \odot q(\underline{W}) \right\|_{\max} \\
&= \left\| (U_1 V_1^\top) \odot (U_2 V_2^\top) - f(\underline{W}) \odot q(\underline{W}) \right\|_{\max} \\
&= \| \widetilde{f}(\underline{W}) \odot \widetilde{q}(\underline{W}) - f(\underline{W}) \odot q(\underline{W}) \|_{\max} \\
&\le \underbrace{\| \widetilde{f}(\underline{W}) \odot \widetilde{q}(\underline{W}) - \widetilde{f}(\underline{W}) \odot q(\underline{W}) \|_{\max}}_{\le \epsilon/\mathrm{poly}(L)} + \underbrace{\| \widetilde{f}(\underline{W}) \odot q(\underline{W}) - f(\underline{W}) \odot q(\underline{W}) \|_{\max}}_{\le \epsilon/\mathrm{poly}(L)} \\
&\le \epsilon/\mathrm{poly}(L). && \left( \text{By Lemma G.3 and Lemma G.5} \right)
\end{aligned}
$$

Computationally, by $k_1, k_2 = L^{o(1)}$, computing $U_3$ and $V_3$ takes $L^{1+o(1)}$ time. This completes the proof. □

---

**Lemma G.7** (Approximate $p_2(\cdot)$). Let $k_1, k_2, k_4 = L^{o(1)}$. Let $p_2(\underline{W}) \in \mathbb{R}^{L \times L}$ follow Definition G.7 such that its $j_0$-th column is $p_2(\underline{W})_{j_0} = f(\underline{W})_{j_0} f(\underline{W})_{j_0}^\top q(\underline{W})_{j_0}$ for each $j_0 \in [L]$. Suppose $U_1, V_1 \in \mathbb{R}^{L \times k_1}$ approximates the $f(\mathbf{X})$ such that $\left\| U_1 V_1^\top - f(\underline{W}) \right\|_{\max} \le \epsilon/\mathrm{poly}(L)$, and $U_2, V_2 \in \mathbb{R}^{L \times k_2}$ approximates the $q(\underline{W}) \in \mathbb{R}^{L \times L}$ such that $\left\| U_2 V_2^\top - q(\underline{W}) \right\|_{\max} \le \epsilon/\mathrm{poly}(L)$. Then there exist matrices $U_4, V_4 \in \mathbb{R}^{L \times k_4}$ such that $\left\| U_4 V_4^\top - p_2() \right\|_{\max} \le \epsilon/\mathrm{poly}(L)$. In addition, it takes $L^{1+o(1)}$ time to construct $U_4, V_4$.

---

*Proof of Lemma G.7.* From Definition G.7,

$$
p_2(\underline{W})_{j_0} := \overbrace{f(\underline{W})_{j_0} \underbrace{f(\underline{W})_{j_0}^\top q(\underline{W})_{j_0}}_{(I)}}^{(II)}.
$$

For (I), we show its low-rank approximation by observing the low-rank-preserving property of the multiplication between $f(\cdot)$ and $q(\cdot)$ (from Lemma G.3 and Lemma G.5). For (II), we show its low-rank approximation by the low-rank structure of $f(\cdot)$ and (I).

**Part (I).** We define a function $r(\underline{W}) : \mathbb{R}^{d^2} \to \mathbb{R}^L$ such that the $j_0$-th component $r(\underline{W})_{j_0} := (f(\underline{W})_{j_0})^\top q(\underline{W})_{j_0}$ for all $j_0 \in [L]$. Let $\widetilde{r}(\underline{W})$ denote the approximation of $r(\underline{W})$ via decomposing into $f(\cdot)$ and $q(\cdot)$:

$$
\begin{aligned}
\widetilde{r}(\underline{W})_{j_0} &:= \left\langle \widetilde{f}(\underline{W})_{j_0}, \widetilde{q}(\underline{W})_{j_0} \right\rangle = (U_1 V_1^\top)[j_0, \cdot] \cdot \left[ (U_2 V_2^\top)[j_0, \cdot] \right]^\top \\
&= U_1[j_0, \cdot] \underbrace{V_1^\top}_{k_1 \times L} \underbrace{V_2}_{L \times k_2} (U_2[j_0, \cdot])^\top,
\end{aligned} \tag{G.6}
$$

for all $j_0 \in [L]$. This allows us to write $p_2(\underline{W}) = f(\underline{W}) \mathrm{diag}(r(\underline{W}))$ with $\mathrm{diag}(\widetilde{r}(\underline{W}))$ denoting a diagonal matrix with diagonal entries being components of $\widetilde{r}(\underline{W})$.

**Part (II).** With $r(\cdot)$, we approximate $p_2(\cdot)$ with $\widetilde{p}_2(\underline{W}) = \widetilde{f}(\underline{W}) \mathrm{diag}(\widetilde{r}(\underline{W}))$ as follows.

Since $\widetilde{f}(\underline{W})$ has low rank representation, and $\mathrm{diag}(\widetilde{r}(\underline{W}))$ is a diagonal matrix, $\widetilde{p}_2(\cdot)$ has low-rank representation by definition. Thus, we set $\widetilde{p}_2(\underline{W}) = U_4 V_4^\top$ with $U_4 = U_1$ and $V_4 = \mathrm{diag}(\widetilde{r}(\underline{W})) V_1$. Then, we bound the approximation error

$$
\left\| U_4 V_4^\top - p_2(\underline{W}) \right\|_{\max}
$$

$$= \left\| \widetilde{p}_2(\underline{W}) - p_2(\underline{W}) \right\|_{\max}$$

$$= \max_{j_0 \in [L]} \left\| \widetilde{f}(\underline{W})_{j_0} \widetilde{r}(\underline{W})_{j_0} - f(\underline{W})_{j_0} r(\underline{W})_{j_0} \right\|_{\max}$$

$$\leq \max_{j_0 \in [L]} \left[ \left\| \widetilde{f}(\underline{W})_{j_0} \widetilde{r}(\underline{W})_{j_0} - f(\underline{W})_{j_0} r(\underline{W})_{j_0} \right\|_{\max} + \left\| \widetilde{f}(\underline{W})_{j_0} \widetilde{r}(\underline{W})_{j_0} - f(\underline{W})_{j_0} r(\underline{W})_{j_0} \right\|_{\max} \right]$$

$$\qquad\qquad\qquad\qquad\qquad\qquad\qquad\qquad\qquad\qquad\qquad\qquad \text{(By triangle inequality)}$$

$$\leq \epsilon/\mathrm{poly}(L).$$

Computationally, computing $V_1^\top V_2$ takes $L^{1+o(1)}$ time by $k_1, k_2 = L^{o(1)}$. Once we have $V_1^\top V_2$ precomputed, (G.6) only takes $O(k_1 k_2)$ time for each $j_0 \in [L]$. Thus, the total time is $O\left(L k_1 k_2\right) = L^{1+o(1)}$. Since $U_1$ and $V_1$ takes $L^{1+o(1)}$ time to construct and $V_4 = \underbrace{\mathrm{diag}(\widetilde{r}(\underline{W}))}_{L \times L} \underbrace{V_1}_{L \times k_1}$ also takes $L^{1+o(1)}$ time, $U_4$ and $V_4$ takes $L^{1+o(1)}$ time to construct. This completes the proof. $\qquad\square$

### G.2 Proof of Theorem 4.1

*Proof of Theorem 4.1.* By the definitions of matrices $p(\cdot)$, $p_1(\cdot)$ and $p_2(\cdot)$ (Definition G.7), we have

$$p(\underline{W}) = p_1(\underline{W}) - p_2(\underline{W}).$$

By Lemma G.2, we have

$$\frac{\mathrm{d}g_2}{\mathrm{d}\underline{W}} = \mathrm{vec}\left(A_1 p(\underline{W}) A_2^\top\right). \tag{G.7}$$

To show the existence of $L^{1+o(1)}$ algorithms for DiT backward computation Problem 1, we prove fast low-rank approximations for $A_1 p_1(\underline{W}) A_2^\top$ and $A_1 p_2(\underline{W}) A_2^\top$ as follows.

Let $\widetilde{p}_1(\underline{W}), \widetilde{p}_2(\underline{W})$ denote the approximations to $p_1(\underline{W}), p_2(\underline{W})$, respectively.

By Lemma G.6, it takes $L^{1+o(1)}$ time to construct $U_3, V_3 \in \mathbb{R}^{L \times k_3}$ such that

$$A_1 \widetilde{p}_1(\underline{W}) A_2^\top = A_1 U_3 V_3^\top A_2^\top.$$

Then, computing $\underbrace{A_1}_{d \times L} \underbrace{U_3}_{L \times k_3} \underbrace{V_3^\top}_{k_3 \times L} \underbrace{A_2^\top}_{L \times d}$ takes $L^{1+o(1)}$ due to the fact that $d, k_1 k_3 = L^{o(1)}$.

Therefore, total running time for $A_1 p_1(\underline{W}) A_2^\top$ is $L \cdot L^{o(1)} = L^{1+o(1)}$.

For the same reason (by Lemma G.7), total running time for $A_1 p_2(\underline{W}) A_2^\top$ is $L \cdot L^{o(1)} = L^{1+o(1)}$.

Lastly, we have

$$\left\| \frac{\partial g_2}{\partial \underline{W}} - \widetilde{G}^{(W)} \right\|_{\max}$$

$$= \left\| \mathrm{vec}\left(A_1 \widetilde{p}(\underline{W}) A_2^\top\right) - \mathrm{vec}\left(A_1 \widetilde{p}(\underline{W}) A_2^\top\right) \right\|_{\max} \qquad\qquad \text{(By Lemma G.2)}$$

$$= \left\| \left(A_1 \widetilde{p}(\underline{W}) A_2^\top\right) - \left(A_1 \widetilde{p}(\underline{W}) A_2^\top\right) \right\|_{\max} \quad \text{(By definition, } \|A\|_{\max} := \max_{i,j} |A_{ij}| \text{ for any matrix } A\text{)}$$

$$\leq \left\| \left(A_1 \left[p_1(\underline{W}) - \widetilde{p}_1(\underline{W})\right] A_2^\top\right) \right\|_{\max} + \left\| \left(A_1 \left[p_2(\underline{W}) - \widetilde{p}_2(\underline{W})\right] A_2^\top\right) \right\|_{\max}$$

$$\qquad\qquad\qquad\qquad\qquad\qquad\qquad\qquad\qquad\qquad \text{(By Definition G.7 and triangle inequality)}$$

$$\leq \|A_1\|_\infty \|A_2\|_\infty \left( \left\| (p_1(\underline{W}) - \widetilde{p}_1(\underline{W})) \right\|_{\max} + \left\| (p_2(\underline{W}) - \widetilde{p}_2(\underline{W})) \right\|_{\max} \right)$$

$$\qquad\qquad\qquad\qquad\qquad\qquad\qquad\qquad\qquad \text{(By the sub-multiplicative property of } \|\cdot\|_\infty)$$

$$\leq \epsilon/\mathrm{poly}(L). \qquad\qquad\qquad\qquad\qquad\qquad\qquad\qquad \text{(By Lemma G.6 and Lemma G.7)}$$

Set $\epsilon = 1/\mathrm{poly}(L)$. We complete the proof. $\qquad\square$

