# OpenReview forum: "On Statistical Rates and  Provably Efficient Criteria of Latent Diffusion Transformers (DiTs)"
_NeurIPS.cc/2024/Conference — NeurIPS 2024 poster_

### Official Review · Reviewer_Zkkk · 2024-07-13

**Soundness:** 3
**Presentation:** 2
**Contribution:** 3
**Rating:** 6
**Confidence:** 1

**Summary:**

In this paper, the authors explore the statistical and computational limits of latent Diffusion Transformers (DiTs) under the assumption of a low-dimensional linear latent space. Their contributions include an approximation error bound for the DiTs score function, which is sub-linear in the latent space dimension, as well as a sample complexity bound demonstrating convergence of the data distribution generated from the estimated score function. Additionally, they identify efficient criteria for forward inference and backward computation, achieving almost-linear time complexity for both processes.

**Strengths:**

- The authors derived approximation error bounds for transformer-based score estimators in latent DiTs, providing practical structural guidance.
- The paper also provided sample complexity bounds for score estimation and demonstrated recovery of initial data distribution.
- The authors provided efficiency criteria and characterized efficient algorithms for latent DiTs, including almost-linear time algorithms for both forward inference and training.

**Weaknesses:**

- I have reviewed Appendix C. How does the proof for DiTs in this paper differ from that presented in [Chen et al., 2023a]? The assumption of a low-dimensional linear latent space in the context of DiTs is somewhat unclear to me. Is this assumption widely accepted within the DiTs community, or has it been adopted primarily to simplify technical analysis and proofs? Additionally, is there potential for the proof to be generalized beyond this assumption? My concerns are further reinforced by the nontrivial gap suggested in Corollary 3.1.1.

**Questions:**

See above.

**Limitations:**

See above.

---

> ### Author Rebuttal · Authors · 2024-08-01
>
> > **Reviewer's Question 1:** I have reviewed Appendix C. How does the proof for DiTs in this paper differ from that presented in [Chen et al., 2023a]?
>
> **Response:**
> Thanks for the question. Here are some clarifications.
>
> * To prove **score approximation** (Theorem 3.1), our approach utilizes the universal approximation of the Transformer network, while the work [Chen23] relies on the approximation capabilities of the ReLU neural network.
> * To prove **score estimation** (Corollary 3.1.1) and distribution estimation (Corollary 3.1.2), our work uses the covering number of the Transformer network, while the work [Chen23] uses the covering number of the ReLU neural network.
> * Our work provides the analysis of the **computational limits** for all possible efficient DiT algorithms/methods for both forward inference and backward training under the strong exponential time hypothesis, while the work [Chen23] does not consider the computational limit analysis.
>
> > **Reviewer's Question 2:** The assumption of a low-dimensional linear latent space in the context of DiTs is somewhat unclear to me. Is this assumption widely accepted within the DiTs community, or has it been adopted primarily to simplify technical analysis and proofs?
>
> **Response:**
> Thank you for your question. Let us provide some additional details.
>
> The assumption is widely accepted within the DiTs community. In practice, existing works [Peebles23, Ma24] use the DiTs with an autoencoder to compress input data into a low-dimensional latent space, where the autoencoder can be a simple linear layer. This aligns with the low-dimensional linear latent space assumption.
>
> > **Reviewer's Question 3:** Additionally, is there potential for the proof to be generalized beyond this assumption? My concerns are further reinforced by the nontrivial gap suggested in Corollary 3.1.1.
>
> **Response:**
> Grateful for your question. Here's some further explanation.
>
> Yes. We can generalize our proof beyond the low-dimensional linear latent space assumption by setting the matrix $B$ in Assumption 2.1 as an identity matrix. However, the linear subspace assumption leads to a more robust conclusion, suggesting that the latent DiTs have the potential to bypass the challenges associated with the high dimensionality of initial data.
>
> ---
>
> We hope these points address the reviewer's questions.
>
> We're open to any further questions or clarifications you might have about our work. Thank you!
>
> ===
>
> * [Chen23] Minshuo Chen, Kaixuan Huang, Tuo Zhao, and Mengdi Wang. Score approximation, estimation and distribution recovery of diffusion models on low-dimensional data. In International Conference on Machine Learning (ICML), 2023.
> * [Peebles23] William Peebles and Saining Xie. Scalable diffusion models with transformers. In Proceedings of the IEEE/CVF International Conference on Computer Vision (ICCV), 2023.
> * [Ma24] Xin Ma, Yaohui Wang, Gengyun Jia, Xinyuan Chen, Ziwei Liu, Yuan-Fang Li, Cunjian Chen, and Yu Qiao. Latte: Latent diffusion transformer for video generation. arXiv preprint arXiv:2401.03048, 2024.

---

### Official Review · Reviewer_cP7q · 2024-07-15

**Soundness:** 4
**Presentation:** 1
**Contribution:** 3
**Rating:** 6
**Confidence:** 1

**Summary:**

This paper explores the statistical and computational limits of latent DiTs:

1. It proves that Transformers are sufficient as universal approximators for the score function in DiTs, with their approximation capacity depending on the latent dimension.
2. Transformer-based score estimators converge to the true score function, indicating that the Transformer architecture is adequate for estimating the original data distribution.
3. It provides provably efficient criteria to demonstrate the existence of almost-linear time algorithms for forward inference and backward computation, offering a theoretical basis for the efficient training and inference of DiTs.

**Strengths:**

The advantages of the paper are as follows:
1. This paper derives an approximation error bound for the score network that is sub-linear in the latent space dimension. This finding not only explains the expressiveness of latent DiTs (under mild assumptions) but also offers guidance for the structural configuration of the score network in practical implementations.
2. This paper prove that learned score estimator is able to recover the initial data distribution, which provides a theoretical basis for the feasibility of using neural networks to estimate the score.
3. This paper proves the existence of almost-linear time DiT training algorithms for forward inference and backward computation, providing a theoretical foundation for efficient training and inference.
4. All statistical and computational results are analyzed in a low-dimensional subspace, demonstrating the feasibility of using VAE for dimensionality reduction in latent DiTs.

**Weaknesses:**

The weaknesses of the paper are as follows:
1. It is meaningful to further elucidate why existing DiTs models are difficult to train.
2. How will these theoretical proofs contribute to the exploration of fast training and sampling algorithms?

**Questions:**

see weaknesses.

**Limitations:**

The paper has outlined its limitations and broader impact.

---

> ### Author Rebuttal · Authors · 2024-08-01
>
> > **Reviewer's Comment 1:** It is meaningful to further elucidate why existing DiTs models are difficult to train.
>
> **Response:**
> Thanks for your comment. We'd like to clarify a few points.
>
> This relates to the high-dimensional latent data representation, which increases both approximation and estimation errors. To demonstrate this, we can generalize our proof to the setting without the low-dimensional linear latent space assumption by using an identity matrix for matrix $B$ in Assumption 2.1, implying a large $d_0$. According to Theorem 3.1, and the proof details of Corollaries 3.1.1 and 3.1.2 (page 48 and 52), we see that the errors in score approximation, score estimation, and distribution estimation all depend on $d_0$. A larger $d_0$ leads to greater errors.
>
> > **Reviewer's Question 1:** How will these theoretical proofs contribute to the exploration of fast training and sampling algorithms?
>
> **Response:**
> Thanks for the question.
>
> In essence, our hardness results provides necessary conditions for designing efficient methods:
> * The latent dim should be small enough $d=O(\log L)$ (Thm 4.1 & Prop 4.1, 4.2)
> * Normalization of $K,Q,V$ in DiT attention heads is beneficial for performance and efficiency. For example:
>   * For efficiency inference: $\max{\|| W_KA_1\||,\||W_QA_2\||,\||W_{OV}A_3\|| }\le B$ with $B=o(\sqrt{\log L})$ (Prop 4.2)
>   * For efficiency training: $\max{\||W_KA_1\||,\||W_QA_2\||,\||W_{OV}A_3\|| }\le \Gamma$ with $\Gamma=o(\sqrt{\log L})$ (Thm 4.1)
>
> We want to emphasize that these conditions are necessary but not sufficient. Sufficient conditions should depend on the detailed designs of specific methods.
>
> We hope these points address the reviewer's concerns. We have revised the latest version of our paper accordingly.
>
> We're open to any further questions or clarifications you might have about our work. Thank you!

---

> ### Comment · Reviewer_cP7q · 2024-08-13
>
> Thank you for your response.
>
>  In summary, the paper is overly theoretical, and I would appreciate it if it could include some toy examples. For instance, it could demonstrate improvements in DiTs by adhering to the design principles for d and QKV  you mentioned in R2, and show a certain performance boost as a result. I would be glad to see such experiments incorporated into the paper.
>
> Considering the limited evaluation as a criterion for scoring, I will deduct one point. I look forward to further discussions that can alleviate my concerns.

---

> > ### Author Response · Authors · 2024-08-13
> > **Why no experiment? It's uncommon to companion computational hardness results with experiments**
> >
> > Thanks for your feedback.
> >
> > We’d like to remind the reviewer that computational hardness (provably criteria) is “there exist” types of results. It is widely accepted that such results do not require and are not meaningful to be supported by specific experiments. For reasons:
> >
> > * **General Applicability:** These results are designed to be general and widely applicable, not specific to particular datasets or experimental setups.
> > * **Purpose:** The purpose of universality/hardness results is to show the limits of what is feasible. This makes any empirical experiment vacuous, incomplete, and hence unnecessary to establish them.
> >
> > Please refer to standard ML/TCS material for more details, for example, [1] from CMU for the nature of such fundamental limits and why they make empirical validation redundant.
> >
> > [1] Toolkit, A. Theorist’S. "Lecture 24: Hardness Assumptions." (2013).
> >
> > ---
> >
> > ### **Experiment Sketch**
> >
> > If you would like a toy example, we can sketch 2 well-known methods that meet our efficiency criteria (assuming you are referring to Prop 4.1) when norm-bounded conditions are satisfied:
> >
> > * DiTs using alternative attention like Performer (random feature transformer) [Choromanski20] can achieve subquadratic time computation under norm-bounded conditions.
> > * DiTs using alternative attention like linear attention [Katharopoulos20] can also achieve subquadratic time computation under norm-bounded conditions.
> >
> > From these examples, it can be easily observed that:
> >
> > * A large temperature parameter $\beta$ will hurt performance, even though they are efficient. Note that a large $\beta$ corresponds to the low temperature region of attention, which is generally known to perform poorly in practice. Thus, efficiency is maintained, but performance suffers when norm-bounded conditions are violated.
> >
> > This aligns with our theory because $\beta$ scales the norms of $K$, $Q$, and $V$ beyond our norm-bounded conditions.
> >
> > Although not necessary given their self-explanatory nature, we can still include these toy experiments in the final version for completeness. We hope this clarifies everything. Thank you for your time!
> >
> > ---
> >
> > [Choromanski20] Choromanski, Krzysztof, et al. "Rethinking attention with performers." arXiv preprint arXiv:2009.14794 (2020).
> >
> > [Katharopoulos20] Katharopoulos, Angelos, et al. "Transformers are rnns: Fast autoregressive transformers with linear attention." ICML, 2020.

---

> > > ### Comment · Reviewer_cP7q · 2024-08-13
> > >
> > > Thank you for your response. I now understand. I believe illustrating your points with toy examples would be beneficial for non-mathematician researchers to gain a deeper understanding.
> > > So, I will maintain my original score.

---

> > > > ### Author Response · Authors · 2024-08-13
> > > >
> > > > Thanks for your feedback and for maintaining your original score.
> > > >
> > > > We are conducting the experiments now and will try to provide the results before the rebuttal deadline.
> > > >
> > > > Best,
> > > >
> > > > Authors

---

> > > > > ### Comment · Reviewer_cP7q · 2024-08-14
> > > > >
> > > > > Looking forward to your next version.

---

> > > > > > ### Author Response · Authors · 2024-08-14
> > > > > >
> > > > > > Dear Reviewer cP7q:
> > > > > >
> > > > > > Thanks for your attention.
> > > > > >
> > > > > > We have successfully completed the development of the necessary code for our study. Currently, the code is in the execution phase. We will add the results to the next version of our manuscript.
> > > > > >
> > > > > > We appreciate your time and understanding!
> > > > > >
> > > > > > Best,
> > > > > >
> > > > > > Authors

---

### Official Review · Reviewer_jLPK · 2024-07-19

**Soundness:** 3
**Presentation:** 1
**Contribution:** 2
**Rating:** 6
**Confidence:** 2

**Summary:**

The paper studies the statistical and computational limits of latent diffusion transformers.

**Strengths:**

The results seem to be new and non-trivial (though I'm not an expert in the field, so I might have a wrong impression).

**Weaknesses:**

The paper is too technical (e.g. there are many long formal definitions), and the results are hard to understand for a non-expert in the area. The formulations of the theorems contain a lot of parameters, it makes them very hard to read. I suggest that you may write down informal versions that would be clear and would illustrate the result (with a reference to the formal version in the appendix).

In addition, while the comparison with prior works is present, it is unclear to me and seems to be imprecise. It would be nice to see something like (for example) "we have this error bound, while all prior works [..., ...] have asymptotically worse errors".

UPDATE: Increased the score and the confidence after the rebuttal.

**Questions:**

Could you write here some informal versions of the theorems (or some simple corollaries) and the clear comparison with prior works? I'll be happy to increase the score if I see that the results are better than the state of the art.

**Limitations:**

The authors adequately addressed the limitations.

---

> ### Author Rebuttal · Authors · 2024-08-01
>
> > **Reviewer's Comment 1:** The paper is too technical (e.g. there are many long formal definitions), and the results are hard to understand for a non-expert in the area. The formulations of the theorems contain a lot of parameters, it makes them very hard to read. I suggest that you may write down informal versions that would be clear and would illustrate the result (with a reference to the formal version in the appendix).
>
> >  **Reviewer's Question 1:** Could you write here some informal versions of the theorems (or some simple corollaries) and the clear comparison with prior works? I'll be happy to increase the score if I see that the results are better than the state of the art.
>
> **Response:**
> Thanks for the suggestion. **We agree adding an informal version of our results will be beneficial for the general audience. We have added them in introduction accordingly**, which are quoted as follows:
>
> * > **Theorem 1.1 (Informal Version of Theorem 3.1).** For any approximation error $\epsilon>0$ and any initial data distribution $P_0$ under Assumptions 2.1 to 2.3, there exists a DiT score network such that for any time $t\in [T_0,T]$ ($T$ is the stopping time in forward process, and $T_0$ is the early stopping time in backward process), the upper bound of the approximation error is
> $\epsilon \cdot \sqrt{d_0}/\sigma(t)$,
> where $\sigma(t)=1-e^{-t}$.
>
> * > **Corollary 1.1.1 (Informal Version of Corollary 3.1.1).** Under Assumptions 2.1 to 2.3 and using $\epsilon \in (0,1)$, we choose the characterized structural configuration of DiT score network in the proof details of Theorem 1.1. Let n denote the sample size, then with probability $1-1/\mathrm{poly}(n)$, the sample complexity for the score estimation is
> $\mathcal{O}(\xi(n, \epsilon))$,
> where $\xi(n, \epsilon)=C_1 \cdot 2^{\epsilon^{-C_2}} \cdot n^{-0.5}+C_3 \cdot \epsilon^2+n^{-1}$,  and $C_1, C_2, C_3$ are positive constants.
>
> * > **Corollary 1.1.2 (Informal Version of Corollary 3.1.2).** With the estimated DiT score network in Corollary 1.1.1, we denote $\hat{P}_{T_0}$ as the generated distribution at time $T_0$. We have the following with probability $1-1/\mathrm{poly}(n)$.
> > 1. The distribution estimation error of $\hat{P}_{T_0}$ in the latent subspace (see Assumption 2.1) is $\mathcal{O}(\sqrt{\xi(n, \epsilon)})$.
> > 2. The estimated distribution $\hat{P}_{T_0}$ in the orthogonal subspace degenerates to a point mass at origin as $T_0 \rightarrow 0$.
>
> * > **Theorem 1.2 (Informal Version of Theorem 4.1).** Assuming SETH and all numerical values are in $O(\log L)$ encoding, there exists an  algorithm for approximating gradient computation of optimizing DiT loss up to $1/\mathrm{poly}(L)$ accuracy that runs in $L^{1+o(1)}$ time, if certain norm bound conditions are satisfied.
>
> * > **Proposition 1.1 (Informal Version of Proposition 4.1).** Assuming SETH, the existence of sub-quadratic time algorithm for approximating DiT inference depends on the norm bounds of $K,Q,V$ of the attention heads in DiT.
>
> * > **Proposition 1.2 (Informal Version of Proposition 4.2).** Assuming SETH, $L^{1+o(1)}$ time DiT inference is possible.
>
> ---
>
> > **Reviewer's Comment 2:** In addition, while the comparison with prior works is present, it is unclear to me and seems to be imprecise. It would be nice to see something like (for example) "we have this error bound, while all prior works [..., ...] have asymptotically worse errors".
>
>
> **Response:**
> Thanks for the comment. We acknowledge the current draft is not precise enough and have made modifications accordingly. We quote them as follows:
>
> * `line 179` After Theorem 3.1:
> > **Remark 3.1 (Comparing with Existing Works.)** We are the first to prove the score approximation capability of DiT, while the prior theoretical works about DiT [Benton24, Wibisono24] merely assume that the score function is well approximated.
> * `line 200` After Corollary 3.1.1:
> > **Remark 3.2 (Comparing with Existing Works.)** We are the first to provide a sample complexity for DiT, while the prior theoretical works on the sample complexity of diffusion models [Zhu23, Chen23] only focus on ReLU-based diffusion models and does not include the attention mechanism.
> * `line 223` After Corollary 3.1.2:
> > **Remark 3.5 (Comparing with Existing Works.)** We are the first to provide a distribution estimation for DiT, incorporating the tail behavior assumption of the latent variable distribution (Assumption 2.2). However, the prior work [Oko23] does not address DiT and relies on assumptions about the initial data distribution that are far from empirical realities.
>
> **These are all "1st known" DiT results (i.e., SOTA?).**
>
> For computational limits, since we are the first analysis on DiT, there is no prior work to compare with.
>
>
>
> ---
> We hope the revisions and clarifications provided in this response address the reviewer's concerns.
>
> We look forward to further feedback and discussion. Thank you for your time and effort!
>
> ===
>
> * [Benton24] Joe Benton, Valentin De Bortoli, Arnaud Doucet, and George Deligiannidis. Nearly d-linear convergence bounds for diffusion models via stochastic localization. In The Twelfth International Conference on Learning Representations (ICLR), 2024.
> * [Wibisono24] Andre Wibisono, Yihong Wu, and Kaylee Yingxi Yang. Optimal score estimation via empirical bayes smoothing. arXiv preprint arXiv:2402.07747, 2024.
> * [Zhu23] Zhenyu Zhu, Francesco Locatello, and Volkan Cevher. Sample complexity bounds for score-matching: Causal discovery and generative modeling. Advances in Neural Information Processing Systems (NeurIPS), 2023.
> * [Chen23] Minshuo Chen, Kaixuan Huang, Tuo Zhao, and Mengdi Wang. Score approximation, estimation and distribution recovery of diffusion models on low-dimensional data. In International Conference on Machine Learning (ICML), 2023.
> * [Oko23] Kazusato Oko, Shunta Akiyama, and Taiji Suzuki. Diffusion models are minimax optimal distribution estimators. In International Conference on Machine Learning (ICML), 2023.

---

> ### Author Response · Authors · 2024-08-13
>
> Dear Reviewer jLPK,
>
> As the discussion period coming to its end, we want to check if our rebuttal has addressed your concerns.
>
> Please let us know if you have any further questions or need clarification. Thank you!
>
> Best regards,
>
> Authors

---

### Official Review · Reviewer_LxhB · 2024-07-27

**Soundness:** 3
**Presentation:** 3
**Contribution:** 3
**Rating:** 7
**Confidence:** 3

**Summary:**

This paper establishes the statistical rates and provably efficient criteria of Latent Diffusion Transformers (DiTs). Specifically, there are three main theoretical results:
* the approximation error bound for the transformer-based score estimator,
* the sample complexity bound for score estimation,
* and the provably efficient criteria for latent DiTs in both forward inference and backward training.

These results closely rely on the low-dimensional linear subspace assumption on input data.

**Strengths:**

1. Given the popularity of generative AI these days, the theoretical understanding of DiTs are of great interest to the community.
2. Most of the concepts and the results are clearly presented.

**Weaknesses:**

1. The practical insights of the provably efficient criteria (Section 4) are relatively scarce.
2. A few noticeable typos were encountered in the paper. For example,
* At line 120 on page 3, 'knonw' should be corrected to 'known' in the sentence 'This is also knonw as ...';
* At line 242 on page 7, there is an extra 'full'.
3. Including simulation results on error bounds against related parameters could enhance the credibility and reliability of theoretical results.

**Questions:**

1. Can the linear subspace assumption on input data be relaxed to a more general one, such as the manifold data? Since it's more natural to consider the intrinsic geometric structures of data.
2. One of the backgrounds of this work comes from the quadratic complexity of transformer blocks with respect to sequence length. While in literature, many alternative attention mechanisms that have a linear complexity in sequence length have been proposed (though not in the context of diffusion). I'm wondering whether it is possible to evaluate these methods' efficiency in your framework.
3. In the right-hand side of equation (3.1), it seems that the second term does not involve $n$ and thus will not goes to zero even when $n$ goes to infinity. Does it mean that the estimation can never be consistent?

**Limitations:**

The limitations of this work have been stated in Section 5 and are left for future work as claimed by the authors.

---

> ### Author Rebuttal · Authors · 2024-08-01
>
> > **Reviewer's Question 1:** Can the linear subspace assumption on input data be relaxed to a more general one, such as the manifold data? Since it's more natural to consider the intrinsic geometric structures of data.
>
> **Response:**
> Thanks for the question. Yes, but not trivial. Here are some clarifications:
> * If we consider that the initial data is manifold without the subspace assumption, we can relax our proof to the manifold data by using an identity matrix for matrix $B$ in Assumption 2.1.
> * If we consider that the low-dimensional latent representation is manifold with the subspace assumption, the price to pay is that we no longer have the score function decomposition (Lemma 2.1). We need new techniques to characterize the rates.
>
> In addition, the linear subspace assumption yields stronger results. The [Chen23] with the subspace assumption obtained a sharper rate than that of [Oko23] on both score estimation and distribution estimation.
>
> > **Reviewer's Question 2:** One of the backgrounds of this work comes from the quadratic complexity of transformer blocks with respect to sequence length. While in literature, many alternative attention mechanisms that have a linear complexity in sequence length have been proposed (though not in the context of diffusion). I'm wondering whether it is possible to evaluate these methods' efficiency in your framework.
>
> **Response:**
> Thanks for the question. We would like to clarify a few points.
>
> Yes.
>
> For the statistical limits (approximation and estimation theory), we can generalize our proof by deriving the universal approximation capability and the covering number of the attention mechanism with linear complexity.
>
> For the computational limits, these alternative attention variants (efficient or not) have already fallen into our framework. This is because we consider “all possible” algorithms in **Problem 1**. That is, any attempt to compute the DiT gradient faster is considered in our analysis. This includes using different attention mechanisms as score estimators.
>
> > **Reviewer's Question 3:** In the right-hand side of equation (3.1), it seems that the second term does not involve and thus will not goes to zero even when goes to infinity. Does it mean that the estimation can never be consistent?
>
> **Response:**
> Appreciate your question. Here are the clarifications.
>
> Yes.  This stems from the double exponential factor $2^{\epsilon^{-2L}}$ mentioned in Remark 3.4, where $\epsilon$ denotes any given approximation error. We plan to explore the possibilities noted in Remark 3.4 to avoid the double exponential factor. By doing so, we can choose $\epsilon$ as a function of the variable $n$ to balance the first and second terms on the right-hand side of the equation (3.1). This approach is expected to lead to consistent estimation.
>
> ---
>
> > **Reviewer's Comment 1:** The practical insights of the provably efficient criteria (Section 4) are relatively scarce.
>
> **Response:**
> Thanks for your comment. We acknolodge the importance of enhancing the applicability of our findings. Here are some clarifications.
>
> Our provably efficient criteria offer some insights for the design of more efficient methods. Specifically, we demonstrate that:
>
> * The latent dim should be small enough $d=O(\log L)$ (Thm 4.1 & Prop 4.1, 4.2)
> * Normalization of $K,Q,V$ in DiT attention heads is beneficial for performance and efficiency. For example:
>   * For efficiency inference: $\max{\|| W_KA_1\||,\||W_QA_2\||,\||W_{OV}A_3\|| }\le B$ with $B=o(\sqrt{\log L})$ (Prop 4.2)
>   * For efficiency training: $\max{\||W_KA_1\||,\||W_QA_2\||,\||W_{OV}A_3\|| }\le \Gamma$ with $\Gamma=o(\sqrt{\log L})$ (Thm 4.1)
>
> We want to emphasize that these conditions are necessary but not sufficient. Sufficient conditions should depend on the detailed designs of specific methods.
>
> We hope these points address your concern and enhance the utility of our criteria in practical settings.
>
> > **Reviewer's Comment 2:** A few noticeable typos were encountered in the paper.
>
> **Response:**
> Thanks for the comment. We apologize for the typos. In response, we have conducted 3 more rounds of proofreading and fixed all typos identified in our latest version. This includes:
> * `line 109`, we corrected “denosing” to “denoising”.
> * `line 120`, we corrected “knonw” to “known”.
> * `line 155`, we corrected “a” to “an”.
> * `line 192`, we corrected “an” to “a”.
> * `line 228`, we corrected “depend” to “depends”.
> * `line 242`, we deleted the extra “full”.
> * `line 253`, we corrected “analyze” to “analyzing”.
> * `line 303`, we corrected “exists” to “exist”.
> * `line 320`, we corrected “motivate” to “motivates”.
> * `line 322`, we corrected “do” to “does”.
> * `line 954`, we corrected “subset” to “subsets”.
> * `line 977`, we corrected “part” to “parts”.
> * `line 989`, we corrected “a” to “an”.
> * `line 1168`, we added “at” after “arrive”.
>
> Your feedback is greatly appreciated and has been essential in enhancing the clarity and readability of our paper. Thank you for your help.
>
> ---
>
>
> We hope the revisions and clarifications provided in this response address the reviewer's concerns.
>
> We welcome additional feedback and look forward to further discussions. Thank you for your time and valuable inputs!
>
> ---
>
> * [Chen23] Minshuo Chen, Kaixuan Huang, Tuo Zhao, and Mengdi Wang. Score approximation, estimation and distribution recovery of diffusion models on low-dimensional data. In International Conference on Machine Learning (ICML), 2023.
> * [Oko23] Kazusato Oko, Shunta Akiyama, and Taiji Suzuki. Diffusion models are minimax optimal distribution estimators. In International Conference on Machine Learning (ICML), 2023.

---

### Comment · Area_Chair_ytVL · 2024-08-08

Dear authors, dear reviewers,

the discussion period has begun as the authors have provided their rebuttals.
I encourage the reviewers to read all the reviews and the corresponding rebuttals: the current period might be an opportunity for further clarification on the paper results and in general to engage in an open and constructive exchange.

Many thanks for your work.
The AC

---

### Decision · Program_Chairs · 2024-09-25

**Decision:**

Accept (poster)

**Comment:**

The submission focuses on statistical and computational bounds for latent Diffusion Transformers (DiTs), alongside provably efficient criteria for both forward inference and backward training. The results are obtained under the assumption that input data derive from linear latent low-dimensional space. The paper is quite dense and technical. The discussion phase helped to partially improve its form and the results might have limited practical impact: concerning these points, authors are encouraged to add numerical evidence of the obtained bounds, if possible. On the other hand, the novelty of the results and the relevance of the considered architecture are of great theoretical interest, as acknowledged during the reviewing process, and therefore I support the acceptance of the manuscript.